# Context-dependent modification of PFKFB3 in hematopoietic stem cells promotes anaerobic glycolysis and ensures stress hematopoiesis

Shintaro Watanuki[1,2†], Hiroshi Kobayashi[1,3*†], Yuki Sugiura[4,5*], Masamichi Yamamoto[6], Daiki Karigane[1,2], Kohei Shiroshita[1,2], Yuriko Sorimachi[1,7], Shinya Fujita[1,2], Takayuki Morikawa[1], Shuhei Koide[8], Motohiko Oshima[8], Akira Nishiyama[9], Koichi Murakami[9,10], Miho Haraguchi[1], Shinpei Tamaki[1], Takehiro Yamamoto[4], Tomohiro Yabushita[11], Yosuke Tanaka[12], Go Nagamatsu[13,14], Hiroaki Honda[15], Shinichiro Okamoto[2], Nobuhito Goda[7], Tomohiko Tamura[9,10], Ayako Nakamura-Ishizu[16], Makoto Suematsu[4,17], Atsushi Iwama[8], Toshio Suda[12,18], Keiyo Takubo[1,3*]

[1]Department of Stem Cell Biology, Research Institute, National Center for Global Health and Medicine, Tokyo, Japan; [2]Division of Hematology, Department of Medicine, Keio University School of Medicine, Tokyo, Japan; [3]Department of Cell Fate Biology and Stem Cell Medicine, Tohoku University Graduate School of Medicine, Sendai, Japan; [4]Department of Biochemistry, Keio University School of Medicine, Tokyo, Japan; [5]Center for Cancer Immunotherapy and Immunobiology, Kyoto University Graduate School of Medicine, Kyoto, Japan; [6]Department of Research Promotion and Management, National Cerebral and Cardiovascular Center, Osaka, Japan; [7]Department of Life Sciences and Medical BioScience, Waseda University School of Advanced Science and Engineering, Tokyo, Japan; [8]Division of Stem Cell and Molecular Medicine, Center for Stem Cell Biology and Regenerative Medicine, The Institute of Medical Science, University of Tokyo, Tokyo, Japan; [9]Department of Immunology, Yokohama City University Graduate School of Medicine, Kanagawa, Japan; [10]Advanced Medical Research Center, Yokohama City University, Kanagawa, Japan; [11]Division of Cellular Therapy, The Institute of Medical Science, The University of Tokyo, Tokyo, Japan; [12]International Research Center for Medical Sciences, Kumamoto University, Kumamoto, Japan; [13]Center for Advanced Assisted Reproductive Technologies, University of Yamanashi, Yamanashi, Japan; [14]Precursory Research for Embryonic Science and Technology, Japan Science and Technology Agency, Saitama, Japan; [15]Field of Human Disease Models, Major in Advanced Life Sciences and Medicine, Institute of Laboratory Animals, Tokyo Women's Medical University, Tokyo, Japan; [16]Department of Microscopic and Developmental Anatomy, Tokyo Women's Medical University, Tokyo, Japan; [17]Live Imaging Center, Central Institute for Experimental Animals, Kanagawa, Japan; [18]Cancer Science Institute of Singapore, National University of Singapore, Singapore, Singapore

*For correspondence:
hikobayashi.md@gmail.com (HK);
yuki.sgi@gmail.com (YS);
keiyot@gmail.com (KT)

†These authors contributed equally to this work

Competing interest: The authors declare that no competing interests exist.

**Abstract** Metabolic pathways are plastic and rapidly change in response to stress or perturbation. Current metabolic profiling techniques require lysis of many cells, complicating the tracking of metabolic changes over time after stress in rare cells such as hematopoietic stem cells (HSCs). Here,

we aimed to identify the key metabolic enzymes that define differences in glycolytic metabolism between steady-state and stress conditions in murine HSCs and elucidate their regulatory mechanisms. Through quantitative $^{13}$C metabolic flux analysis of glucose metabolism using high-sensitivity glucose tracing and mathematical modeling, we found that HSCs activate the glycolytic rate-limiting enzyme phosphofructokinase (PFK) during proliferation and oxidative phosphorylation (OXPHOS) inhibition. Real-time measurement of ATP levels in single HSCs demonstrated that proliferative stress or OXPHOS inhibition led to accelerated glycolysis via increased activity of PFKFB3, the enzyme regulating an allosteric PFK activator, within seconds to meet ATP requirements. Furthermore, varying stresses differentially activated PFKFB3 via PRMT1-dependent methylation during proliferative stress and via AMPK-dependent phosphorylation during OXPHOS inhibition. Overexpression of *Pfkfb3* induced HSC proliferation and promoted differentiated cell production, whereas inhibition or loss of *Pfkfb3* suppressed them. This study reveals the flexible and multilayered regulation of HSC glycolytic metabolism to sustain hematopoiesis under stress and provides techniques to better understand the physiological metabolism of rare hematopoietic cells.

## eLife assessment

This **important** study provides novel strategies to overcome certain limitations when investigating the metabolism of hematopoietic stem cells, mainly due to their low abundance. The study provides **compelling** evidence suggesting that proliferative hematopoietic stem cells mainly use glycolysis (rather than mitochondrial OXPHOS or TCA cycle) as their primary energy source during emergency hematopoiesis. The article provides direct links between metabolic features and cell proliferation and explores alternative energy sources, and is of great interest to stem cell biologists.

## Introduction

Activities governing nutrient requirements and metabolic pathways in individual cells maintain tissue homeostasis and respond to stress through metabolite production. ATP, produced via cytosolic glycolysis and mitochondrial oxidative phosphorylation (OXPHOS), is the universal energy currency of all organisms; it regulates all anabolic or catabolic cellular activities (*Schirmer and Evans, 1990*; *Denton et al., 1975*; *Harris et al., 1997*). Precise control of intracellular ATP concentrations is critical, as ATP is the rate determiner of many ATP-dependent biochemical reactions (*Sols, 1981*; *Gabriel et al., 1985*; *Frieden, 1965*; *Hardie et al., 2012*; *Lin and Hardie, 2018*; *Hardie and Carling, 1997*).

Hematopoietic stem cells (HSCs) are tissue stem cells at the apex of the hematopoietic hierarchy; their function is maintained throughout life by a rigorous metabolic program and a complex interplay of gene expression, epigenetic regulation, intracellular signaling, chromatin remodeling, autophagy, and environmental factors (*Pinho and Frenette, 2019*; *Crane et al., 2017*; *de Haan and Lazare, 2018*; *Mejia-Ramirez and Florian, 2020*; *Orkin and Zon, 2008*). Conventional analyses of the metabolic programs of hematopoietic stem and progenitor cells (HSPCs) have revealed diverse differentiation potentials and cell-cycling statuses and coordinated activities that maintain hematopoiesis (*Nakada et al., 2010*; *Gurumurthy et al., 2010*; *Gan et al., 2010*; *Sahin et al., 2011*; *Luchsinger et al., 2016*; *de Almeida et al., 2017*; *Ansó et al., 2017*; *Nakamura-Ishizu et al., 2018*; *Qi et al., 2021*; *Filippi and Ghaffari, 2019*; *Guitart et al., 2017*; *Hsu and Qu, 2013*). Among the HSPC fractions, HSCs possess unique cell cycle quiescence, high self-renewal and differentiation capacity in response to stimuli, and resistance to cellular stress, including reactive oxygen species and physiological aging (*Pinho and Frenette, 2019*; *Busch et al., 2015*; *Sun et al., 2014*; *Ho et al., 2017*; *Laurenti and Göttgens, 2018*). These properties are regulated by a balance between glycolysis and mitochondrial OXPHOS, requiring biosynthesis of ATP and various macromolecules that confer resilience to stress (*Nakamura-Ishizu et al., 2020*). Among the known regulators of ATP-producing pathways, glycolytic enzymes maintain HSCs and hematopoietic progenitor cells (HPCs) by regulating cellular survival and cell cycle quiescence (*Takubo et al., 2013*; *Wang et al., 2014*; *Simsek et al., 2010*). Loss of mitochondrial genes in HPSCs also induces HSC differentiation defects (*Inoue et al., 2010*; *Yu et al., 2013*; *Bejarano-García et al., 2016*). Moreover, disrupting the mitochondrial complex III subunit depletes both differentiated hematopoietic cells and quiescent HSCs (*Ansó et al., 2017*). Although glycolysis and the tricarboxylic acid (TCA) cycle are metabolically linked, pyruvate dehydrogenase kinase

activity, which can uncouple these pathways, is required to maintain HSC function (*Takubo et al., 2013*; *Halvarsson et al., 2017*).

During HSC division, cell metabolism is reprogrammed to activate fatty acid β-oxidation (FAO) and purine metabolism (*Ito et al., 2012*; *Karigane et al., 2016*; *Umemoto et al., 2022*). Furthermore, Liang et al. reported that activated HSCs mainly rely on glycolysis as their energy source (*Liang et al., 2020*). However, the mechanisms by which each ATP-producing pathway and their connections are differentially regulated between HSCs and differentiated cells at steady state, during cell cycling, or during stress remain unknown. Recently, it has been shown that deeply quiescent HSCs do not activate cell cycle under stress (*Bowling et al., 2020*; *Fanti et al., 2023*; *Munz et al., 2023*). Therefore, it remains unclear whether metabolic changes such as the individual ATP-producing pathways and their interconnections occur uniformly in all HSCs, including these deeply quiescent HSCs. Furthermore, the underlying hub metabolic enzyme responsible for changes in the metabolic system of HSCs under stress has not been identified. HSCs are essential for cell therapy, including HSC transplantation, and in order to comprehensively elucidate the metabolic systems that have attracted attention as their regulatory mechanisms, recent studies have included metabolomic analyses using rare cell types such as HSCs (*Qi et al., 2021*; *Agathocleous et al., 2017*; *DeVilbiss et al., 2021*; *Lengefeld et al., 2021*; *Schönberger et al., 2022*), as well as isotope tracer analyses of undifferentiated hematopoietic cells purified after in vivo administration of isotopic glucose (*Jun et al., 2021*). Although these approaches are useful for obtaining comprehensive information on intracellular metabolites, they are not suited to track real-time changes in cellular metabolism at high resolution. Therefore, new approaches are necessary to analyze metabolites quantitatively and continuously without disturbing the physiological states of single cells while integrating the recently reported metabolome analysis techniques. In this study, we aimed to identify the key metabolic enzymes that define differences in glycolytic metabolism between steady-state and stress conditions in HSCs and elucidate their regulatory mechanisms using a quantitative and mathematical approach. Our findings provide a platform for quantitative metabolic analysis of rare cells such as HSCs, characterize the overall metabolic reprogramming of HSCs during stress loading, and highlight the key enzyme involved in this process.

## Results

### HSC cell cycling increases anaerobic glycolytic flux

To determine how cell cycle progression alters HSC metabolism in vivo, we intraperitoneally and intravenously treated mice with 5-fluorouracil (5-FU) to induce HSC cell cycling (*Figure 1—figure supplement 1A*). For analysis after 5-FU administration, the Lineage (Lin)$^-$ Sca-1$^+$ c-Kit$^+$ (LSK) gate was expanded to include HSCs with decreased c-Kit expression levels early after 5-FU treatment, for example high Sca-1-expressing cells and c-Kit-high to -dim Lin$^-$ cells, based on the previous report (*Arai et al., 2004*; *Umemoto et al., 2022*; *Figure 1—figure supplement 1B*). This expanded LSK gate was consistent with the patterns of c-Kit expression observed in endothelial protein C receptor (EPCR)$^+$ Lin$^-$ CD150$^+$ CD48$^-$ cells (*Figure 1—figure supplement 1C*) with high stem cell activity after 5-FU administration (*Umemoto et al., 2022*). We observed a transient decrease in the number of quiescent HSCs (Ki67$^-$) and an increase in the number of cell-cycling HSCs (Ki67$^+$) on day 6 after 5-FU treatment (*Figure 1—figure supplement 1D*). Along with the loss of cell quiescence, ATP concentration in HSCs decreased transiently on day 6 (*Figure 1—figure supplement 1E*). Because the route of administration of 5-FU (intraperitoneal or intravenous) made no difference in the Ki67 positivity rate of HSCs (*Figure 1—figure supplement 1F*), we administered 5-FU intraperitoneally for remaining experiments. Two methods were used to test whether cell cycle progression of HSCs after 5-FU treatment depends on the expression of EPCR. First, phosphorylation of Rb (pRb), a marker of cell cycle progression (*Miller et al., 2018*), was analyzed in HSCs after 5-FU treatment. Analysis of EPCR$^+$ and EPCR$^-$ HSCs showed increased pRb in HSCs from 5-FU-treated mice in both fractions compared to HSCs from phosphate-buffered saline (PBS)-treated mice, regardless of EPCR expression (*Figure 1—figure supplement 1G–H*). Second, we used a G$_0$ marker mouse line (*Fukushima et al., 2019*). These mice expressed a fusion protein of the p27 inactivation mutant p27K$^-$ and the fluorescent protein mVenus (G$_0$ marker), allowing prospective identification of G$_0$ cells. We tested whether the expression of G$_0$ marker in HSCs was altered after 5-FU administration to the G$_0$ marker mice (*Figure 1—figure supplement 1I*) and found that 5-FU treatment reduced the frequency of G$_0$ marker-positive HSCs,

regardless of the EPCR expression (*Figure 1—figure supplement 1J–K*). This was not observed in the PBS group. These results indicated that 5-FU administration induced cell cycle progression of entire HSCs in mice.

HSC cell cycling is preceded by the activation of intracellular ATP-related pathways that metabolize extracellular nutrients, including glucose (*Ito et al., 2012*; *Karigane et al., 2016*), which are utilized in both ATP-producing and -consuming pathways, determining cellular ATP levels. Therefore, we examined the metabolic flux of glucose by performing in vitro IC-MS tracer analysis with uniformly carbon-labeled (U-$^{13}$C$_6$) glucose to determine the pathways driving changes in ATP in 5-FU-treated HSCs (*Figure 1A*; *Supplementary file 2*). To avoid metabolite changes, samples were continuously chilled on ice during cell preparation, and the process from euthanasia to cell preparation was performed in the shortest possible time (see 'Preparation and storage of in vitro U-$^{13}$C$_6$-glucose tracer samples' section under 'Materials and methods' for more information). We found that changes in metabolite levels before and after sorting were present but limited (*Figure 1—figure supplement 2A*). This result is consistent with the finding that the cell purification process does not significantly affect metabolite levels when sufficient care is taken in cell preparation (*Jun et al., 2021*). In 5-FU-treated HSCs, the levels of glycolytic metabolites derived from U-$^{13}$C$_6$-glucose were double those observed in PBS-treated HSCs (*Figure 1B–C*; *Figure 1—figure supplement 2B*). The total levels of TCA cycle intermediates derived from U-$^{13}$C$_6$-glucose were similar between PBS- and 5-FU-treated cells (*Figure 1D*; *Figure 1—figure supplement 2B*). Levels of U-$^{13}$C$_6$-glucose-derived intermediates involved in the pentose phosphate pathway (PPP) and nucleic acid synthesis (NAS) were twofold higher in 5-FU-treated than in PBS-treated HSCs, whereas no significant differences in the levels of metabolites were observed between both groups (*Figure 1E–F*; *Figure 1—figure supplement 2B*). Notably, the labeling rate of metabolites during the first half of glycolysis was almost 100% in both groups, allowing us to easily track the labeled metabolites (*Figure 1—figure supplement 2C–E*). This was thought to be due to the rapid replacement of unlabeled metabolites with labeled metabolites during exposure to U-$^{13}$C$_6$-glucose because of the generally rapid glycolytic reaction. Conversely, the labeling rate of TCA cycle intermediates was consistently lower than that of glycolysis and PPP (*Figure 1—figure supplement 2D*), suggesting that PBS- and 5-FU-treated HSCs prefer anaerobic glycolysis over aerobic glycolysis. To directly compare the metabolic systems of PBS- or 5-FU-treated HSCs, we conducted a Mito stress test using a Seahorse flux analyzer. Compared to PBS-treated HSCs, 5-FU-treated HSCs exhibited a higher extracellular acidification rate (ECAR), while their oxygen consumption rate (OCR) remained equal to that of PBS-treated HSCs (*Figure 1G–H*; *Figure 1—figure supplement 3A–B*). After oligomycin treatment, PBS- and 5-FU-treated HSCs showed an increase in ECAR, suggesting a flexible activation of glycolysis upon OXPHOS inhibition (*Figure 1G*; *Figure 1—figure supplement 3A*). Meanwhile, a decrease in OCR was more clearly observed in the 5-FU-treated HSCs (*Figure 1H*; *Figure 1—figure supplement 3B*). Next, we evaluated whether glucose uptake in HSCs after 5-FU administration was differentially affected by the expression of EPCR. The fluorescent analog of glucose, 2-(N-(7-nitrobenz-2-oxa-1,3-diazol-4-yl)amino)–2-deoxyglucose (2-NBDG), was administered intravenously to mice (*Jun et al., 2021*) and its uptake in EPCR$^+$ and EPCR$^-$ HSCs was assayed (*Figure 1I*). Regardless of the EPCR expression, the 2-NBDG uptake was greater in HSCs treated with 5-FU than in those treated with PBS (*Figure 1J–L*). Increased 2-NBDG uptake in 5-FU-treated HSCs was also observed in an in vitro 2-NBDG assay (*Figure 1—figure supplement 1L*). Notably, even in the PBS-treated group, HSCs with high NBDG uptake were more proliferative than those with low NBDG uptake, similar to the state of HSCs after 5-FU administration (*Figure 1—figure supplement 1M*). After 5-FU administration, there was an overall shift of the population from the G$_0$ to G$_1$ phase and a correlation between NBDG uptake and cell cycle progression was also observed (*Figure 1—figure supplement 1M*). In both PBS- and 5-FU-treated groups, the marked variation in glucose utilization depending on the cell cycle suggests a direct link between HSC proliferation and increased glycolytic activity. Furthermore, compared to HSCs cultured under the quiescence-maintaining conditions of HSC achieved by hypoxia, abundant fatty acids, and low cytokines as we previously reported (*Kobayashi et al., 2019*), HSCs cultured under cytokine-rich proliferative conditions were more resistant to the inhibition of OXPHOS by oligomycin (*Figure 1—figure supplement 1N*; *Supplementary file 1*). Overall, the results showed that 5-FU-treated HSCs exhibited activated glycolytic flux, increasing the turnover of ATP. Moreover, glycolytic flux into mitochondria was equally unchanged in PBS- and 5-FU-treated-HSCs, supporting that 5-FU activated anaerobic glycolysis in HSCs.

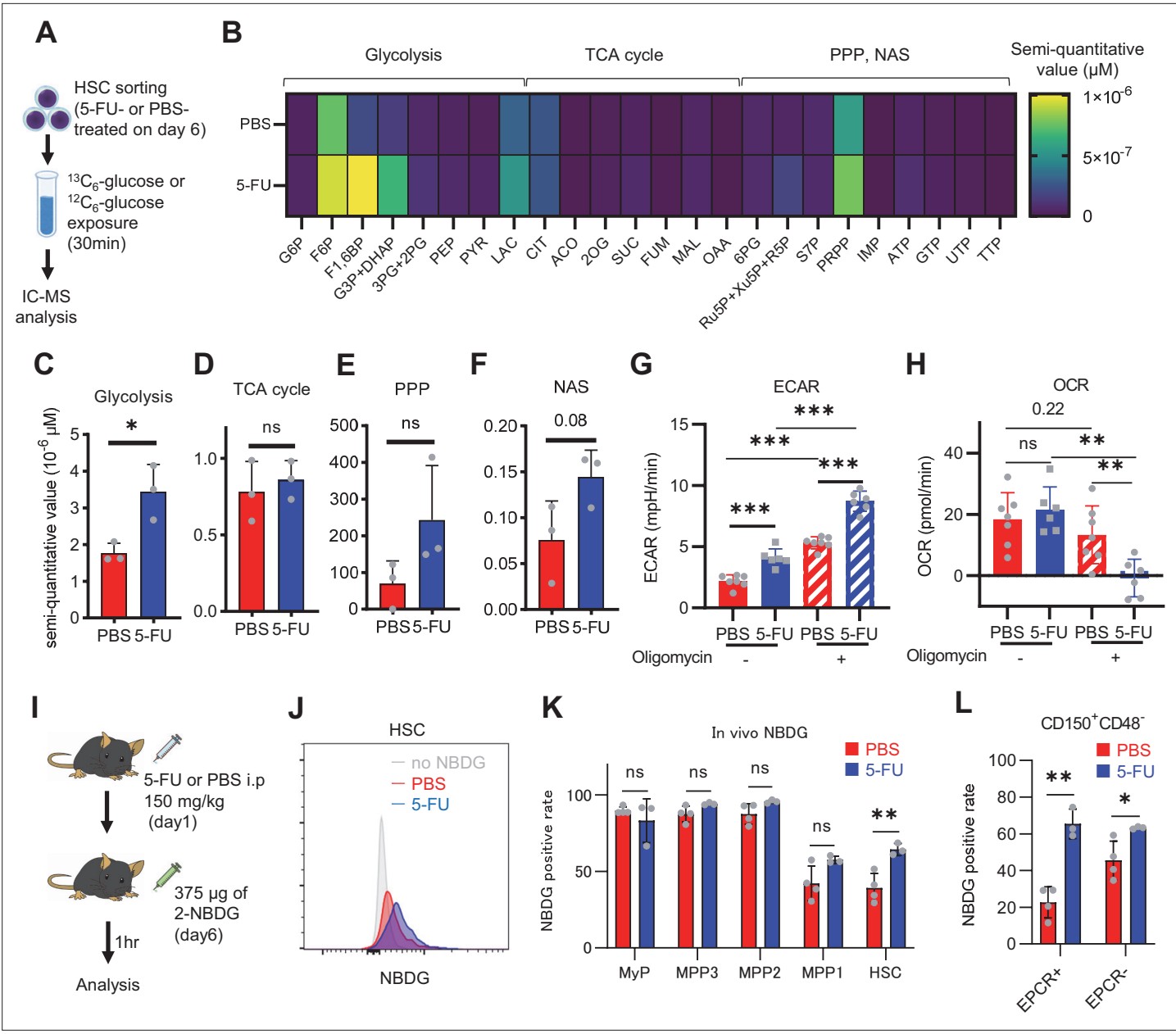

**Figure 1.** HSC cell cycling increases overall glycolytic flux, but not flux into mitochondria. (**A**) Experimental design used for glucose isotope tracer analysis in HSCs from 5-FU- or PBS-treated mice. (**B**) Heat map of metabolite levels in HSCs derived from mice treated with PBS or 5-FU. (**C–F**) The semi-quantitative value ($10^{-6}$ μM) of U-$^{13}C_6$-glucose-derived metabolites in glycolysis (**C**), the first round of TCA cycle (**D**), the PPP, and nucleotide synthesis (**F**) in HSCs from 5-FU- or PBS-treated mice (PBS group = 1.0); In (**B-F**), biological replicates from the PBS and 5-FU groups, obtained on three separate days, were pooled, analyzed by IC-MS, quantified based on calibration curve data for each metabolite (see 'Ion chromatography mass spectrometry (IC-MS) analysis' section in 'Materials and methods' for details). (**G–H**) A Mito Stress test with the Seahorse flux analyzer on HSCs derived from mice treated with PBS or 5-FU; ECAR (**G**) and OCR (**H**) before and after oligomycin treatment. (Data were obtained from n=7 technical replicates for PBS-treated HSCs and n=6 for 5-FU-treated HSCs.) (**I**) Experimental schema of in vivo 2-NBDG analysis. (**J**) Representative histograms of 2-NBDG analysis (gray: no 2-NBDG, red: PBS group, blue: 5-FU group). (**K**) 2-NBDG positivity in each fraction; data represent four pooled biological replicates for the PBS group and three for the 5-FU group; MyP: myeloid progenitor. (**L**) EPCR expression and 2-NBDG positivity within HSC fractions. Data were extracted from each individual in (**K**). Data are presented as mean ± SD. * p≤0.05, ** p≤0.01, *** p≤0.001 as determined by Student's *t*-test (**C–F, G–H** when comparing the PBS and 5-FU groups, and **K–L**) or paired-samples t-test (**G–H** when comparing the conditions before and after exposure to oligomycin within the PBS/5-FU group). Abbreviations: G6P, glucose-6-phosphate; F6P, fructose-6-phosphate; F1,6BP, fructose-1,6-bisphosphate; G3P, glycerol-3-phosphate; DHAP, dihydroxyacetone phosphate; 3 PG, 3-phosphoglycerate; 2 PG, 2-phosphoglycerate; PEP, phosphoenolpyruvate; PYR, pyruvate; LAC, lactate; Ac-CoA; acetyl-CoA; CIT, citrate; ACO, cis-aconitic acid, isocitrate; 2OG, 2-oxoglutarate; SUC, succinate; FUM, fumarate; MAL, malate; OAA, oxaloacetate; 6 PG, 6-phosphogluconate; Ru5P, ribulose-5-phosphate; Xu5P, xylulose-5-phosphate; R5P, ribose-5-phosphate; S7P, sedoheptulose-7-phosphate; E4P,

*Figure 1 continued on next page*

Figure 1 continued

erythrose-4-phosphate; PRPP, phosphoribosyl pyrophosphate; IMP, inosine monophosphate; ATP, adenosine triphosphate; GTP, guanine triphosphate; UMP, uridine monophosphate; UTP, uridine triphosphate; TTP, thymidine triphosphate. See also *Figure 1—figure supplements 1–3*.

The online version of this article includes the following source data and figure supplement(s) for figure 1:

**Source data 1.** Raw data for *Figure 1B–H, K and L*.

**Figure supplement 1.** Dependence on glycolysis increases with cell cycle progression of HSCs.

**Figure supplement 1—source data 1.** Raw data for *Figure 1—figure supplement 1D–F, H, K-N*.

**Figure supplement 2.** Quantified metabolite pool in HSCs under quiescence, proliferation, or OXPHOS-inhibition.

**Figure supplement 2—source data 1.** Raw data for *Figure 1—figure supplement 2A–H*.

**Figure supplement 3.** Mito Stress test results using the Seahorse flux analyzer.

**Figure supplement 3—source data 1.** Raw data for *Figure 1—figure supplement 3A–D*.

## OXPHOS-inhibited HSCs exhibit compensatory glycolytic flux

Previous studies using mouse models of mitochondrial disease or defects in genes involved in electron transport chain and OXPHOS suggest that mitochondrial energy production is essential for maintaining HSC function (*Ansó et al., 2017*; *Inoue et al., 2010*; *Yu et al., 2013*; *Bejarano-García et al., 2016*), as is the glycolytic system. However, there have been no quantitative reports on how OXPHOS-inhibited HSCs can adapt their metabolism. To understand HSC metabolism under OXPHOS inhibition, we performed in vitro U-$^{13}$C$_6$-glucose tracer analysis of oligomycin-treated HSCs (*Figure 2A*; *Supplementary file 3*). Similar to 5-FU-treated HSCs (*Figure 1*), oligomycin-treated HSCs exhibited glycolytic system activation (*Figure 2B–C*; *Figure 1—figure supplement 2B*). Metabolite flux to the TCA cycle and PPP was unchanged, but flux to the NAS was significantly reduced in oligomycin-treated HSCs compared to that in steady-state HSCs (*Figure 2D–F*; *Figure 1—figure supplement 2B*). The results suggested that OXPHOS-inhibited HSCs activated compensatory glycolytic flux and suppressed NAS flux. As with 5-FU-treated HSCs, analysis of oligomycin-treated HSCs also showed almost 100% labeling of metabolites in the first half of glycolysis (*Figure 1—figure supplement 2F–H*), allowing us to easily track the labeled metabolites. To further validate the compensatory glycolytic activation of HSCs under OXPHOS inhibition, a Mito Stress test was performed on HSCs and other differentiated myeloid progenitors (MyPs, Lin$^-$Sca-1$^-$c-Kit$^+$ (LKS$^-$) cells). The results showed that ECAR were elevated in HSCs after oligomycin treatment compared to before oligomycin treatment (*Figure 2G*; *Figure 1—figure supplement 3C*). No increase in ECAR was observed in MyPs (*Figure 2G*; *Figure 1—figure supplement 3C*), supporting that inhibition of OXPHOS activated anaerobic glycolysis specifically in HSCs. Meanwhile, in HSCs, the decrease in OCR after oligomycin administration was less evident compared to MyPs (*Figure 2H*; *Figure 1—figure supplement 3D*). In MyPs, both ECAR and OCR were downregulated (*Figure 2G–H*; *Figure 1—figure supplement 3C–D*).

## Phosphofructokinase (PFK) metabolism in HSCs is activated during proliferation and OXPHOS inhibition

To investigate whether glycolytic activation in HSCs after 5-FU treatment and OXPHOS inhibition could be demonstrated through unbiased mathematical simulations, we performed quantitative $^{13}$C metabolic flux analysis ($^{13}$C-MFA). After generating a metabolic model for isotope labeling enrichment and setting appropriate lactate efflux values, a simulation was conducted using the labeled metabolite abundance data obtained from isotope tracer analysis. The appropriate lactate efflux for quiescent HSC (PBS-treated HSC) was determined to 65 after experimenting with values from 0 to 100. The lactate efflux of 5-FU- or oligomycin-treated HSCs was higher than that of quiescent HSCs based on the observation that labeled glycolytic metabolite levels were particularly elevated in in vitro tracer analysis (see 'Quantitative $^{13}$C-MFA with OpenMebius' under 'Materials and methods' for more information). As a result, the variation in the flux values of all enzymatic reactions calculated in HSCs after 5-FU or oligomycin treatment became smaller compared to quiescent HSCs, suggesting that HSCs strictly regulated their metabolism in response to stress (*Figure 3—figure supplement 1A–C*). Unlike PBS-treated HSCs, those treated with 5-FU or oligomycin exhibited preferential glycolytic activation rather than TCA- or PPP-based metabolic strategies; the first half of the glycolytic system appeared to be the site of metabolic activation (*Figure 3A–J*; *Figure 3—figure supplement 1D–U*, *Supplementary*

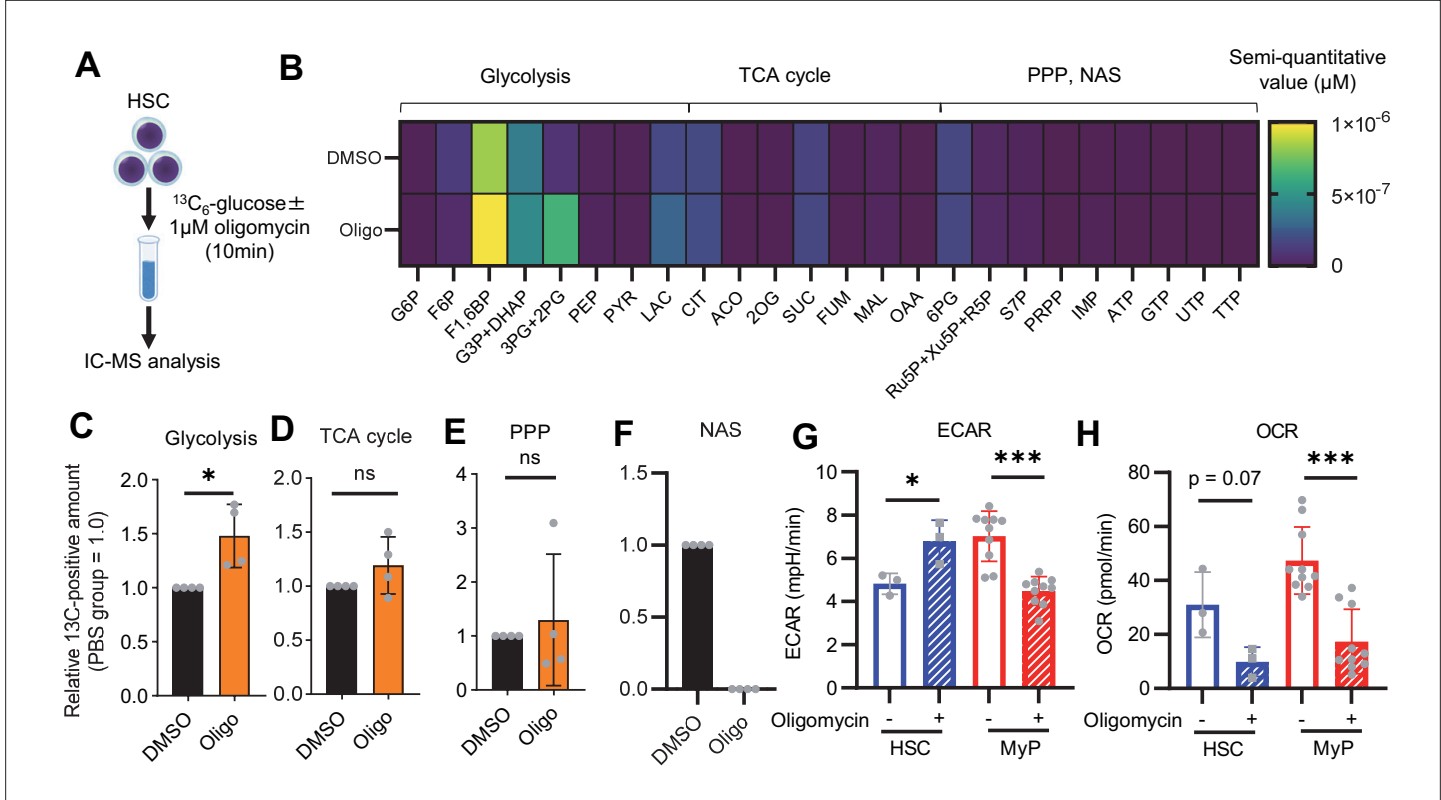

**Figure 2.** OXPHOS inhibition activates compensatory glycolysis in HSCs. (**A**) Experimental design used for glucose isotope tracer analysis in HSCs treated with the OXPHOS inhibitor oligomycin. (**B**) Heat map of metabolite levels detected by in vitro tracer analysis of U-$^{13}$C$_6$-glucose in HSCs treated with DMSO or oligomycin (Oligo). (**C–F**) Relative amounts of U-$^{13}$C$_6$-glucose-derived metabolites in glycolysis (**C**), the first round of TCA cycle (**D**), the PPP(E), and nucleotide synthesis (**F**) in DMSO- (black) or oligomycin-treated (orange) HSCs; In (**B-F**), biological replicates of the DMSO and oligomycin groups obtained on four separate days were pooled, analyzed by IC-MS, and quantified based on calibration curve data for each metabolite (see 'Ion chromatography mass spectrometry (IC-MS) analysis' section in 'Materials and methods' for details). (**G–H**) Mito Stress test on the Seahorse flux analyzer for HSC and MyPs; ECAR (**G**) and OCR (**H**) before and after oligomycin treatment. (Data were obtained from n=3 technical replicates for HSCs and n=10 technical replicates for MyPs.). Data are shown as mean ± SD. * p≤0.05, ** p≤0.01, *** p≤0.001 as determined by paired-samples t-test (**C-E and G–H**). Abbreviations: G6P, glucose-6-phosphate; F6P, fructose-6-phosphate; F1,6BP, fructose-1,6-bisphosphate; G3P, glycerol-3-phosphate; DHAP, dihydroxyacetone phosphate; 3 PG, 3-phosphoglycerate; 2 PG, 2-phosphoglycerate; PEP, phosphoenolpyruvate; PYR, pyruvate; LAC, lactate; Ac-CoA; acetyl-CoA; CIT, citrate; ACO, cis-aconitic acid, isocitrate; 2OG, 2-oxoglutarate; SUC, succinate; FUM, fumarate; MAL, malate; OAA, oxaloacetate; 6 PG, 6-phosphogluconate; Ru5P, ribulose-5-phosphate; Xu5P, xylulose-5-phosphate; R5P, ribose-5-phosphate; S7P, sedoheptulose-7-phosphate; E4P, erythrose-4-phosphate; PRPP, phosphoribosyl pyrophosphate; IMP, inosine monophosphate; ATP, adenosine triphosphate; GTP, guanine triphosphate; UMP, uridine monophosphate; UTP, uridine triphosphate; TTP, thymidine triphosphate. See also *Figure 1—figure supplements 1–3*.

The online version of this article includes the following source data for figure 2:

**Source data 1.** Raw data for *Figure 2B–H*.

*file 4*). This increase in metabolic flux upstream of the glycolytic pathway was also supported by our in vitro tracer analysis (*Figure 1B* and *Figure 2B*), suggesting that $^{13}$C-MFA was a valid metabolic simulation. Among the reactions in the first half of glycolysis, phosphorylation of fructose 6-phosphate (F6P) by PFK is the irreversible and rate-limiting reaction (*Dunaway, 1983*). A detailed review of in vitro isotope tracer analysis results showed that the ratio of fructose 1,6-bisphosphate (F1,6BP; the product of PFK) to F6P (the substrate of PFK) was greatly elevated in HSCs during proliferation and OXPHOS inhibition (*Figure 3K–L*). Together with the results of quantitative $^{13}$C-MFA, these findings suggested that HSCs exhibit elevated glycolytic flux relative to mitochondrial activity by increasing PFK enzyme activity under various stress conditions.

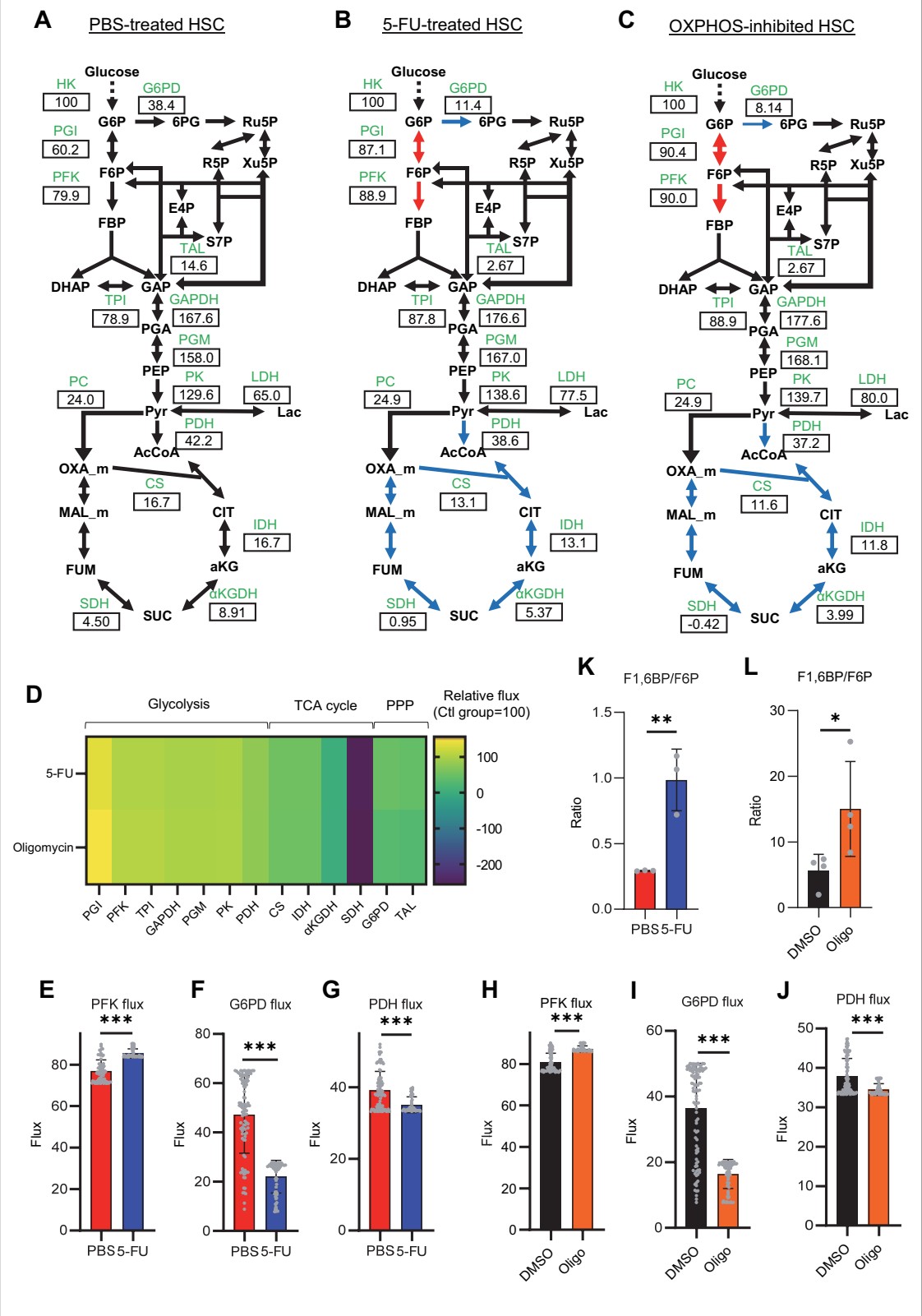

**Figure 3.** Quantitative [13]C-MFA of quiescent, proliferative, and stressed HSCs. (**A–C**) Overview of quantitative [13]C-MFA of PBS-treated HSCs (**A**), 5-FU-treated HSCs (**B**), and OXPHOS-inhibited HSCs (**C**). The representative net flux for each reaction with glucose uptake as 100 is shown in the squares below the catalytic enzymes for each reaction listed in green letters. Red arrows indicate reactions with particularly elevated fluxes and blue arrows indicate reactions with particularly decreased fluxes. (**D**) Heatmap of the relative flux of each enzyme in the 5-FU or oligomycin groups compared to

*Figure 3 continued on next page*

*Figure 3 continued*

that in the quiescent (Ctl) HSC (The metabolic flux of each enzyme in the Ctl group was standardized as 100.). (**E–J**) Fluxes due to reactions with PFK (**E, H**), G6PD (**F, I**), and PDH (**G, J**). Fluxes of HSCs derived from mice treated with 5-FU (blue bars) or PBS (red bars) (**D–F**) and of HSCs treated with DMSO (black bars) or oligomycin (orange bars) (**G–I**) are shown. Data is obtained from 100 simulations in OpenMebius, and flux data for each enzyme is displayed (*Supplementary file 4*). (**K–L**) Ratio of fructose 1,6-bisphosphate (F1,6BP) to fructose-6-phosphate (F6P) calculated from tracer experiments shown in *Figure 1B* and *Figure 2B*. Effects of 5-FU administration (**K**) or mitochondrial inhibition by oligomycin (**L**) are summarized. Data are shown as mean ± SD. * $p \leq 0.05$, ** $p \leq 0.01$, *** $p \leq 0.001$ as determined by Student's *t*-test (**E–L**). Abbreviations: HK, hexokinase; PGI, glucose-6-phosphate isomerase; PFK, phosphofructokinase; TPI, triose phosphate isomerase; GAPDH, glyceraldehyde-3-phosphate dehydrogenase; PGM, phosphoglycerate mutase; PK, pyruvate kinase; LDH, lactate dehydrogenase; PC, pyruvate carboxylase; PDH, pyruvate dehydrogenase; CS; citrate synthase; IDH, isocitrate dehydrogenase; αKGDH, α-ketoglutaric acid dehydrogenase; SDH, succinate dehydrogenase; G6PD, glucose-6-phosphate dehydrogenase; TAL, transaldolase. See also *Figure 3—figure supplements 1–2*.

The online version of this article includes the following source data and figure supplement(s) for figure 3:

Source data 1. Raw data for *Figure 3D–L*.

Figure supplement 1. Quantitative [13]C-MFA of HSCs under quiescence, proliferation, and OXPHOS inhibition.

Figure supplement 1—source data 1. Raw data for *Figure 3—figure supplement 1D–U*.

Figure supplement 2. Quantified metabolite pool in HSCs from PBS- or 5-FU-treated mice.

Figure supplement 2—source data 1. Raw data for *Figure 3—figure supplement 2B–J*.

## HSCs under stress exhibit activation of glycolysis-initiated TCA cycle and NAS

To investigate the long-term glucose utilization of HSCs, we performed an in vivo tracer analysis with U-[13]$C_6$ glucose based on recent reports (*DeVilbiss et al., 2021*; *Jun et al., 2021*; *Figure 3—figure supplement 2A*; see 'Preparation and storage of in vivo U-[13]$C_6$-glucose tracer samples' under 'Materials and methods' for more information). In HSCs from 5-FU-treated mice, we observed increased labeling of glycolytic metabolites such as dihydroxyacetone phosphate, glycerol-3-phosphate, and phosphoenolpyruvate, as well as NAS metabolites such as inosine monophosphate and ATP, and those derived from TCA cycle such as aspartic acid and glutamate, compared to HSCs from PBS-treated mice (*Figure 3—figure supplement 2B–I*, *Supplementary file 5*). When the amount of U-[13]$C_6$-glucose-derived labeled metabolites in each pathway was calculated, more glucose-derived metabolites entered TCA cycle in the 5-FU-treated group than PBS-treated group (*Figure 3—figure supplement 2J*). Thus, although short-term (10–30 min) in vitro tracer analysis showed that HSCs exhibited more potent activation of anaerobic glycolysis than of other pathways in response to 5-FU administration, long-term (approximately 3 hr) labeling by in vivo tracer analysis revealed that glycolysis-initiated TCA cycle and NAS flux were activated in addition to enhanced anaerobic glycolysis. Importantly, despite differences in labeling times and supplementation of U-[13]$C_6$ glucose metabolites from non-HSCs to HSCs in vivo, the activation of the glycolytic system was a common finding.

## PFKFB3 accelerates glycolytic ATP production during HSC cell cycling

In vitro and in vivo tracer analysis results collectively suggested that the activation of glycolysis catalyzed by PFK may have been the starting point for the activation of the entire HSC metabolism. To analyze the contribution of PFK to ATP metabolism in steady-state or stressed HSCs, we needed to develop an experimental system that could measure the dynamics of ATP concentrations in HSCs in a non-destructive, real-time manner. To this end, we used knock-in GO-ATeam2 mice as a FRET-based biosensor of ATP concentration (see 'Conversion of GO-ATeam2 fluorescence to ATP concentration' under 'Materials and methods' for more information.). The number of bone marrow mononuclear cells (BMMNCs), as well as the frequency of HSCs (CD150[+]CD48[-]LSK) and other progenitor cells, in the bone marrow (BM) of GO-ATeam2[+] mice were almost unchanged compared to C57BL/6J mice, except for a mild decrease in the Lin[-] fraction (*Figure 4—figure supplement 1A–C*). Using BMMNCs derived from GO-ATeam2[+] mice, we developed a method to detect changes in ATP concentration with high temporal resolution when the activity of PFK was modulated (*Figure 4—figure supplement 1D–F*). To validate our methods, we measured ATP concentrations in HSCs and MyPs with or without various nutrients (see 'Time-course analysis of FRET values' under 'Materials and methods' for more information.). MyPs showed more rapid decreases in ATP concentration than HSCs, suggesting higher ATP consumption by progenitors (*Figure 4—figure supplement 1G–H*). Adding glucose to the medium

suppressed this decrease in MyPs; however, other metabolites (e.g. pyruvate, lactate, and fatty acids) had minimal effects, suggesting that ATP levels are glycolysis-dependent in MyPs (*Figure 4—figure supplement 1G–H*), consistent with previous reports that the aerobic glycolytic enzyme M2 pyruvate kinase isoform (PKM2) is required for progenitor cell function (*Wang et al., 2014*).

Further, we analyzed ATP consumption and metabolic dependency of cell-cycling HSCs after 5-FU administration (*Figure 4A*). After inhibiting glycolysis using 2-deoxy-D-glucose (2-DG) with other mitochondrial substrates, 5-FU-treated HSCs showed more rapid decreases in ATP concentration than PBS-treated HSCs (*Figure 4B–C*). In contrast, OXPHOS inhibition by oligomycin without glucose or mitochondrial substrates decreased the ATP concentration to a similar extent in both 5-FU- and PBS-treated HSCs, although 5-FU-treated HSCs showed earlier ATP exhaustion (*Figure 4D–E*). These data suggest that 5-FU-treated-HSCs upregulated ATP production via glycolysis, rather than relying on mitochondria. Apoptosis assay revealed a slight increase in early apoptotic cells (annexin V$^+$ propidium iodide [PI]$^-$) after 2-DG treatment and a slight decrease in the number of viable cells (Annexin V$^-$ PI$^-$) after oligomycin treatment, both to a very limited extent (approximately 5%) compared to the degree of ATP decrease, suggesting that the decrease in ATP after 2-DG or oligomycin treatment did not simply reflect cell death (*Figure 4—figure supplement 1I*). Importantly, no metabolic changes in glycolysis or OXPHOS were observed in HSCs without cell cycle progression after 5-FU administration (very early phase: day 3; late phase: day 15) (*Figure 4—figure supplement 2A–H*).

PFK is allosterically activated by 6-phosphofructo-2-kinase/fructose-2,6-bisphosphatase (PFKFB). Among the four isozymes of mammalian PFKFB, PFKFB3 is the most favorable for PFK activation (*Yalcin et al., 2009*), and is the most highly expressed in HSCs (*Figure 4F*). Therefore, we investigated whether PFKFB3 contributes to glycolytic plasticity in HSCs during proliferation. When treated with the PFKFB3-specific inhibitor AZ PFKFB3 26 (*Boyd et al., 2015*), compared with HSCs from PBS-treated mice, HSCs from 5-FU-treated mice showed decreased ATP levels (*Figure 4G, I*; *Figure 4—figure supplement 2I*). Although AMPK activates PFKFB3 in other contexts (*Marsin et al., 2002*), AMPK inhibition by dorsomorphin did not alter ATP concentration in 5-FU-treated-HSCs (*Figure 4H and J*).

Finally, we investigated the nutrients that drive OXPHOS in PBS- or 5-FU-treated HSCs. Exposure of PBS- or 5-FU-treated HSCs to either etomoxir, a FAO inhibitor, or 6-diazo-5-oxo-L-norleucine (DON), a glutaminolysis inhibitor, alone or in combination, did not decrease ATP concentrations (*Figure 4—figure supplement 2J–M*). Subsequent assessment of FAO activity using FAOBlue, a fluorescent probe for the FAO activity assay (*Uchinomiya et al., 2020,*) showed no significant differences between PBS- and 5-FU-treated HSCs (*Figure 4—figure supplement 2N*). Thus, neither FAO nor glutaminolysis appeared to be essential for the short-term maintenance of ATP levels in cell-cycling HSCs after 5-FU administration. Notably, the addition of glucose and a PFKFB3 inhibitor to etomoxir rapidly reduced ATP concentrations in HSCs (*Figure 4—figure supplement 2O–P*). This suggests that etomoxir may partially mimic the effects of oligomycin, indicating that OXPHOS is primarily driven by FAO, but can be compensated by PFKFB3-accelerated glycolysis in HSCs. Conversely, exposure of HSCs to DON in combination with a PFKFB3 inhibitor did not decrease ATP concentrations (*Figure 4—figure supplement 2O–P*), suggesting that ATP production via glutaminolysis is limited in HSCs.

## OXPHOS inhibition accelerates glycolysis to sustain ATP levels in HSCs, but not in progenitors

To assess differences in metabolic dependence between steady-state or stressed HSCs and naturally proliferating HPCs, we altered ATP metabolism in HSCs and progenitors using 2-DG or oligomycin (*Figure 5A*). Oligomycin treatment rapidly depleted ATP in HSCs and all HPC fractions (green lines in *Figure 5B–C*; *Figure 5—figure supplement 1A–D*). Treatment with 2-DG decreased ATP concentrations for a short amount of time (~12 min) in HSCs and HPCs, but ATP reduction was less evident than that induced by oligomycin (blue lines in *Figure 5B–C*; *Figure 5—figure supplement 1A–D*). The ATP reduction induced by 2-DG treatment was particularly low (~15%) in HSCs, multipotent progenitor cells (MPPs), and common lymphoid progenitors (CLPs) relative to that in common myeloid progenitors (CMPs), granulocytes-macrophage progenitors (GMPs), and megakaryocyte-erythrocyte progenitors (MEPs; *Figure 5D*).

Next, we investigated the role of glycolysis in ATP production during OXPHOS inhibition by combining oligomycin administration and glucose supplementation. ATP concentration remained

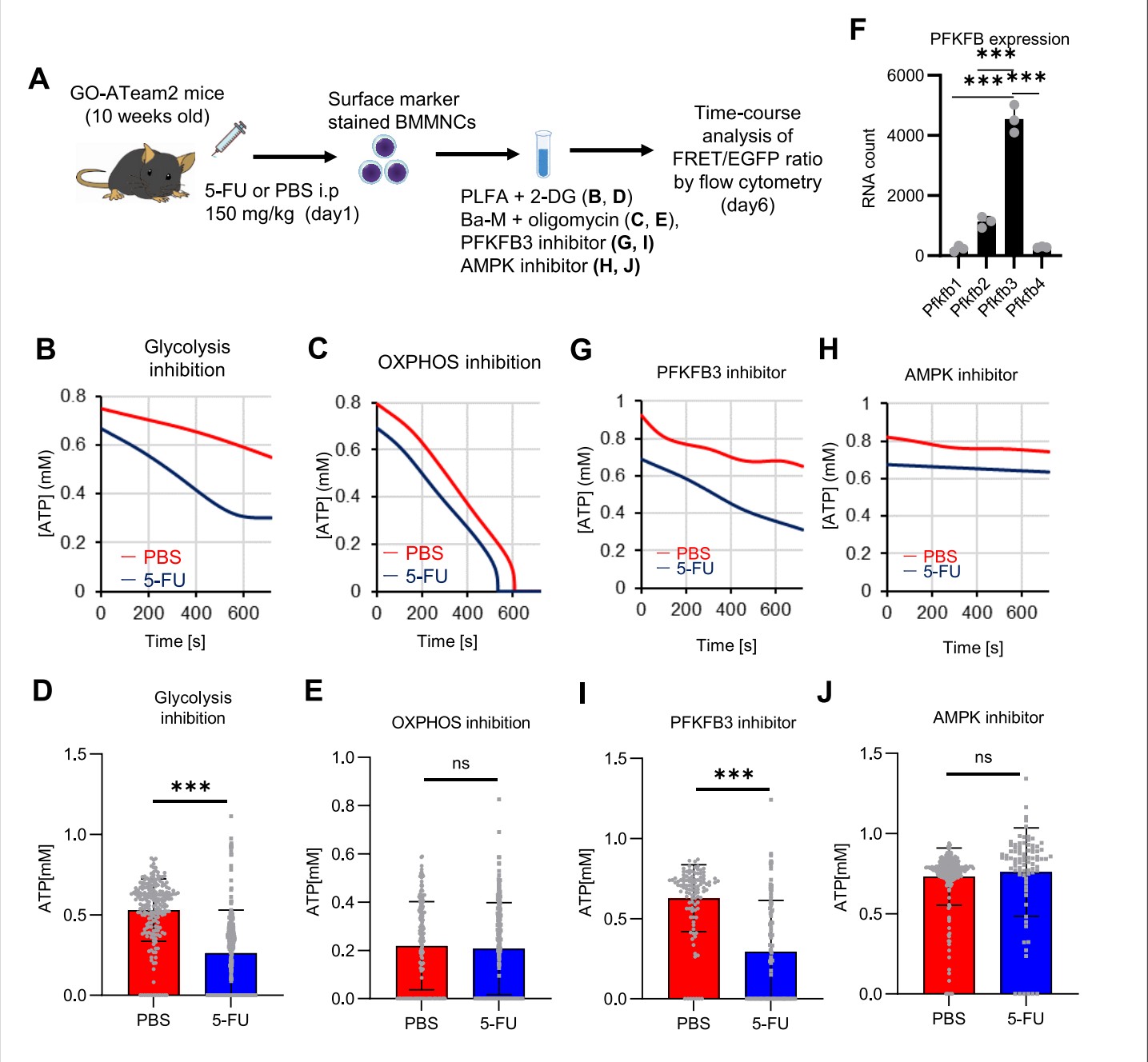

**Figure 4.** PFKFB3 activates the glycolytic system in proliferating HSCs. (**A**) Experimental design used to conduct real-time ATP analysis of HSCs treated with 5-FU or PBS. PLFA medium containing mitochondrial substrates (pyruvate, lactate, fatty acids, and amino acids) but no glucose, was used for experiments with 2-DG; Ba-M containing neither mitochondrial substrates nor glucose was used for experiments with oligomycin, PFKFB3 inhibitor, or AMPK inhibitor. (**B–E**) Results of real-time ATP analysis of PBS- (red) or 5-FU-treated (blue) HSCs after treatment with 2-DG (**B, D**), oligomycin (**C, E**). (**F**) Normalized mRNA counts of PFKFB isozymes based on the RNA sequencing of HSCs. (**G-J**) Results of real-time ATP analysis of PBS- (red) or 5-FU-treated (blue) HSCs after treatment with PFKFB3 inhibitor (**G, I**), or AMPK inhibitor (**H, J**). Bar graphs show corrected ATP concentrations for the last 2 min (**D**) of (**B**), 6–7 min (**E**) of (**C**), or the last 1 min (**I, J**) of (**G, H**) for PFKFB3 and AMPK inhibitors, respectively. Each group represents at least 60 cells. Data are representative results of pooled samples from three biological replicates. (see 'Time-course analysis of FRET values' in 'Materials and methods' for details of the correction method used to calculate ATP concentration.) Data are presented as mean ± SD. * p≤0.05, ** p≤0.01, *** p≤0.001 as determined by Student's t-test (**D, E, I, and J**) or a one-way ANOVA followed by Tukey's test (**F**). See also *Figure 4—figure supplements 1–2*.

The online version of this article includes the following source data and figure supplement(s) for figure 4:

**Source data 1.** Raw data for *Figure 4D–F, I and J*.

*Figure 4 continued on next page*

*Figure 4 continued*

**Figure supplement 1.** Establishment of a real-time ATP concentration analysis system using GO-ATeam2.

**Figure supplement 1—source data 1.** Raw data for *Figure 4—figure supplement 1B*, C and I.

**Figure supplement 2.** FAO is not active in proliferating HSCs.

**Figure supplement 2—source data 1.** Raw data for *Figure 4—figure supplement 2C, D, G-I, K, M, N and P*.

more stable in HSCs treated with oligomycin and glucose than in those treated only with oligomycin. Similar results were not seen in HPCs, indicating that HSCs have the plasticity to upregulate glycolytic ATP production to meet demands (orange lines in *Figure 5B–C*; *Figure 5—figure supplement 1A–D*, summarized in *Figure 5E*). Similar to oligomycin treatment, rotenone (complex I inhibitor) and carbonyl cyanide 4-(trifluoromethoxy)phenylhydrazone (FCCP, mitochondrial uncoupler) treatments, which inhibit OXPHOS-derived ATP production, also decreased ATP concentrations in HSCs, but not when administered simultaneously with glucose (*Figure 5—figure supplement 1E–F*). Furthermore, with oligomycin, HSCs, but not HPCs, maintained ATP concentrations at low glucose levels (50 mg/dL) (*Figure 5—figure supplement 1G*). These analyses suggest that ATP was produced by mitochondrial OXPHOS in steady-state HSCs, and that only HSCs, but not HPCs, maintained ATP production by glycolysis when OXPHOS was compromised.

## PFKFB3 accelerates glycolytic ATP production during OXPHOS inhibition

Next, to understand whether PFKFB3 contributes to ATP production in HSCs under OXPHOS inhibition, we evaluated PFKFB3 function under OXPHOS inhibition using the GO-ATeam2$^+$ BMMNCs. In oligomycin-treated HSCs, PFKFB3 inhibition led to rapidly decreased ATP concentration that was not observed in HSCs not treated with oligomycin (*Figure 5F–I*). We examined the effects of HSPC metabolic regulators on ATP levels in oligomycin-treated HSCs. Inhibiting PKM2, which accelerates glycolysis in steady-state progenitors (*Wang et al., 2014*), significantly reduced ATP levels in oligomycin-treated HSCs (*Figure 5—figure supplement 1H*, J). Inhibiting LKB1, a kinase upstream of AMPK (*Hardie, 2014*; *Long and Zierath, 2006*), did not affect the ATP concentration in oligomycin-treated HSCs (*Figure 5—figure supplement 1I, K*), whereas levels of adenosine monophosphate (AMP), which also activates AMPK, increased in oligomycin-treated but not in 5-FU-treated HSCs (*Figure 5—figure supplement 1L*). This may explain differences in AMPK-dependent ATP production between proliferative HSCs and HSCs under OXPHOS inhibition.

Next, we tested the effects of PFKFB3 on ATP concentration in HPCs. Unlike HSCs, HPCs exhibited PFKFB3-dependent ATP production, even without oligomycin (*Figure 5—figure supplement 1M–Q*). Therefore, ATP production in steady-state HSCs was PFKFB3-independent, and proliferative stimulation or OXPHOS inhibition plastically activated glycolytic ATP production in a PFKFB3-dependent manner to meet ATP demand.

## PFKFB3 activity renders HSCs dependent on glycolysis

Next, we investigated whether PFKFB3 activity itself confers glycolytic dependence on HSCs. We retrovirally overexpressed *Pfkfb3* in HSCs and performed cell cycle analysis (*Figure 5J*). *Pfkfb3*-overexpressed HSCs increased the proportion of cells in the S/G2/M phase and decreased the number of G$_0$ cells compared to *mock*-overexpressed HSCs (*Figure 5K*). Next, we retrovirally overexpressed *Pfkfb3* in GO-ATeam2$^+$ HSCs and performed real-time ATP measurement (*Figure 5J*). *Pfkfb3*-overexpressing GO-ATeam2$^+$ HSCs did not show changes in ATP concentrations relative to those in *mock*-transduced cells (*Figure 5L*; *Figure 5—figure supplement 1R*). Upon 2-DG treatment, *Pfkfb3*-overexpressing HSCs showed a greater decrease in ATP concentration than *mock*-transduced HSCs did (*Figure 5L*; *Figure 5—figure supplement 1S*). However, oligomycin treatment of both *mock*-transduced and *Pfkfb3*-overexpressing HSCs decreased ATP concentration to comparable levels (*Figure 5M*; *Figure 5—figure supplement 1T*). Notably, *Pfkfb3*-overexpressing HSCs recovered ATP levels more effectively under low glucose conditions (12.5 mg/dL) than did *mock*-transduced HSCs (*Figure 5M*; *Figure 5—figure supplement 1U*). These data suggest that PFKFB3 directly conferred glycolytic dependence onto HSCs by modulating the cell cycle and increasing their ATP-generating capacity via glycolysis under metabolic stress.

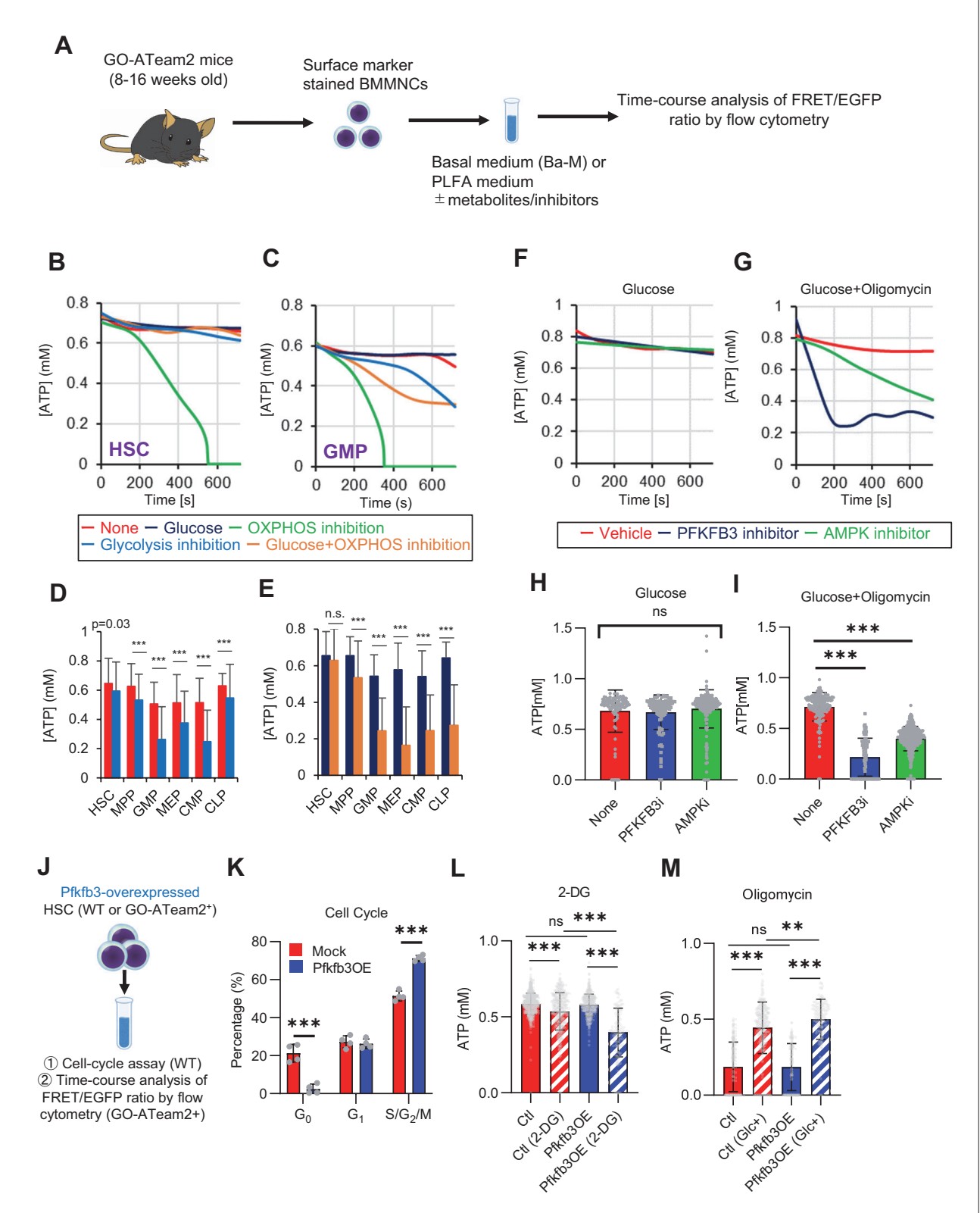

**Figure 5.** PFKFB3 accelerates glycolysis in HSCs under OXPHOS inhibition in an AMPK-dependent manner. (**A**) Experimental design of real-time ATP analysis using GO-ATeam2 knock-in BMMNCs. Ba-M was used in experiments with oligomycin. For other experiments, PLFA medium was used. (**B–C**) Evaluation of factors affecting ATP concentration in HSCs (**B**) and GMPs (**C**) based on the GO-ATeam2 system. GO-ATeam2 knock-in BMMNCs were incubated with glucose, oligomycin, 2-DG, or glucose plus oligomycin, and the FRET/EGFP ratio was calculated. (**D**) ATP concentration in indicated

*Figure 5 continued on next page*

*Figure 5 continued*

stem/progenitor fractions in PLFA medium (red bars) alone or PLFA medium plus 2-DG (blue bars). ATP concentration for the last 2 min of the analysis time is shown. Data is summarized from (**B, C**) and *Figure 5—figure supplement 1*. Each group represents at least 110 cells. Data are representative results of pooled samples from three biological replicates. (**E**) ATP concentration in indicated stem/progenitor fractions in Ba-M plus glucose (dark blue bars) or Ba-M plus glucose and oligomycin (orange bars). ATP concentration for the last 1 min of the analysis period is shown. Data is summarized from (**B, C**) and *Figure 5—figure supplement 1*. Each group represents at least 43 cells. Data are representative results of pooled samples from three biological replicates. (**F–I**) Effects of PFKFB3 or AMPK inhibitors (PFKFB3i or AMPKi, respectively) on ATP concentration in HSCs from GO-ATeam2 mice in Ba-M plus glucose only (**F**) or Ba-M plus glucose and oligomycin (**G**). ATP concentrations for the last 1 min of the analysis period are shown in (**H**) and (**I**) for glucose only and glucose with oligomycin groups, respectively. Each group represents at least 90 cells. Data are representative results of pooled samples from three biological replicates. (**J**) Experimental schema for cell cycle assay and real-time ATP concentration analysis after overexpression of *Pfkfb3*. (**K**) Cell cycle status of *Pfkfb3*-overexpressing (*Pfkfb3*OE) and *mock*-transduced HSCs. (**L–M**) Effects of inhibitors on ATP concentration in *Pfkfb3*-overexpressing GO-ATeam2+ HSCs. Cells were exposed to vehicle or 2-DG (**L**), oligomycin in the presence or absence of glucose 12.5 mg/dL (**M**), and ATP concentrations for the last 2 min (**L**) or 1 min (**M**) of the analysis period were calculated. Data are representative results of pooled samples from three biological replicates. Data are presented as mean ± SD. * p≤0.05, ** p≤0.01, *** p≤0.001 as determined by Student's *t*-test (**D, E, and K**) or one-way ANOVA followed by Tukey's test (**H, I, L, and M**). See also *Figure 5—figure supplement 1*.

The online version of this article includes the following source data and figure supplement(s) for figure 5:

**Source data 1.** Raw data for *Figure 5D, E, H, I, K, L and M*.

**Figure supplement 1.** Steady-state PFKFB3 activity defines HSC and HPC metabolic kinetics and cell cycle.

**Figure supplement 1—source data 1.** Raw data for *Figure 5—figure supplement 1F,J-L, Q*.

## PFKFB3 methylation by PRMT1 supports ATP production by cell-cycling HSCs

Next, we investigated how 5-FU-treated-HSCs regulate PFKFB3 independently of AMPK (*Figure 4G–J*). PFKFB3 activity is regulated at multiple levels (*Shi et al., 2017*), and PFKFB3 transcript and protein levels in HSCs remained unchanged during 5-FU-induced cell cycling (*Figure 6A–B*). Phosphorylation can also regulate PFKFB3 activity (*Marsin et al., 2002*; *Novellasdemunt et al., 2013*; *Okamura and Sakakibara, 1998*); however, we observed no change in PFKFB3 phosphorylation in 5-FU-treated-HSCs (*Figure 6C*). Upon oligomycin exposure, PFKFB3 was phosphorylated by AMPK in the HSCs (*Figure 6D*). PFKFB3 is also methylated, and its activity is upregulated by protein arginine methyltransferase 1 (PRMT1; *Yamamoto et al., 2014*). We observed that *Prmt1* expression increased in 5-FU-treated-HSCs relative to that in PBS-treated-HSCs (*Figure 6E*). Furthermore, PFKFB3 methylation was significantly induced in 5-FU-treated-HSCs than in PBS-treated-HSCs (*Figure 6F*). Treatment of HSCs with a PRMT1 inhibitor decreased PFKFB3 methylation (*Figure 6G*), suggesting that PRMT1 catalyzed PFKFB3 methylation. In contrast, the number of transcripts regulated by PRMT1 decreased or was unchanged (*Figure 6—figure supplement 1*), suggesting that the transcriptional regulatory function of PRMT1 is limited. To investigate whether glycolytic activity in HSCs was regulated by methylated-PFKFB3 (m-PFKFB3), mice treated with PBS or 5-FU were injected with 2-NBDG, and m-PFKFB3 levels in HSCs with high and low 2-NBDG uptake were quantified. Regardless of PBS or 5-FU treatment, HSCs with high 2-NBDG uptake exhibited higher m-PFKFB3 levels than those with low uptake (*Figure 6H*), suggesting that m-PFKFB3 regulated the activity of the glycolytic system in HSCs.

Further, we analyzed the potential effects of PRMT1 inhibition on ATP concentration in GO-ATeam2+ HSCs. Treatment with the PRMT1 inhibitor significantly decreased ATP levels in 5-FU-treated-HSCs than in PBS-treated-HSCs (*Figure 6I*). In contrast, the retroviral overexpression of *Pfkfb3* in GO-ATeam2+ HSCs abolished the effect of the PRMT1 inhibitor on ATP reduction (*Figure 6J*). These findings indicated that ATP levels in 5-FU-treated-HSCs were supported by PRMT1 methylation–mediated PFKFB3 activation.

## PFKFB3 contributes to HSPC pool expansion and stress hematopoiesis maintenance

Finally, we analyzed PFKFB3 function in HSCs during hematopoiesis. We cultured HSCs with a PFKFB3 inhibitor in vitro under quiescence-maintaining or proliferative conditions (*Figure 7—figure supplement 1A*; *Kobayashi et al., 2019*). Cell count in HSC-derived colonies decreased following treatment with a PFKFB3 inhibitor under proliferative, but not quiescence-maintaining, conditions (*Figure 7—figure supplement 1B*).

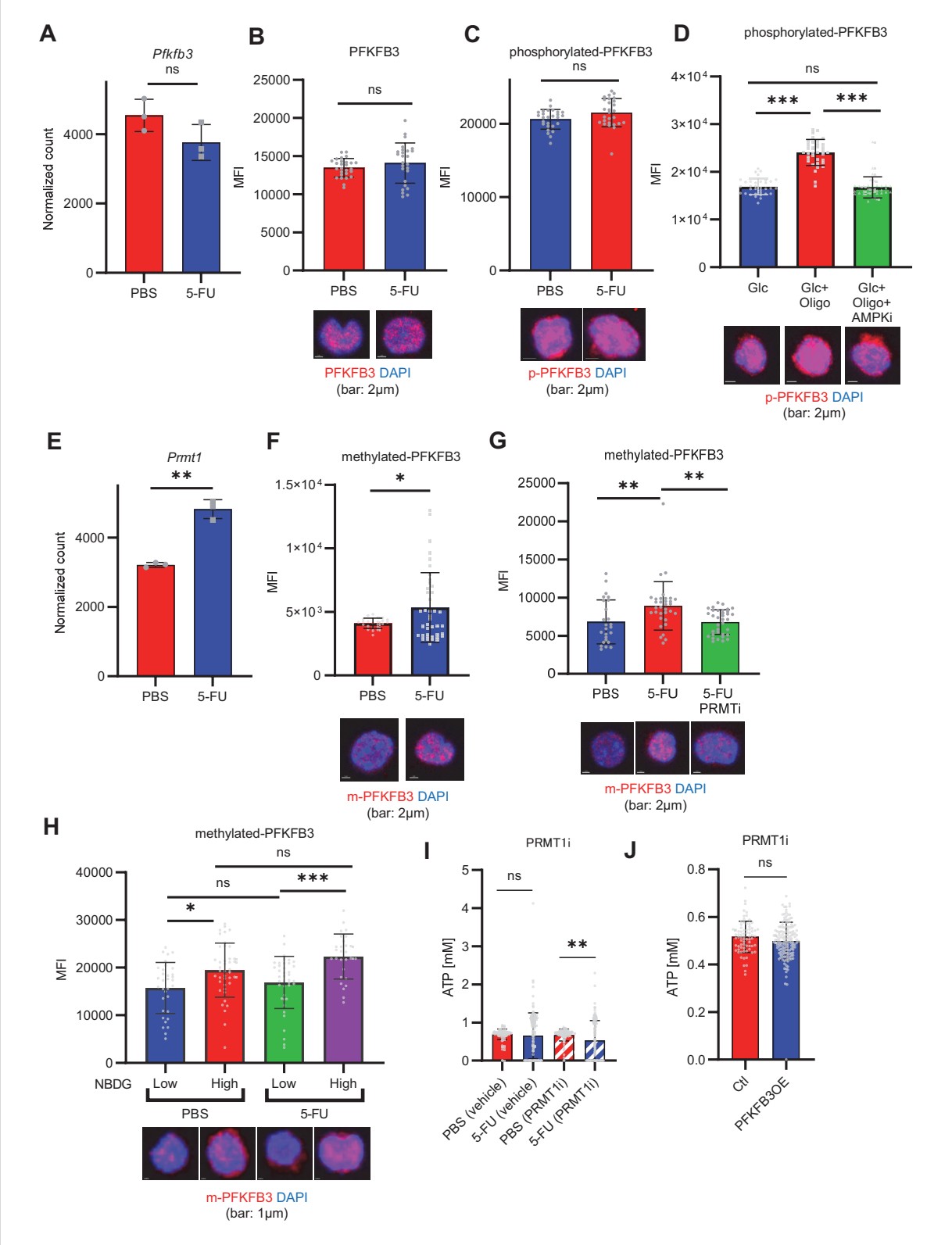

**Figure 6.** PFKFB3 methylation by PRMT1 enables ATP production by cell-cycling HSCs. (**A**) Normalized *Pfkfb3* mRNA counts based on RNA sequencing of PBS-treated (red) or 5-FU-treated (blue) HSCs. Data are representative results of pooled samples from three biological replicates. Data were extracted from the same pooled samples as in *Figure 4J* and *Figure 6—figure supplement 1*. (**B**) Quantification of mean fluorescence intensity (MFI) of PFKFB3 protein in PBS- or 5-FU-treated HSCs. The lower part of the graph shows representative images of immunocytochemistry of PFKFB3

*Figure 6 continued on next page*

*Figure 6 continued*

in each group. n=26–27 single HSCs for each group. The data are representative results from two independent experiments. (**C**) Quantification of MFI of phosphorylated-PFKFB3 (p-PFKFB3) protein in PBS- or 5-FU-treated HSCs. The lower part of the graph shows representative images of immunocytochemistry of p-PFKFB3 in each group. n=27 single HSCs for each group. The data are representative results from two independent experiments. (**D**) Quantification of MFI of p-PFKFB3 in HSCs treated with glucose (200 mg/dL); glucose plus oligomycin (1 μM); and glucose, oligomycin, and dorsomorphin (100 μM) for 5 min. The lower part of the graph shows representative images of immunocytochemistry of p-PFKFB3 in each group. n=32–36 for each group. The data are representative results from two independent experiments. (**E**) Normalized *Prmt1* mRNA counts based on RNA sequencing of PBS-treated (red) or 5-FU-treated (blue) HSCs. Data are representative results of pooled samples from three biological replicates. (**F**) MFI quantification of methylated-PFKFB3 (m-PFKFB3) in PBS- or 5-FU-treated HSCs. The lower part of the graph shows representative images of immunocytochemistry of m-PFKFB3 in each group. n=23–41 for each group. The data are representative results from three independent experiments. (**G**) Quantification of MFI of m-PFKFB3 in PBS- or 5-FU-treated HSCs or 5-FU-treated HSCs after 15 min treatment with a PRMT1 inhibitor (90 μg/mL GSK3368715); n=25–35 single HSCs for each group. The lower part of the graph shows representative images showing immunocytochemistry of m-PFKFB3. Data represent a single experiment. (**H**) Quantitation of m-PFKFB3 in NBDG-positive or -negative HSCs in mice treated with PBS or 5-FU. The lower part of the graph shows representative images of immunocytochemistry of m-PFKFB3 in each group. n=28–41 for each group. The data are representative results from two independent experiments. (**I**) Corrected ATP levels in PBS- (red) or 5-FU-treated (blue) HSCs 15 min after treatment with vehicle or a PRMT1 inhibitor (90 μg/mL GSK3368715). Each group represents at least 101 cells. Data are representative results of pooled samples of two biological replicates. (see 'Time-course analysis of FRET values' in 'Materials and methods' for details of the correction method used to calculate ATP concentration.) (**J**) ATP concentration in mock-transduced (Ctl) or *Pfkfb3*-overexpressed (OE) HSCs after treatment with the PRMT1 inhibitor (90 μg/mL GSK3368715). ATP concentration for the last 1 min of the analysis period is shown. Data are presented as mean ± SD. * p≤0.05, ** p≤0.01, *** p≤0.001 as determined by Student's *t*-test (**A-C, E-F, and I-J**) or one-way ANOVA followed by Tukey's test (**D, G, and H**). See also *Figure 6—figure supplement 1*.

The online version of this article includes the following source data and figure supplement(s) for figure 6:

**Source data 1.** Raw data for *Figure 6A–J*.

**Figure supplement 1.** Gene expression related to Prmt1 in proliferating HSCs.

**Figure supplement 1—source data 1.** Raw data for *Figure 6—figure supplement 1*.

We also knocked out *Pfkfb3* in HSCs using the less toxic, vector-free CRISPR-Cas9 system and cultured the cells under quiescence-maintaining or proliferative conditions (*Figure 7—figure supplement 1A*) based on recent reports by *Shiroshita et al., 2022*. Again, cell numbers in *Pfkfb3*-knockout (KO) HSC–derived colonies decreased only in proliferative cultures when compared to control cultures (*Rosa26*-KO HSCs) (*Figure 7—figure supplement 1C*, E, F). We retrovirally overexpressed *Pfkfb3* in HSCs and cultured them under quiescence maintenance or proliferative conditions (*Figure 7—figure supplement 1A*). *Pfkfb3*-overexpressing HSC colonies showed increased cell count compared to that of *mock*-transduced cells, but only under proliferative conditions (*Figure 7—figure supplement 1D*).

To assess PFKFB3 function in HSCs in vivo, we transplanted *Pfkfb3*-KO HSCs (Ly5.2[+]) or wild type (WT) control HSCs into lethally irradiated recipients (Ly5.1[+]) as well as Ly5.1[+] competitor cells (*Figure 7A*), and the behavior of *Pfkfb3*-KO cells was evaluated by Sanger sequencing of peripheral blood (PB) cells (*Shiroshita et al., 2022*). In the KO group, donor-derived chimerism in PB cells decreased relative to that in the WT control group during the early phase (1 month post-transplant) but recovered thereafter (*Figure 7B*). Next, we retrovirally transduced Ly5.2[+] HSCs with *Pfkfb3* S461E (*Pfkfb3*CA), a constitutively active PFKFB3 mutant, and transplanted them into lethally irradiated recipients (Ly5.2[+]), along with Ly5.1[+] competitor cells (*Figure 7A*, *Figure 7—figure supplement 1G*). Donor chimerism during the early post-transplant period in the *Pfkfb3*CA-overexpressing group was significantly higher than that in the *mock*-transduced group (*Figure 7C*). These findings suggest that PFKFB3 may play a role in the differentiation and proliferation of HSCs. Therefore, we compared the contribution of PFKFB3 to HSPC function at steady state and after myeloproliferative stimulation. *Pfkfb3*- or *Rosa26*-KO HSPCs were transplanted into recipients (Ly5.1[+]). After 2 months, recipients received 5-FU intraperitoneally, and the dynamics of *Pfkfb3*- or *Rosa26*-KO cell abundance in PB was assessed (*Figure 7D*). In PB cells prior to 5-FU administration, *Pfkfb3*- or *Rosa26*-KO HSPC-derived blood cells were almost equally present, suggesting a limited involvement of PFKFB3 in steady-state blood cell production (*Figure 7E*). However, after 5-FU administration, *Pfkfb3*-KO HSPC-derived blood cell abundance was reduced compared to that in the *Rosa26*-KO group (*Figure 7E*). This change occurred on day 6 after 5-FU administration (day 1), when the cell cycle of HSCs was activated (*Figure 1—figure supplement 1D*), supporting the idea that PFKFB3 contributes to HSC proliferation and differentiation into HSPCs.

To investigate the mechanisms underlying the short-term effects of PFKFB3 on hematopoiesis after bone marrow transplantation (BMT), we evaluated cell cycle and apoptosis of *Pfkfb3*-KO or

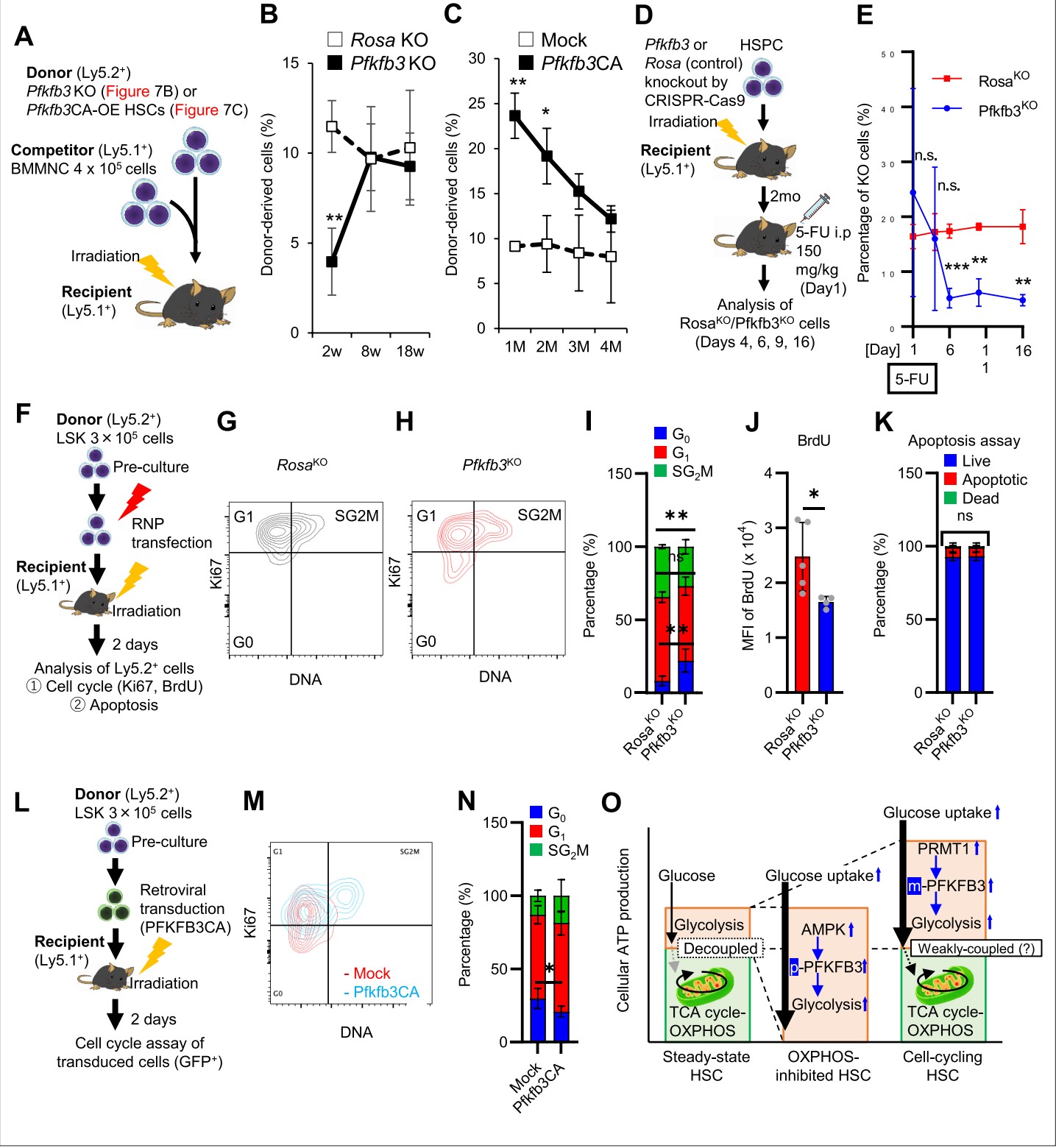

**Figure 7.** PFKFB3 maintains HSC function under proliferative stress. (**A–C**) Transplant analysis of *Pfkfb3*-KO or *Pfkfb3*CA-overexpressing HSCs. Experimental design (**A**). PB chimerism of donor-derived cells at 4 months post-transplant. *Pfkfb3*-KO group, n=6; *Rosa26*-KO group, n=4; (**B**) *Pfkfb3* group, n=5; pMY-IRES-GFP group, n=4. (**C**) The data are representative results from two independent experiments. (**D–E**) 5-FU administration after bone marrow reconstruction with *Pfkfb3*- or *Rosa26*-KO HSPCs. Experimental schema (**D**). Behavior of the *Pfkfb3*- or *Rosa26*-KO cells in PB after 5-FU administration (**E**). n=5 for each group. (**F–K**) Cell cycle analysis and apoptosis assay of *Pfkfb3*- or *Rosa26*-KO HSPCs on day 2 post-BMT. Experimental

*Figure 7 continued on next page*

*Figure 7 continued*

schema (**F**). Representative plots of Ki67/Hoechst33432 staining of *Rosa26*-KO (**G**) or *Pfkfb3*-KO (**H**) HSPCs and summary of analysis (**I**); summary of in vivo BrdU labeling assay (**J**). Apoptosis assay results (**K**). n=4–5 biological replicates for each group. (**L–N**) Cell cycle analysis of *Pfkfb3*CA or *Mock*-overexpressing HSPCs on day 2 after BMT. Experimental Schema (**L**). Representative plot of Ki67/Hoechst33432 staining for both groups (**M**) and summary of analysis (**N**). n=5 biological replicates for each group. (**O**) Models showing ATP production and regulation in quiescent, OXPHOS-inhibited, and cell-cycling HSCs. Note that the GO-ATeam2 system identified plastic acceleration of glycolysis by PFKFB3 in response to different types of stress maintains ATP levels. Data are presented as mean ± SD. * p≤0.05, ** p≤0.01, *** p≤0.001 as determined by Student's *t*-test (**B, C, E, I-K, and N**). See also *Figure 7—figure supplement 1*.

The online version of this article includes the following source data and figure supplement(s) for figure 7:

**Source data 1.** Raw data for *Figure 7B, C, E, I–K and N*.

**Figure supplement 1.** PFKFB3 contributes to HSC proliferation and differentiation in vitro.

**Figure supplement 1—source data 1.** Raw data for *Figure 7—figure supplement 1B–D, G, H*.

-overexpressing HSPCs on day 2 after BMT (*Figure 7F*). Cell cycle was analyzed by Ki67/Hoechst33432 staining and in vivo BrdU labeling (*Jun et al., 2021*), which showed that cell cycle progression was suppressed in *Pfkfb3*-KO HSPCs (*Figure 7G–J*). In contrast, *Pfkfb3*-KO cells did not show increased apoptotic rates or decreased homing efficiency after BMT (*Figure 7K*; *Figure 7—figure supplement 1H*). Furthermore, we examined the cell cycle of HSPCs overexpressing *Pfkfb3*CA on day 2 after BMT (*Figure 7L*) and found that *Pfkfb3*CA-overexpressing HSPCs showed accelerated cell cycle compared to *mock*-overexpressing HSPCs (*Figure 7M–N*). These data suggest that PFKFB3 contributes to HSC proliferation and differentiates cell production in in vitro and in vivo proliferative environments (cytokine stimulation and transplantation).

## Discussion

In this study, by combining metabolomic tracing of U-$^{13}$C$_6$-labeled glucose and $^{13}$C-MFA, we quantitatively identified the metabolic programs used by HSCs during steady-state, cell-cycling, and OXPHOS inhibition. Under proliferative stress, HSCs uniformly shift from mitochondrial respiration to glycolytic ATP production and PPP activation, which represent hallmarks of cell-cycling mammalian cells (*Intlekofer and Finley, 2019*). Previous reports have emphasized the importance of glycolysis in maintaining HSC quiescence, but have primarily analyzed HSCs in transplant assays, wherein HSCs must enter the cell cycle (*Takubo et al., 2013*; *Takubo et al., 2010*). Prior analysis of repopulation capacity, which is positively correlated with enhanced glycolysis, may have overestimated glycolytic ATP production and overlooked mitochondrial ATP production during native hematopoiesis. In fact, some studies have suggested that OXPHOS activity is important for HSC maintenance and function (*Ansó et al., 2017*).

Our method was based on recently reported quantitative metabolic analysis techniques for very small numbers of cells (*Qi et al., 2021*; *Agathocleous et al., 2017*; *DeVilbiss et al., 2021*; *Lengefeld et al., 2021*; *Schönberger et al., 2022*; *Jun et al., 2021*), such as HSCs, and expands our knowledge of HSC metabolism during stress hematopoiesis. In our study, 5-FU administration in mice transiently decreased ATP concentration in HSCs in parallel with cell cycle progression, suggesting that HSC differentiation and cell cycle progression are closely related to intracellular metabolism and can be monitored by measuring ATP concentration. We mainly analyzed a mixture of EPCR$^+$ and EPCR$^-$ HSCs, and we believe that the observed cell cycle progression and promotion of glycolysis in both EPCR$^+$ and EPCR$^-$ HSCs support the validity of our claims (*Figure 1L*, *Figure 1—figure supplement 1G–K*). According to $^{13}$C-MFA enzymatic reaction flux of PFK in 5-FU-treated HSCs indicated a relative increase of approximately 10%. However, the flux value obtained by $^{13}$C-MFA was calculated with glucose uptake as 100. Thus, when combined with the overall increase in the glycolytic pool demonstrated by in vitro isotopic glucose tracer analysis and in vivo NBDG analysis, rapid acceleration of glycolysis becomes evident throughout the HSCs, including subpopulations that were less responsive to stress (*Bowling et al., 2020*; *Fanti et al., 2023*; *Munz et al., 2023*). These findings are consistent with reports suggesting that HSCs have relatively low biosynthetic activity (*Signer et al., 2014*; *Essers et al., 2009*) that is rapidly activated in response to cell proliferation stimuli (*Karigane et al., 2016*; *Umemoto et al., 2018*). Notably, we found that HSCs could accelerate glycolytic ATP production to fully compensate for mitochondrial ATP production under OXPHOS inhibition, a phenomenon that is difficult to identify without real-time ATP analysis. Thus, HSCs exposed to acute stresses choose to

change the efficiency of glucose utilization (accelerated glycolytic ATP production) rather than other energy sources. In vivo, a completely glucose-deficient environment is improbable. Therefore, even under conditions such as hypoxia, where OXPHOS is inhibited, it is conceivable that glycolysis is accelerated to maintain ATP concentrations. Glucose tracer analysis showed NAS suppression under OXPHOS inhibition, leading to glycolysis without cell proliferation (*Figure 2C–F*; *Figure 1—figure supplement 1N*). This suppression can be attributed to several factors: phosphates derived from ATP are added to nucleotide mono-/di-phosphates during NAS; the primary source of ATP production, OXPHOS, is impaired; and the presence of enzymes, such as dihydroorotate dehydrogenase, which are conjugated with OXPHOS (*Liu et al., 2000*). Such multifactorial effects raise new questions about the relationship between OXPHOS and nucleotide synthesis. On the other hand, we observed that ATP production in steady-state or cell-cycling HSCs and in naturally proliferating HPCs depended more on mitochondrial OXPHOS than on glycolysis; inhibiting glycolysis in steady-state HSCs resulted in only mild ATP decreases, suggesting that OXPHOS is still the major source of ATP production even in a medium saturated with hypoxia mimicking the BM environment. The p50 value of mitochondria (the partial pressure of oxygen at which respiration is half maximal) is less than 0.1 kPa, corresponding to an oxygen concentration of less than 0.1% under atmospheric pressure (*Gnaiger et al., 2000*), suggesting that even under hypoxic conditions, OXPHOS can maintain some level of activity. Because FAO and the mitochondrial respiratory chain are necessary for HSC self-renewal and quiescence (*Ansó et al., 2017*; *Bejarano-García et al., 2016*; *Ito et al., 2012*; *Kobayashi et al., 2019*), fatty acids may support mitochondrial ATP production independently of fluxes from glycolysis. FAO and glutaminolysis were not immediately essential for ATP production in HSCs. Given reports on the long-term necessity of FAO and glutaminolysis for HSC maintenance (*Ito et al., 2012*; *Oburoglu et al., 2014*), ATP concentrations could be maintained in the short term by compensatory pathways. Furthermore, although glycolysis and TCA cycle are decoupled in steady-state HSCs, in response to cell cycle progression, anaerobic glycolytic metabolism in HSCs is enhanced (*Figure 1*) and fluxes to TCA cycle and PPP from the glycolytic system are also promoted (*Figure 3—figure supplement 2*). The degree of glycolysis and TCA cycle coupling observed by in vitro and in vivo tracer analysis differed, likely due to differences in labeling time (10–30 min in vitro and 3 hr in vivo). In particular, in vivo tracer analysis allows all cells to be capable of metabolizing U-$^{13}$C$_6$-glucose and providing its metabolites to HSCs, and there is a significant amount of time, approximately 120–180 min, after glucose labeling to purify HSCs. Metabolic reactions will continue during this time and subsequent processing on ice, which may increase the influx of labeled carbon into the TCA cycle. This complex dynamic in the in vivo tracer analysis makes it difficult to determine whether the labeled carbon influx is the result of direct influx from glycolysis or the re-uptake of metabolites by HSCs that have been processed by other cells. This is in contrast to in vitro analysis where such extended metabolic processing does not occur. Furthermore, despite an increased carbon influx into the TCA cycle in vivo, ATP production from mitochondria does not show a corresponding increase after 5-FU treatment, as shown by the GO-ATeam2 analysis shown in *Figure 4C*. Despite these technical differences, an essential common finding from both in vivo and in vitro analyses is the activation of glycolysis and nucleotide synthesis (NAS) in 5-FU-treated HSCs, highlighting critical metabolic changes in response to treatment. Moreover, these data provide direct evidence that glycolysis and TCA cycle become functionally uncoupled in quiescent HSCs (*Takubo et al., 2013*; *Halvarsson et al., 2017*). Our findings are also consistent with previous reports of OXPHOS activation associated with HSC proliferation (*Takubo et al., 2013*; *Yu et al., 2013*; *Maryanovich et al., 2015*; *Ito et al., 2012*). In other words, HSCs exhibit an increased proportion of anaerobic glycolysis–derived ATP by PFKFB3 upon proliferation and OXPHOS inhibition; furthermore, the glycolytic system is the starting point of metabolic activation and is indispensable for the overall enhancement of HSC metabolism (*Figure 7H*).

HPCs and leukemic cells accelerate glycolytic ATP production using PKM2 for differentiation and transformation, respectively *Wang et al., 2014*; however, we demonstrated that glycolytic acceleration does not fully compensate for mitochondrial ATP production in HPCs. Mechanistically, PFKFB3 increased glycolytic activity in HSCs to maintain ATP concentrations during proliferation and OXPHOS inhibition. Furthermore, inhibition of PFKFB3 in addition to OXPHOS does not result in a complete loss of ATP in HSCs, suggesting the robustness of HSC metabolism (*Figure 5G*). Under steady-state conditions, naturally proliferating HPCs rely on PFKFB3 for ATP production, whereas HSCs do not. This may explain the reduction of ECAR after oligomycin treatment in MyPs

as shown by the Mito stress test (*Figure 2G*). In other words, while PFKFB3-dependent active glycolysis and mitochondria must always be coupled in MyPs, this is not necessarily the case in HSCs, even after 5-FU treatment (*Figure 1G*). Therefore, we can infer that quiescent HSCs at steady state can produce ATP via PFKFB3 activation in response to stress, enabling additional ATP generation. Furthermore, overexpression of *Pfkfb3* in HSCs increased glycolytic dependency, suggesting that PFKFB3 itself can modulate metabolic dependency in HSCs. Changes in glycolytic dependency in HSCs overexpressing *Pfkfb3* may seem small (0.06–0.13 mM; *Figure 5L and M*). However, it is noteworthy that the rate of the reaction catalyzed by PFK varies greatly within a very narrow range of ATP concentrations, less than 1 mM. Webb et al. analyzed the factors controlling PFK activity and reported that the reaction rate of PFK varies by approximately 40% in the 0.3–1 mM ATP concentration range (*Webb et al., 2015*). The reason that differences in glycolytic dependence could be detected in cells overexpressing *Pfkfb3* may be that the ATP concentration at the time of analysis was approximately 0.5–0.6 mM, which is within the range where a small change in ATP concentration can dynamically alter PFK activity.

PFKFB3 supports hematopoiesis in contexts that require robust HSPC proliferation in vitro and in vivo. We showed that the positive or negative effect of *Pfkfb3* overexpression or KO on differentiated blood cell production is gradually lost after BMT. This is because HSPCs require PFKFB3 for cell cycle progression during stress hematopoiesis in the early phase after BMT (*Figure 7F–J and L–N*). However, even during stress hematopoiesis, PFKFB3 is not involved in cell death or homing efficiency (*Figure 7K*; *Figure 7—figure supplement 1H*) and appears to contribute primarily to the regulation of transient HSPC proliferation in the BM cavity. HSCs no longer require PFKFB3 for a certain period of time after BMT, probably because they regain a quiescent state. This is consistent with the fact that inhibition of PFKFB3 in quiescent HSCs does not reduce the ATP concentration (*Figure 5F and H*), suggesting that the activity of PFKFB3 is plastically modified. HSC metabolic plasticity is also illustrated by the mode of PFKFB3 activation, differing depending on stress type. During proliferative stress, PRMT1 methylates PFKFB3 in the HSCs to promote glycolytic ATP production, a modification that increases its activity (*Yamamoto et al., 2014*). PRMT1 is required for stress hematopoiesis (*Zhu et al., 2019*), but its downstream targets in HSCs remain unclear. Our results strongly suggest that PRMT1 targets PFKFB3 to stimulate glycolysis in HSCs. In contrast, under OXPHOS inhibition, PFKFB3 phosphorylation by AMPK is induced—another modification that also upregulates its activity. These two PFKFB3 protein modifications allow for flexible regulation of ATP production by glycolysis, even under simultaneous and different stresses. In fact, the constitutively active S461E PFKFB3 mutant, designed to mimic phosphorylation in response to OXPHOS inhibition, enhanced HSC reconstitution capacity after transplantation, suggesting that even if PFKFB3 is activated by one stress (in this case, proliferative), it has the activation capacity to respond to a different stress (i.e. mitochondrial). Therefore, the functions of phosphorylated and methylated forms of PFKFB3 are to some extent interchangeable, and either modification can be used to handle diverse stresses.

In summary, we found that HSCs exhibit a highly dynamic range of glycolytic flux. Our study highlights glycolysis as a pivotal source of energy production in stressed HSCs, and indicates that OXPHOS, although an important source of ATP, can be uncoupled from glycolysis in steady-state HSCs without compromising ATP levels. Because multiple PFKFB3 modifications safeguard HSCs against different stresses by accelerating glycolysis, interventions targeting these might effectively induce or manage stress hematopoiesis. This study provides a platform for comprehensive and quantitative real-time analysis of ATP concentration and its dynamics in HSPCs. Our approach allows for analysis of metabolic programs in rare cells and detection of various metabolic activities within a diverse cell population, making it applicable to the analysis of various tissue systems in normal and diseased states.

## Limitations of the study

In this study, 5-FU-treated HSCs were analyzed as cell-cycling HSCs, but if more sensitive and time-saving glucose tracer analysis methods (especially after in vivo labeling with isotopic glucose) are developed, it may be possible to prospectively differentiate and quantitatively analyze HSC metabolism based on the cell surface antigens and cell cycle status. Although our assay uses media that mimic the BM environment, in the near future, in vivo GO-ATeam2 analysis will allow us to measure ATP concentrations in physiologically hypoxic BM.

# Materials and methods

## Mice and genotyping

C57BL/6 mice (7–16 weeks old, Ly5.2$^+$) were purchased from Japan SLC (Shizuoka, Japan). C57BL/6 mice (Ly5.1$^+$) were purchased from CLEA Japan (Shizuoka, Japan). Knock-in mice harboring GO-ATeam2 (*Imamura et al., 2009*; *Nakano et al., 2011*; *Yamamoto et al., 2019*) in the *Rosa26* locus were generated in the Yamamoto laboratory. The GO-ATeam2 mice (8–16 weeks old) were used to analyze HSPCs. Ubc-GFP reporter mice (Ubc-GFP mice) were from the Jackson Laboratory and genotyped using PCR-based assays. GO-ATeam2 mice were genotyped by PCR of tail DNA or by transdermal GFP fluorescence. The PCR protocol was as follows: 94 °C for 5 min; 34 cycles of 94 °C for 30 s, 56 °C for 30 s, 72 °C for 30 s; 72 °C for 5 min; and 4 °C hold. Primers for GO-ATeam2 or Ubc-GFP mice are listed in *Supplementary file 6*. mVenus-p27K$^-$ mice (17–20 weeks old) were provided by Kitamura Laboratory and used for cell cycle analysis (*Fukushima et al., 2019*). Mice were genotyped using PCR-based assays of tail DNA or transdermal Venus fluorescence. All mice were maintained in the animal facility at the National Center for Global Health and Medicine Research Institute under specific pathogen-free conditions and fed ad libitum. Mice were euthanized by cervical dislocation. All animal experiments were approved by the Institutional Animal Care and Use Committee (IACUC) at the National Center for Global Health and Medicine Research Institute. Both male and female mice were used.

## Cell preparation

For C57BL/6 mice, bone marrow (BM) cells were isolated from bilateral femurs and tibiae by flushing with PBS + 2% fetal calf serum (FCS) (Gibco) using a 21-gauge needle (Terumo Corporation, Tokyo, Japan) and a 10 mL syringe (Terumo). As an exception, for U-$^{13}$C$_6$-labeled glucose tracer experiments using C57BL/6 mice, BM was flushed with PBS +0.1% bovine serum albumin (BSA, Cat# A4503). The BM plug was dispersed by refluxing through the needle, and the suspension was centrifuged 680 × *g* for 5 min at 4 °C. Cells were lysed with lysis buffer (0.17 M NH$_4$Cl, 1 mM EDTA, 10 mM NaHCO$_3$) at room temperature (RT) for 5 min, washed with two volumes PBS + 2% FCS (or PBS +0.1% BSA for tracer experiments), and centrifuged at 680 × *g* for 5 min at 4 °C. Cells were resuspended in PBS +2% FCS (or PBS +0.1% BSA for tracer experiments) and filtered through 40 μm nylon mesh (BD Biosciences). Cells were again centrifuged 680 × *g* for 5 min at 4 °C and treated with anti-CD16/32 antibody for Fc-receptor block (2 μL/mouse; BD Biosciences, Cat# 553152) for 10 min at 4 °C. Anti-c-Kit magnetic beads (Miltenyi Biotec, Bergisch Gladbach, Germany, Cat# 130-091-224) were added at a 1:5 v/v ratio for 15 min at 4 °C. After removing the antibody with two PBS +2% FCS (or PBS +0.1% BSA for tracer experiments) washes, c-Kit-positive cells were isolated using Auto-MACS Pro (Miltenyi Biotec) with the Possel-s or Possel-d2 program. Isolated cells were centrifuged once at 340 × *g* for 5 min and stained with an antibody cocktail for flow cytometry.

For analysis of the GO-ATeam2 hematopoietic cells, BM from GO-ATeam2 mice was flushed with PBS + 0.1% BSA to minimize exposure to nutrients in FCS. Hemolysis, centrifugation, filtering, and Fc receptor blocking were performed in the same manner as for cell preparation using C57BL/6 mice. Cells were stained for 30 min with an antibody cocktail at 4 °C and then washed and suspended in 1000 μL PBS +0.1% BSA and centrifuged at 340 × *g* at 4 °C for 5 min. Supernatants were discarded in preparation for flow cytometry.

## Flow cytometry and cell sorting

Murine hematopoietic stem and progenitor fractions were labeled as follows: To stain cells from C57BL/6 mice, lineage (Lin) markers (CD4, CD8a, Gr-1, Mac-1, Ter-119, B220)-PerCP-Cy5.5 (BD Biosciences for CD4 (Cat# 550954), Gr-1 (Cat# 552093), Mac-1 (Cat# 550993), B220 (Cat# 552771) and BioLegend for CD8a (Cat# 100734) and Ter-119 (Cat# 116228) antibodies), c-Kit-APC-Cy7 (BioLegend, Cat# 105826), Sca-1-PE-Cy7 (BioLegend, Cat# 122514), CD150-PE (BioLegend, Cat# 115904), CD48-FITC (BioLegend, Cat# 103404), and Flt3-APC (BioLegend, Cat# 135310) were used. For HSC collection five days after 5-FU administration (intraperitoneally or intravenously), Mac-1 antibody was excluded from the antibody cocktail, and the LSK gate was expanded to include c-Kit-high to -dim Lin$^-$ cells to include functional HSCs early after 5-FU administration as previously reported (*Arai et al., 2004*; *Umemoto et al., 2022*). We did not expand the LSK gate at any time other than five days after 5-FU administration. When sorting or analyzing EPCR$^+$CD150$^+$CD48$^-$LSK cells

from C57BL/6 mice or mVenus-p27K-mice, CD150-BV421 (BioLegend, Cat# 115926), CD48-APC (BioLegend, Cat# 103412), and EPCR-PE (Biolegend, Cat# 141503) were used in addition to LSK for staining, and FLT3 staining was excluded. To stain cells from GO-ATeam2 mice or C57BL/6 mice for the 2-NBDG assay or homing assay using Ubc-GFP mice, lineage markers (CD4, CD8a, Gr-1, Mac-1, Ter-119, B220)-PerCP-Cy5.5, c-Kit-APC-Cy7, Sca-1-PE-Cy7, CD150-BV421, and CD48-APC were used. In the analysis using GO-ATeam2 mice, Flt3 was not used to define HSCs because the fluorescence of the FRET sensor (EGFP, mKO) limits the available fluorochromes for surface marker staining. In analysis using the AMPK inhibitor dorsomorphin (Cayman Chemical, Cat# 21207), CD150-APC (BioLegend, Cat# 115910) and CD48-Alexa Fluor700 (BioLegend, Cat# 103426) were used to stain LSK-SLAM to eliminate effects of dorsomorphin fluorescence on cell staining. Cells were resuspended in 0.5–2 mL of PBS +2% FCS+0.1% propidium iodide (PI) (Invitrogen, Cat# P3566) (for C57BL/6 mice) or PBS +0.1% BSA (for GO-ATeam2 mice) and sorted using the FACSAria IIIu Cell Sorter (BD Biosciences) into RPMI1640 (without glucose) (Nacalai Tesque, Cat# 09892–15) containing 4% w/v BSA or GO-ATeam2 basal medium (Ba-M, *Supplementary file 1* with 4% w/v BSA) (custom made by Gmep Inc). Murine HSCs were defined as $CD150^+CD48^-Flt3^-LSK$ (for C57BL/6 mice) or $CD150^+CD48^-LSK$ (for GO-ATeam2 mice and mVenus-p27K- mice, and when EPCR was included in the antibody cocktail against C57BL/6 mice) cells. MPPs were defined as $CD150^-CD48^+Flt3^-$ LSK (for C57BL/6 mice) or $CD150^-CD48^+LSK$ (for GO-ATeam2 mice) cells. Among myeloid progenitors (MyPs), GMPs/MEPs/CMPs were defined as follows: GMPs ($CD16/32^+$ $CD34^+$), MEPs (CD16/32- $CD34^-$), and CMPs (CD16/32- $CD34^+$). CLPs were defined as $Lin^-Sca-1^{low}c-Kit^{low}Flt3^+IL7R\alpha^+$ cells. Data were analyzed using FlowJo V10 (Tree Star) software.

## Intracellular staining for phosphorylated Rb (pRb)

$EPCR^+$ or $EPCR^-$ LSK-SLAM cells from PBS- or 5-FU-treated C57BL/6 mice were purified separately (see "Flow cytometry and cell sorting" for details). Anti-phospho-Rb (Ser807/811) antibody (CST, Cat# 8516T) was used as the primary antibody and Anti-rabbit IgG (H+L), F(ab') Fragment (Alexa Fluor488 Conjugate) (CST, Cat# 4412) was used as the secondary antibody. Fixation and permeabilization were performed according to the manufacturer protocol. pRb and DNA content (stained with PI) were analyzed by flow cytometry.

## Analysis of mVenus-p27K-mouse-derived BM cells

Surface-marker-stained BM mononuclear cells (MNCs) (see 'Flow cytometry and cell sorting' for details) were analyzed by flow cytometry to determine the frequency of G0 marker positivity for $EPCR^+$ or $EPCR^-$ $CD150^+CD48^-LSK$ or progenitor cells.

## Seahorse flux analyzer

The extracellular acidification rate (ECAR) and oxygen consumption rate (OCR) were measured using a Seahorse XFe96 extracellular flux analyzer according to the manufacturer's instructions (Agilent Technologies). Briefly, sorted cells were dispensed to culture plates pre-coated with Cell-Tak (Corning) and then the media was replaced with pre-warmed XF-DMEM medium (Agilent) supplemented with 10 mM glucose, 1 mM pyruvate, and 2 mM glutamine, followed by centrifugation at 200 × *g* for 5 min. OCR and ECAR were measured at baseline and again after sequential addition of respiratory inhibitors at final concentrations of 1 μM oligomycin (an inhibitor of ATP synthase), 2 μM FCCP (an uncoupling agent of mitochondrial respiration), 0.5 μM rotenone/antimycin (an inhibitor of mitochondrial complex I/III) and 50 mM 2-deoxy-D-glucose (an inhibitor of glycolysis). The experiment was performed by dispensing 75,000 HSCs (PBS or 5-FU treated) or MyPs per well.

## 2-NBDG assay

For the in vitro 2-NBDG assay, sorted HSCs were exposed to 200 μM 2-NBDG (Cayman Chemical, Cat# 11046) for 30 min. HSCs were then centrifuged at 340 × *g* at 4 °C for 5 min, and the supernatant was removed. The uptake of 2-NBDG was measured using FACS Aria IIIu. As a negative control, HSCs were simultaneously exposed to 54 μg/ml phloretin or 20 μg/ml cytochalasin B with 2-NBDG.

For the in vivo 2-NBDG assay, C57BL/6 mice treated with PBS or 5-FU were subjected to an in vivo 2-NBDG assay as reported by *Jun et al., 2021*. Mice received a bolus dose of 375 μg 2-NBDG intravenously and were euthanized by cervical dislocation after 1 hr. Mice were immediately placed on ice,

and all subsequent cell preparation processes were performed while the cells were chilled on ice. The 2-NBDG positive cell fraction was detected by flow cytometry.

## Conversion of GO-ATeam2 fluorescence to ATP concentration

The GO-ATeam2 knock-in mice were reported by *Yamamoto et al., 2019*. Briefly, we used a CAG promoter-based knock-in strategy targeting the *Rosa26* locus to generate GO-ATeam2 knock-in mice. A study presenting the significance of measuring the absolute concentration of ATP at the single-cell level is currently in preparation for submission, but briefly, the FRET efficiency was converted to the absolute concentration of ATP using the following method (Watanuki et al., *in preparation*). To permeabilize BM cells, α-hemolysin stock solution (Sigma-Aldrich, St. Louis, MO, USA) was diluted in permeabilization buffer (140 mM KCl (Wako, Cat# 163–03545), 6 mM NaCl (Wako, Cat# 191–01665), 0.1 mM EGTA (Wako, Cat# QB-6401), and 10 mM HEPES (Wako, Cat# 342–01375) [pH 7.4]) to a final concentration of 50 μg/mL α-hemolysin. GO-ATeam2-knock-in BMMNCs were added to the buffer and permeabilized for 30 min at 37 °C under 5% $CO_2$. To calibrate ATP concentration, calibration buffer (140 mM KCl, 6 mM NaCl, 0.5 mM MgCl2 (Wako, Cat# 136–03995), and 10 mM HEPES [pH 7.4]) and Mg-ATP stock solution (Sigma-Aldrich, Cat# A9187) were prepared. After washing GO-ATeam2-knock-in BMMNCs with calibration buffer, fresh calibration buffer without ATP was added. Mg-ATP was gradually added to increase ATP concentration in the cell suspension, and FRET values of the GO-ATeam2 biosensor at defined ATP concentrations were analyzed by flow cytometry. The FRET value (relative ratio of FRET to EGFP fluorescence intensities) was calculated by the following equation.

$$FRET\ value = \frac{\text{Fluorescence of FRET}}{\text{Fluorescence of EGFP}} \tag{1}$$

The excitation wavelength of FRET and EGFP was set at 488 nm.
The FRET value was then fitted to Hill's formula (*Hill, 1910*) as a function of ATP concentration:

$$\theta = \frac{[L]^n}{[K_A]^n + [L]^n} \tag{2}$$

where $\theta$ is the original percentage of receptor proteins occupied by the ligand, $[L]$ is the free (unbound) ligand concentration, $K_A$ is the concentration of ligand at half saturation, and $n$ is Hill's coefficient.
*Equation 2* was transformed as

$$\log\left(\frac{\theta}{1-\theta}\right) = n\ \log[L] - n\ \log\ K_A \tag{3}$$

such that $\theta$ could be expressed by the FRET value as follows:

$$\theta = \frac{FRET\ value - 1.4}{6} \tag{4}$$

We estimated parameters $n$ and $K_A$ by fitting observed FRET values to the linear regression model represented in *Equation 3*. In our experiment, n=3.1234, and $K_A$ = 0.84699. Using these parameters, cellular ATP concentration, $[L]$, was estimated.

## Time-course analysis of FRET values

GO-ATeam2 is a ratiometric biosensor that monitors ATP concentration through Förster resonance energy transfer (FRET) from EGFP to the monomeric version of Kusabira Orange (mKO), regardless of the sensor expression levels (*Nakano et al., 2011*). Surface-marker-stained BMMNCs from GO-ATeam2 mice were dispensed into a basal medium (Ba-M) containing minimal salts, vitamins, and buffers (HEPES and sodium bicarbonate), but no glucose or mitochondrial substrates (*Supplementary file 1*), or into a medium containing mitochondrial substrates, pyruvate, lactate, fatty acids, and amino acids, but no glucose (PLFA medium). Depending on the experiment, fresh and surface marker-stained BMMNCs obtained from mice 2, 5, or 14 days after intraperitoneal administration of PBS or 5-FU were dispensed into a Ba-M or PLFA medium. The FRET/EGFP ratio data was imported continuously during analysis in a real-time manner using the BD FACSAria IIIu under ambient pressure. Depending on their purpose, experiments were conducted in the presence or absence of various nutrients or

metabolic modulators (*Figure 4—figure supplement 1D*). For this platform, 2 mL of Ba-M or PLFA medium per tube was pre-saturated with 1% $O_2$/5% $CO_2$/94% $N_2$ to stabilize ATP levels of BMMNCs (*Figure 4—figure supplement 1E–F*) and mimic the hypoxic BM environment; when medium was not pre-saturated, ATP concentrations rapidly decreased, even in the presence of glucose, pyruvate, or lactate (*Figure 4—figure supplement 1E–F*).

To reduce the effect of autofluorescence as much as possible, the top 40–50% of EGFP and FRET fractions of MFI were used in the analysis (MFI >1000 for EGFP and FRET). Then, data reporting EGFP and FRET fluorescence values in individual cells from each gating (e.g. HSCs, MPPs) were extracted along with time course data. Relevant nutrients and inhibitors were added to medium with samples for analysis. Data acquired by the FACSAria IIIu device and retrieved as FCS files were analyzed by the flowCore package in R software. The FRET/EGFP ratio of each set of single cells was fitted to a generalized additive model using the 'gam' function in the 'mgcv' package with 's', a spline-based smoothing function, in default settings as a function of time, then smoothened using the 'predict' function. Pseudocolor plots of the FRET/EGFP ratio were created using the 'kde2d function'. If needed, fitted data were converted to ATP concentration using the model described above.

To compare changes in ATP concentrations in PBS- and 5-FU-treated groups, we corrected differences in baseline ATP concentrations by multiplying all data from the PBS-treated group by the following value: ATP concentration at 0 s in the 5-FU group/ATP concentration at 0 s in the PBS group.

## Ki67/Hoechst staining

Ki67 (BD Biosciences, Cat# 558617) and Hoechst 33432 (Invitrogen, Cat# H3570) were used for cell cycle analysis of fixed cells from C57BL/6 mice. A total of $4 \times 10^6$ BMMNCs/sample were stained with anti-CD150-APC, anti-CD48-FITC, anti-lineage (CD4, CD8a, Gr-1, Mac-1, Ter-119, B220)-PerCP-Cy5.5, anti-c-Kit-APC-Cy7, and anti-Sca-1-PE-Cy7 antibodies. To stain samples after 5-FU treatment, Mac-1 was excluded from the antibody cocktail. Stained samples were centrifuged at 340 × *g* and 4 °C for 5 min. To analyze HSCs derived from mice after in vivo 2-NBDG administration, anti-CD150-PE and anti-CD48-Alexa Fluor700 antibodies were alternatively used to sort and purify HSCs with high or low NBDG uptake. These HSCs were then subjected to Ki67/Hoechst staining. Next, 250 µL of BD Cytofix/Cytoperm (BD Biosciences, Cat# 555028) was added, and samples were incubated for 20 min at 4 °C for fixation. Fixed cells were centrifuged and washed twice at 340×*g* at 4 °C, with 1 mL BD Perm/Wash buffer (BD Biosciences, Cat# 554723) diluted 10-fold. Each sample was stained with 10 µL of Ki67-Alexa Fluor555 or Ki67-eFlour660 (for 2-NBDG stained HSC) antibodies for 1 h at RT, shaded from light. Ki67-stained cells were centrifuged and washed twice at 340 × *g* and 4 °C with PBS. Samples were resuspended in 500 µL of PBS + 10 µg/mL Hoechst 33432, filtered, and analyzed with the BD FACSAria IIIu instrument.

## FAOBlue assay

Surface marker-stained BMMNCs from PBS- or 5-FU-treated mice were dispensed at $3 \times 10^5$ cells in 500 µL Ba-M, which had been pre-saturated for 48 hr under 1% $O_2$ and 5% $CO_2$ conditions and contained 200 mg/dL glucose and 50 µM verapamil. These cells were then exposed to 5 µM FAOBlue (Funakoshi) for 15 min. As a negative control, BMMNCs were exposed to 100 µM etomoxir simultaneously with FAOBlue. The FAOBlue-stained BMMNCs were then centrifuged at 340 × *g* for 5 min at 4 °C, and the supernatant was discarded. The fluorescence of FAOBlue was excited at a wavelength of 405 nm and detected in the V 450/50 channel. After analysis, the HSC fraction data were extracted.

## Analysis of peripheral blood and BM chimerism

Periorbitally collected peripheral blood from BMT recipients was centrifuged for 3 min at 340 × *g* and the supernatant discarded. Samples were subjected to hemolysis with 1000 µL of 0.17 M $NH_4Cl$ for 40–50 min and centrifuged at 340 × *g* for 5 min. The supernatant was discarded, and samples were again subjected to hemolysis with 1000 µL of 0.17 M $NH_4Cl$ for 10–20 min. Samples were centrifuged again at 340 × *g* for 5 min and the supernatant was discarded. Pellets were then resuspended in 50 µL PBS and 0.3 µL Fc receptor block and incubated at 4 °C for 5 min. Surface antigen staining was performed using the following antibody panel: Gr-1-PE-Cy7 (BioLegend, Cat# 108416), Mac-1-PE-Cy7 (BioLegend, Cat# 101216), B220-APC (BioLegend, Cat# 103212), CD4-PerCP-Cy5.5, CD8a-PerCP-Cy5.5, CD45.1-PE (BD Biosciences, Cat# 553776), and CD45.2-FITC (BD Biosciences, Cat#

553772). An antibody cocktail was prepared by mixing 0.3 µL of each antibody. The frequency (%) of donor-derived cells was calculated as follows:

The frequency (%) of donor-derived cells = 100 × Donor-derived (Ly5.2$^+$Ly5.1$^-$) cells (%) / Donor-derived cells [%]+Competitor or recipient-derived [Ly5.2$^-$Ly5.1$^+$] cells [%]

Myeloid, B, and T cells were identified by Gr-1$^+$ or Mac-1$^+$, B220$^+$, or CD4$^+$ or CD8$^+$, respectively.

Four months after BM transplant, the frequency of donor-derived cells in BM was determined using one femur and tibia per recipient. Anti-CD150-BV421, anti-CD48-PE (BD Biosciences, Cat# 557485), anti-lineage (CD4, CD8a, Gr-1, Mac-1, Ter-119, B220)-PerCP-Cy5.5, anti-c-Kit-APC-Cy7, anti-Sca-1-PE-Cy7, anti-Ly5.1-Alexa Fluor700 (BioLegend, Cat# 110724), and anti-Ly5.2-FITC antibodies were used for surface antigen detection. An antibody cocktail was prepared by mixing 1 µL of each antibody.

## Comparison of metabolite levels before and after sorting

c-Kit-positive cells were isolated using Auto-MACS Pro (Miltenyi Biotec) with the Possel-s or Possel-d2 program as described above (see 'Cell preparation' for details). Isolated cells were counted, and $1 \times 10^5$ viable cells were dispensed into methanol containing an internal standard as a pre-sorting cell sample and stored at −80 °C until IC-MS analysis. To the isolated cell suspension, 0.1% PI was added and samples were sorted using the FACS Aria IIIu. A total of $1 \times 10^5$ viable cells (PI$^-$ cells) were sorted directly into methanol containing an internal standard as a post-sorting cell sample and stored at −80 °C until IC-MS analysis. The detected metabolites were quantified based on calibration curve data (see 'Ion chromatography mass spectrometry (IC-MS) analysis' for details).

## Preparation and storage of in vitro U-$^{13}$C$_6$-glucose tracer samples

For tracer analysis, C57BL/6 mice were euthanized to obtain 25,000–50,000 cells per sample of each fraction (HSC, MPP, GMP, CLP) from BM using the FACSAria IIIu instrument. Numbers of mice used to obtain each fraction were as follows: 30–35 each for steady state HSCs and MPPs, 60–65 each for 5-FU treated HSCs, 10 each for GMPs and CLPs. In addition, bone and BM cells were chilled by placing dishes and tubes on ice during the cell preparation process; samples were washed with ice-cold buffer throughout the entire process before cell sorting. Experiments and experimental manipulations regarding the sampling of mouse femurs and tibias were also performed in the shortest amount of time possible by skilled personnel. Cells were sorted in 0.1% BSA +PBS and sorted cells were centrifuged at 340 × $g$ and 4 °C for 5 min. After discarding the supernatant, cells were added to 1 mL pre-saturated (under 1% O$_2$ and 5% CO$_2$) GO-ATeam2 Ba-M +0.1% BSA+200 mg/dL U-$^{13}$C$_6$-(Sigma-Aldrich, Cat# 389374) or U-$^{12}$C$_6$-glucose and incubated 10 or 30 min. If the process of pre-saturation was omitted, ATP levels dropped rapidly within a short time (*Figure 4—figure supplement 1F*).When using oligomycin (1 µM Cell Signaling Technology, Cat# 9996), exposure time was set to 10 min. Samples were then immediately centrifuged at 1000 × $g$ and 4 °C for 3 min. After discarding supernatants, cells were frozen and stored at −80 °C.

## Preparation and storage of in vivo U-$^{13}$C$_6$-glucose tracer samples

U-$^{13}$C$_6$-glucose administration to C57BL/6 mice was performed based on the methods of *Jun et al., 2021*, with some modifications. Mice were intraperitoneally administered medetomidine hydrochloride, midazolam, and butorphanol tartrate at 0.75 mg/kg, 4 mg/kg, and 5 mg/kg, respectively. After anesthesia, mice were kept warm on a hot plate set at 37 °C while a 27-gauge needle was placed in the external tail vein and U-$^{13}$C$_6$-glucose was continuously administered. The dose and duration of U-$^{13}$C$_6$-glucose administration followed (*Jun et al., 2021*), and 0.4125 mg/g body mass was administered in 1 min, followed by 0.008 mg/g body mass per minute for 3 hr. After U-$^{13}$C$_6$-glucose administration, mice were euthanized by cervical dislocation and immediately placed on ice. For in vivo tracer analysis, BMMNCs from the bilateral femur, tibia, pelvis, and sternum of each mouse were used to prepare sufficient numbers of HSCs, and pre-chilled 0.1% BSA +PBS was used for BM flushing and washing. HSCs were directly sorted in methanol and stored at −80 °C until IC-MS analysis. A total of $1–3 \times 10^4$ HSCs were purified from one or two mice in the PBS group and from two or three mice in the 5-FU group.

When generating the heat map of labeling rates in each metabolite, 1 was added as a pseudo number to the labeling rate of all metabolites. When calculating the total amount of $^{13}$C labeled metabolites for each pathway, metabolites other than M+0 were summed in each metabolite.

## Metabolite extraction

Frozen samples were mixed with 500 μL methanol containing internal standards and sonicated for 10 s. Then, 200 μL ddH$_2$O (Invitrogen, Cat# 10977–015) and 400 μL chloroform (Nacalai tesque, Cat# 08402–55) were added and samples were centrifuged at 10,000 × $g$ and 4 °C for 3 min. The aqueous phase was transferred to an Amicon ultrafiltration system (Human Metabolome Technologies, Inc, Cat# UFC3LCCNB-HMT) and centrifuged at 9100 × $g$ and 4 °C for 3 hr. Filtered samples were analyzed by IC-MS.

## Ion chromatography mass spectrometry (IC-MS) analysis

For metabolome analysis focused on glycolytic metabolites and nucleotides, anionic metabolites were measured using an orbitrap-type MS (Q-Exactive Focus; Thermo Fisher Scientific, Waltham, MA, USA) connected to a high-performance IC system (ICS-5000+, Thermo Fisher Scientific), enabling highly selective and sensitive metabolite quantification owing to the IC-separation and Fourier Transfer MS principle (*Miyajima et al., 2017*). The IC instrument was equipped with an anion electrolytic suppressor (Dionex AERS 500; Thermo Fisher Scientific) to convert the potassium hydroxide gradient into pure water before the sample entered the mass spectrometer. Separation was performed using a Dionex IonPac AS11-HC-4 μm IC column (Thermo Fisher Scientific). The IC flow rate was 0.25 mL/min supplemented post-column with a 0.18 mL/min makeup flow of MeOH. The potassium hydroxide gradient conditions for IC separation were as follows: 1–100 mM (0–40 min), 100 mM (40–50 min), and 1 mM (50.1–60 min), with a column temperature of 30 °C. The Q-Exactive Focus mass spectrometer was operated under the ESI negative mode for all detections. A full mass scan (*m/z* 70–900) was performed at a resolution of 70,000. The automatic gain control target was set at 3×10$^6$ ions, and the maximum ion injection time was 100ms. Source ionization parameters were optimized with a spray voltage of 3 kV, and other parameters were as follows: transfer temperature, 320 °C; S-lens level, 50; heater temperature, 300 °C; sheath gas, 36; and aux gas, 10. Metabolite amounts were quantified from calibration curve data generated based on peak areas and respective metabolite amounts.

## Quantitative $^{13}$C-MFA with OpenMebius

OpenMebius (Open source software for $^{13}$C-MFA) provides the platform to simulate isotope labeling enrichment from a user-defined metabolic model setup worksheet developed in MATLAB (MathWorks, Natick, MA, USA; *Kajihata et al., 2014*). Quantitative $^{13}$C-MFA was performed according to a manual prepared by the software developer (http://www-shimizu.ist.osaka-u.ac.jp/hp/en/software/OpenMebius.html), but some metabolic model modifications were made to more faithfully reflect our measured data. Specifically, the model was modified to include (a) the conversion of pyruvate to lactate catalyzed by lactate dehydrogenase, (b) the formation of citrate from acetyl CoA and oxaloacetate catalyzed by citrate synthase, (c) the synthesis of alpha-ketoglutarate from citrate catalyzed by aconitase and isocitrate dehydrogenase, and (d) the synthesis of fumarate from succinate by succinate dehydrogenase. Reactions with pyruvate formate lyase performed by *Escherichia coli*, *Streptococcus spp.*, and ethanol fermentation of acetyl CoA were excluded from the default metabolic network sheet.

The lactate efflux values in $^{13}$C-MFA were determined using the following trial and error method. First, various values (0–100) were entered as candidate lactate efflux values and simulations were run to determine the optimal lactate efflux. When the lactate efflux value was set low (below 50), either the simulation could not be run and an error occurred, or the simulation resulted in the glycolytic system progressing in the opposite direction. These results suggested that the appropriate solution was not obtained because the lactate efflux was unnatural compared to the level of glycolytic metabolites. This was validated by experimental data showing that isotopic labeling rates for most glycolytic metabolites were close to 100% at short labeling times (*Figure 1—figure supplement 2C*). Therefore, we ran the simulation with a higher lactate efflux value. Finally, we set the lactate efflux to 65, which yielded reasonably satisfactory results for nearly 100% labeling of glycolytic and PPP metabolites in PBS- or DMSO-treated HSCs.

The rate of lactate efflux 5-FU-treated HSCs with the rate of glucose uptake set to 100 was defined using the following equation, with the flux in stationary phase HSC set to 65:

65×(Percentage of glycolytic metabolites labeled with $^{13}$C in the total $^{13}$C-labelled metabolites [5-FU-treated HSCs])/(Percentage of glycolytic metabolites labeled with $^{13}$C in the total $^{13}$C-labelled metabolites [PBS-treated HSC])

In the metabolic flux measurements of HSCs under mitochondrial stress, the lactate efflux determined by the above method exceeded the maximum value that could be modeled (85>), so we decreased the lactate efflux flux by 5 and adopted the maximum value, 80, at which modeling became possible. For values of efflux other than those of lactate efflux flux, the values specified by the Open-Mebius manual were used to eliminate arbitrary factors as much as possible.

The metabolic substrate used for labeling was set to 100% U-$^{13}$C$_6$ glucose. Metabolites used in the analysis included the first intermediate metabolite produced when U-$^{13}$C$_6$ glucose is metabolized (e.g. G6P or F6P with all carbons labeled, the labeled metabolite of the first cycle of the TCA cycle) and the unlabeled metabolite that was measured. Some of the labeled metabolites in the TCA cycle (e.g. citrate [M2]) and erythrose 4-phosphate (M4) in PPP were detected with non-negligible amounts of natural isotopes (>5% even when labeled with U-$^{12}$C$_6$ glucose compared to U-$^{13}$C$_6$ glucose). The presence of such natural isotopes may result in overestimation of the amount of increased labeling with U-$^{13}$C$_6$ glucose. In such cases, the amount of natural isotope detected when labeled with U-$^{12}$C$_6$ glucose was subtracted from the amount of labeled metabolite detected with U-$^{13}$C$_6$ glucose. If the resulting true labeled isotope abundance was negative, the labeled amount was modeled as zero. When analyzing in MATLAB, the number of modeling cycles was set to 100, and the iteration time was set to a maximum of 2000 cycles.

## Luminometric ATP measurement

HSCs were sorted from C57BL/6 mice treated with PBS or 5-FU and dispensed into pre-saturated GO-ATeam2 medium with 0.1% BSA in a 1%O$_2$/5%CO$_2$ incubator. HSCs were then exposed to 15 μM of PFKFB3 inhibitor (AZ PFKFB3 26) or DMSO and placed in a 1%O$_2$/5%CO$_2$ incubator for 10 min. Cells were centrifuged at 4 °C and 340 × *g* and the supernatant was removed. ATP measurements were performed according to manufacturer instructions using Cell ATP Assay Reagent Ver. 2 (Toyo B-Net Corporation). The amount of ATP per cell was calculated by dividing the amount of ATP detected by the number of cells used for analysis.

## Apoptosis assay of HSC after 2-DG or oligomycin treatment

Purified C57BL/6 mouse-derived HSCs were exposed to 2-DG (50 mM) and oligomycin (1 μM) in pre-saturated 0.1% BSA +GO-ATeam2 medium under 1% O$_2$/5% CO$_2$ conditions for 10 min and subjected to apoptosis assay using the PE Annexin V Apoptosis Detection Kit I (BD Biosciences, Cat# 559763) according to manufacturer instructions.

## CRISPR/Cas9 knockout (KO) of *Pfkfb3*

Target sequences of single guide RNA (sgRNA) were provided in a previous report (*Chu et al., 2016*) and identified using the web tool GenScript (https://www.genscript.com) for *Pfkfb3*. sgRNAs were synthesized using a CUGA7 gRNA Synthesis Kit (Nippon Gene, Tokyo, Japan, Cat# 314–08691) following manufacturer instructions, diluted to 1.5 μg/μL, and cryopreserved at −80 °C until use. CD150$^+$CD48$^-$Flt3$^-$ LSK cells sorted by FACSAria IIIu were cultured in SF-O3 medium supplemented with stem cell factor (SCF) (50 ng/mL) (Peprotech, Cat# 250–03) and thrombopoietin (TPO) (Peprotech, Cat# 300–18) (50 ng/mL) (S50T50 medium) and incubated under 20% O$_2$/5% CO$_2$ conditions for 16–24 hr, enabling subsequent HSC-specific gene editing with the CRISPR-Cas9 system. Ribonucleoprotein complex preparation and electroporation were conducted as previously reported (*Gundry et al., 2016*). Briefly, 3 μg Cas9 protein (TrueCut Cas9 Protein v2, Thermo Fisher Scientific, Cat# A36496) plus 3 μg of sgRNA were incubated in Buffer T (Invitrogen, Cat# MPK10096) for 20 min at RT in a volume 6 μL. Cultured cells were resuspended in 30 μL Buffer T and added to ribonucleoprotein at a total volume of 36 μL. Cells were electroporated using the Neon Transfection System (Thermo Fisher Scientific) at 1700 V for 20ms with one pulse. The cell suspension was transferred to S50T50 medium and cultured under 20% O$_2$/5% CO$_2$ conditions. To evaluate gene editing efficiency, genomic DNA from LSK cells was extracted using the NucleoSpin system (Macherey-Nagel, Dürin, Germany) 2–3 d after electroporation. PCR was performed using the following settings: 95 °C for 2 min; 35 cycles of 95 °C for 30 s, 60 °C for 30 s, and 72 °C for 30 s; followed by final extension at 72 °C for

5 min. PCR products were purified using Wizard SV Gel and the PCR Clean-Up System (Promega Corporation, Madison, WI, USA, Cat# A9281) following manufacturer instructions. A tracking of indels by decomposition (TIDE) assay (*Brinkman et al., 2014*) or inference of CRISPR edits analysis (*Conant et al., 2022*) was performed to analyze the sequence data of each PCR product obtained by Sanger sequencing. Among five sgRNAs, Pfkfb3-sg1 displayed the best editing efficiency and was used for subsequent transplant and culture experiments.

## BM transplant of *Pfkfb3*-KO HSCs

Either *Rosa26* (control) or *Pfkfb3* sequences in HSCs were targeted using CRISPR/Cas9. After electroporation, HSCs were incubated for 2–3 hr in S50T50 medium under 5%$CO_2$/20%$O_2$ conditions, and then counted using a TC10 Automated Cell Counter (Bio-Rad Laboratories, Inc, Hercules, CA, USA). Subsequently, 500 gene-edited HSCs together with $2×10^6$ BM cells from Ly5.1 congenic mice were transplanted retro-orbitally into lethally (9.5 Gy using MBR-1520R with a 125 kV 10 mA, 0.5 mm Al, 0.2 mm Cu filter)-irradiated Ly5.1 mice. During *Pfkfb3* KO using the vector-free CRISPR-Cas9 system, the KO efficiency was not 100%, so the transplanted cells were a mixture of *Pfkfb3*-KO cells and wild-type cells. Therefore, after 2, 8, and 16 weeks, peripheral blood was collected and donor-derived chimerism was assessed by a TIDE assay based on a recent study by *Shiroshita et al., 2022*. The following oligonucleotides for sgRNA synthesis and primers for post-knockout genomic PCR were used.

> For *Rosa26* region KO: sgRNA target: 5'-ACTCCAGTCTTTCTAGAAGA-3'
> Forward primer 1: 5'-CCAAAGTCGCTCTGAGTTGTTATCAGT-3'
> Reverse primer 1: 5'-GGAGCGGGAGAAATGGATATGAAG-3'
> Forward primer 2: 5'-CCAAAGTCGCTCTGAGTTGTTATCAGT-3'
> Reverse primer 2: 5'-GGAGCGGGAGAAATGGATATGAAG-3'
> Sequence primer: 5'-ACATAGTCTAACTCGCGACAC-3'
> For *Pfkfb3* KO: sgRNA target: 5'-GTTGGTCAGCTTCGGCCCAC-3
> Forward primer: 5'-AATTGTGTAGCACAGGATCACC-3'
> Reverse primer: 5'-GCCACTAAAGGAAGGCTAGTTAC-3'
> Sequence primer: 5'-CTCAATCTTCCCGAGTCTGTCTC-3'
> For *CD45* KO: sgRNA target: 5'-GGGTTTGTGGCTCAAACTTC-3'
> Forward primer: 5'-AGAAGCCATTGCACTGACTTTG-3'
> Reverse primer: 5'-GTGTGATCTTTCCCCGAAACAT-3'
> Sequence primer: 5'-CTGCAAAGAGGACCCTTTACAGT-3'

To calculate the KO efficiency of the *Rosa26* locus, primer 1 or primer 2 was used for PCR amplification.

## *Pfkfb3* overexpression in GO-ATeam2[+] HSCs and time-course analysis of FRET values

cDNA encoding *Pfkfb3* was subcloned into pMY-IRES-hCD8 upstream of IRES-hCD8. To produce a recombinant retrovirus, plasmid DNA was transfected into Plat-E cells using FuGENE HD Transfection Reagent (Promega, Cat# E2311). Cell supernatants were then used to transduce GO-ATeam2[+] HSCs pre-cultured with SCF and TPO for 16 hr. At 48 hr post-transduction, surface-marker-stained, retrovirally *pfkfb3*-overexpressed GO-ATeam2[+] cells were used for time-course analysis of FRET values as described above subsection 'Time-course analysis of FRET values'. Cells transduced with pMY-IRES-hCD8 retrovirus served as controls. Transduced cells were stained with the following antibody panel: lineage markers (CD4, CD8a, Gr-1, Mac-1, Ter-119, B220)-PerCP-Cy5.5, c-Kit-APC-Cy7, Sca-1-PE-Cy7, CD150-BV421, CD48-BV510 (BD Biosciences, Cat# 563536), and hCD8-APC (BioLegend, Cat# 980904). FRET value data for hCD8-positive cells were used for subsequent conversion to ATP concentration.

## *Pfkfb3*/*Pfkfb3*CA overexpression in HSCs and BMT

cDNA encoding *Pfkfb3* or the constitutively active S461E *Pfkfb3* mutant (*Pfkfb3*CA *Bando et al., 2005*) was subcloned into pMY-IRES-hCD8 upstream of IRES-hCD8 or into pMY-IRES-EGFP upstream of IRES-EGFP (*Nosaka et al., 1999*), respectively. To produce a recombinant retrovirus, plasmid

DNA was transfected into Plat-E cells using the FuGENE HD Transfection Reagent. Cell supernatants containing virus were then filtered with Millex-HV Syringe Filter Unit (0.45 µm, PVDF, 33 mm, gamma sterilized, Millipore) and used to transduce Ly5.1$^+$ HSCs pre-cultured in SCF and TPO for 16 hr.

At 48 hr post-transduction, 2000 transduced GFP$^+$ cells were sorted and transplanted, together with 4×10$^5$ BMMNCs from C57BL/6-Ly5.2 mice, into lethally (9.5 Gy using MBR-1520R with a 125 kV 10 mA, 0.5 mm Al, 0.2 mm Cu filter)-irradiated C57BL/6-Ly5.2 mice. Cells transduced with pMY-IRES-EGFP retrovirus served as controls. After 1–4 months, peripheral blood was collected and donor-derived chimerism was analyzed by flow cytometry. The frequency (%) of donor-derived cells was calculated as follows:

100×Donor-derived (Ly5.2$^-$Ly5.1$^+$) cells (%) / (Donor-derived cells [%]+Competitor or recipient-derived [Ly5.2$^+$Ly5.1$^-$] cells [%])

## Knockout and overexpression of *Pfkfb3* in HSPC and non-competitive BMT

PFKFB3 was knocked out and overexpressed in FACS-sorted Lin$^-$Sca-1$^+$c-Kit$^+$ and Ly5.2$^+$ cells, respectively. Methods were partially modified from those described in the 'CRISPR/Cas9 KO of *Pfkfb3*' and '*Pfkfb3*/*Pfkfb3*CA overexpression in HSCs and BMT' sections.

For KO of *Pfkfb3*, triple-gRNA purchased from Synthego (Redwood City, CA, USA) was used. After gene editing, Ly5.2$^+$ HSPCs were collected and cultured in S50T50 medium under 5% $CO_2$/20% $O_2$ conditions for 2–3 hr, and 3×10$^5$ HSPCs were transplanted retro-orbitally into lethally-irradiated (8.5 Gy using MBR-1520R-3 (Hitachi Power Solutions) with a 125 kV 10 mA, 0.5 mm Al, 0.2 mm Cu filter) recipient Ly5.1 mice noncompetitively.

The sequences of triple-gRNA and the primer set used to confirm KO efficiency were as follows.

> sgRNA sequences:
> 5'-AGACCUGGCUUACCUUUCGU-3'
> 5'-UGGAGAUGUAAGUCUUACCC-3'
> 5'-GUUGGUCAGCUUCGGCCCAC-3'
> Forward Primer: 5'-CAAAGGAAAAGTCCCATGGAGA-3'
> Reverse Primer: 5'-GGGCTTTGGCATGTGGAATG-3'
> Sequencing Primer: 5'-CAAAGGAAAAGTCCCATGGAGAATG-3'

For *Pfkfb3* overexpression, HSPCs were cultured in S50T50 medium under 5% $CO_2$/20% $O_2$ conditions for 8–16 hr after retroviral transduction, and the equivalent of 3×10$^5$ HSPCs were noncompetitively transplanted retro-orbitally into lethally-irradiated (8.5 Gy using MBR-1520R-3) recipient Ly5.1 mice. After transduction, a group of the cells was cultured in S50T50 medium for 48 hr to confirm that transduction (GFP positivity) had been established.

## Cell cycle analysis and apoptosis assay of *Pfkfb3*-KO/overexpressing HSPCs after non-competitive BMT

BMMNCs were collected from the bilateral femur, tibia, pelvic bone, and sternum of each individual recipient mouse on day 2 after noncompetitive BMT. Recipient BMMNCs were then stained with Lineage-marker-PerCP-Cy5.5, Ly5.1-PerCP-Cy5.5, and Ly5.2-PE (cell cycle analysis) or Lineage-marker-FITC, Ly5.1-FITC, and Ly5.2-Alexa Fluor700 (apoptosis assay). For the analysis, all BMMNCs from each recipient were used in one analysis, and all lineage-marker negative Ly5.2$^+$ cells were analyzed. Cell cycle analysis (Ki67/Hoechst33432 staining) was performed as described in the 'Ki67/Hoechst33432 staining' section. In vivo BrdU labeling assays were performed as reported by *Jun et al., 2021* using the FITC BrdU Flow Kit (BD Biosciences, Cat# 559619). Apoptosis assays were performed using the PE Annexin V Apoptosis Detection Kit I according to manufacturer instructions.

Cell cycle analysis (Ki67/Hoechst33432 staining) of *Pfkfb3*-overexpressing HSPCs after transplantation was also performed using all BMMNCs from each recipient mouse, and the analysis was performed on all *Pfkfb3*-overexpressing cells (GFP$^+$).

## 5-FU administration after BM recovery in *Pfkfb3*-KO HSPCs

PFKFB3 was gene-edited in HSPCs using triple-gRNA as described above, and the equivalent of 3×10$^5$ LSK cells were transplanted retro-orbitally into lethally-irradiated (8.5 Gy using MBR-1520R-3)

recipient Ly5.1 mice noncompetitively. After 2 months, recipient mice were treated with 150 mg/kg of 5-FU intraperitoneally. Peripheral blood was collected on the day of 5-FU administration (day 1), and on days 4, 6, 9, and 16. The dynamics of *Pfkfb3*- or *Rosa26*-KO cell abundance (as control group) were analyzed by Sanger sequencing as described above.

## Homing assay of *Pfkfb3*-KO HSPCs

PFKFB3 was gene-edited in GFP[+] HSPCs using triple-gRNA as described above. After editing, $2 \times 10^5$ cells were retro-orbitally transplanted into lethally-irradiated (8.5 Gy) C57BL/6 mice. After 16 hours, BMMNCs from recipients were stained for surface antigens and analyzed for the percentage of GFP [+] cells within the PI-negative cells.

## Immunocytochemistry

HSCs from PBS- or 5-FU-treated C57BL/6 mice were subjected to immunocytochemistry using antibodies for PFKFB3 (Abcam, Cat# ab181861), phosphorylated-PFKFB3 (Bioss, Cat# bs-3331R), and methylated-PFKFB3 (developed by Takehiro Yamamoto) (*Yamamoto et al., 2014*). Purified HSCs were resuspended in 50% FCS-PBS and cytospun using the Thermo Scientific Cytospin 4 system (Thermo Fisher Scientific). When using 2-NBDG-positive or -negative HSCs, C57BL/6 mice were given 2-NBDG intravenously (see 'In vivo 2-NBDG assay' for details) and subjected to cytospinning. Cytospun cells were fixed using 4% paraformaldehyde in PBS pH 7.4 for 10 min at RT. Fixed cells were washed twice with ice-cold PBS. For permeabilization, cells were incubated for 5 min with PBS containing 0.1% Triton X-100. Permeabilized cells were washed once with ice-cold PBS. After blocking with 3% BSA-PBS for 30 min, cells were incubated in the diluted antibody with 0.3% BSA-PBS in a humidified chamber overnight at 4 °C. A dilution factor of 1:100 was used for all antibodies. The next day, cells were incubated with Goat anti-Mouse IgG2a Secondary Antibody, Alexa Fluor 555 (Thermo Fisher Scientific, Cat# A-21137) and DAPI in 0.3% BSA-PBS for 1 hr at RT. After two washes with ice-cold PBS, samples were coverslipped with a drop of mounting medium and imaged with a Zeiss LSM 880 microscope (ZEISS, Jena, Germany). Images were acquired at room temperature under darkened conditions using a 100 x oil immersion lens. The obtained image data was analyzed using Imaris software (Bitplane) to calculate the MFI of the target for each cell.

## RNA sequencing

Library preparation for RNA-seq was performed on 3000–3500 HSCs derived from mice after 5-FU or PBS administration. Total RNA was prepared using Rneasy Micro kit (QIAGEN, Hilden, Germany). cDNA was synthesized and amplified using SMART-Seq v4 Ultra Low Input RNA Kit for Sequencing (Takara Bio, Inc, Shiga, Japan). RNA-seq libraries were prepared using the Nextera XT Kit (Illumina, San Diego, CA, USA). Single-end 75 bp sequencing was performed on a NextSeq 500 platform (Illumina). RNA-seq data were obtained from three independent experiments (biological replicates) for each cell type. TopHat (version 2.0.13; with default parameters) was used for mapping to the reference genome (UCSC/mm10) with annotation data from iGenomes (Illumina). Then, gene expression levels were quantified using Cuffdiff (Cufflinks version 2.2.1; with default parameters).

## MACSQuant analysis of cell number

After single GO-ATeam2 knock-in HSC culture, most of the medium (150–170 µL) in wells of a 96-well plate was aspirated and samples were stained with 10 µL antibody cocktail for 30 min at 4 °C. Antibodies used were anti-lineage markers (CD4, CD8a, Gr-1, Mac-1, B220, Ter-119)-PerCP-Cy5.5, anti-c-Kit-APC-Cy7, anti-Sca-1-PE-Cy7, anti-CD150-BV421, and anti-CD48-APC for LSK-SLAM analysis. Antibody cocktail was prepared by mixing 0.1 µL of each antibody. After incubation, 100 µL PBS +2% FCS was added to wells, and the plates were centrifuged for 5 min at 4 °C and 400 × *g* with low acceleration and medium deceleration. Then, 100 µL supernatant was aspirated and cell pellets were resuspended in 200 µL PBS +2% FCS+0.1% PI+0.25% Flow-Check Fluorospheres (Beckman Coulter, Brea, CA, USA, Cat# A69183). Samples were acquired in fast mode in the MACSquant analysis settings, and volumes of 100 µL (large colonies) or 150–170 µL (small colonies) were analyzed. Data were exported as FCS files and analyzed using FlowJo software. Cell number was corrected by bead count of Flow-Check (~1000 cells/µL). HSCs were counted using CD150[+]CD48[-]LSK cell counts.

Megakaryocytes were identified as cells with high forward scatter and side scatter, as well as high CD150 and CD41 expression.

## cDNA synthesis and quantitative RT-PCR

cDNA synthesis and RT-PCR using PFKFB3CA overexpressing cells were performed as previously reported. The primers used were as follows:

> MA069663-F: 5'-GGGCATGGCGAGAATGAGTACAA-3'
> MA069663-R: 5'-TTCAGCTGGGCTGGTCCACAC-3'

## Statistical analysis

Data are presented as means ± SD unless otherwise stated. For multiple comparisons, statistical significance was determined by Tukey's multiple comparison test using the Tukey HSD function in the $R \times 64$ 4.0.3 software (R Core Team, Vienna, Austria). A paired or unpaired two-tailed Student's $t$-test and two-way ANOVA with Sidak's test were used for experiments with two groups. A p-value < 0.05 was considered statistically significant.

## Acknowledgements

We thank E Lamar for preparation of the manuscript, T Kitamura for providing mVenus-p27K- mice, and N Toyama-Sorimachi and H Shindou for their critical reading of the manuscript. This work was supported in part by KAKENHI grants from MEXT/JSPS (JP19K17847, JP21K08431 to HK; JP19K17877, JP21J01690, JP22K08493 to DK; JP18H02845, JP20K21621, JP21H02957, JP 22K19550 to K.T.), AMED grants (JP22zf0127007 to MS; JP18ck0106444, JP18ae0201014, JP20bm0704042, JP20gm1210011 to KT), grants from the National Center for Global Health and Medicine (29–1015, 20A1010, 23A1004 to HK; 26–001, 21A2001, 23A2002 to KT), the Takeda Science Foundation (to DK and KT), a JB Research Grant (to DK), Kaketsuken Grant for Young Researchers (KT), the Human Biology Microbiome Quantum Research Center (WPI-Bio2Q) supported by MEXT (to MS), and the MEXT Joint Usage/Research Center Program at the Advanced Medical Research Center, Yokohama City University (to KT).

## Additional information

### Funding

| Funder | Grant reference number | Author |
|---|---|---|
| MEXT/JSPS | JP19K17847 | Hiroshi Kobayashi |
| MEXT/JSPS | JP19K17877 | Daiki Karigane |
| MEXT/JSPS | JP18H02845 | Keiyo Takubo |
| Japan Agency for Medical Research and Development | JP22zf0127007 | Makoto Suematsu |
| Japan Agency for Medical Research and Development | JP18ck0106444 | Keiyo Takubo |
| National Center for Global Health and Medicine | 29-1015 | Hiroshi Kobayashi |
| National Center for Global Health and Medicine | 26-001 | Keiyo Takubo |
| Takeda Science Foundation | | Daiki Karigane Keiyo Takubo |
| JB Research Grant | | Daiki Karigane |

| Funder | Grant reference number | Author |
|---|---|---|
| Ministry of Education, Culture, Sports, Science and Technology | Human Biology Microbiome Quantum Research Center (WPI-Bio2Q) | Makoto Suematsu |
| Ministry of Education, Culture, Sports, Science and Technology | MEXT Joint Usage/ Research Center Program at the Advanced Medical Research Center Yokohama City Univ | Keiyo Takubo |
| MEXT/JSPS | JP21K08431 | Hiroshi Kobayashi |
| MEXT/JSPS | JP21J01690 | Daiki Karigane |
| MEXT/JSPS | JP22K08493 | Daiki Karigane |
| MEXT/JSPS | JP20K21621 | Keiyo Takubo |
| MEXT/JSPS | JP21H02957 | Keiyo Takubo |
| MEXT/JSPS | JP 22K19550 | Keiyo Takubo |
| Japan Agency for Medical Research and Development | JP18ae0201014 | Keiyo Takubo |
| Japan Agency for Medical Research and Development | JP20bm0704042 | Keiyo Takubo |
| Japan Agency for Medical Research and Development | JP20gm1210011 | Keiyo Takubo |
| National Center for Global Health and Medicine | 20A1010 | Hiroshi Kobayashi |
| National Center for Global Health and Medicine | 23A1004 | Hiroshi Kobayashi |
| National Center for Global Health and Medicine | 21A2001 | Keiyo Takubo |
| National Center for Global Health and Medicine | 23A2002 | Keiyo Takubo |
| Kaketsuken Grant for Young Researchers | | Keiyo Takubo |

The funders had no role in study design, data collection and interpretation, or the decision to submit the work for publication.

## Author contributions

Shintaro Watanuki, Data curation, Formal analysis, Validation, Investigation, Visualization, Methodology, Writing – original draft; Hiroshi Kobayashi, Funding acquisition, Visualization, Methodology, Writing – review and editing; Yuki Sugiura, Data curation, Formal analysis, Investigation, Methodology; Masamichi Yamamoto, Resources, Methodology; Daiki Karigane, Funding acquisition, Methodology; Kohei Shiroshita, Shinya Fujita, Takayuki Morikawa, Methodology; Yuriko Sorimachi, Formal analysis, Investigation; Shuhei Koide, Akira Nishiyama, Koichi Murakami, Data curation, Investigation; Motohiko Oshima, Data curation; Miho Haraguchi, Shinpei Tamaki, Investigation; Takehiro Yamamoto, Tomohiro Yabushita, Yosuke Tanaka, Go Nagamatsu, Hiroaki Honda, Shinichiro Okamoto, Nobuhito Goda, Tomohiko Tamura, Resources; Ayako Nakamura-Ishizu, Validation, Writing – review and editing; Makoto Suematsu, Resources, Funding acquisition, Validation; Atsushi Iwama, Toshio Suda, Resources, Validation; Keiyo Takubo, Conceptualization, Resources, Data curation, Formal analysis, Supervision, Funding acquisition, Validation, Investigation, Writing – original draft, Project administration, Writing – review and editing

## Author ORCIDs

Shintaro Watanuki (iD) http://orcid.org/0000-0002-9229-6712

Hiroshi Kobayashi (ID) https://orcid.org/0000-0002-0924-1252
Yuki Sugiura (ID) http://orcid.org/0000-0002-6983-8958
Daiki Karigane (ID) https://orcid.org/0000-0002-4017-5193
Takehiro Yamamoto (ID) https://orcid.org/0000-0003-4974-9859
Atsushi Iwama (ID) https://orcid.org/0000-0001-9410-8992
Keiyo Takubo (ID) https://orcid.org/0000-0002-1736-7592

### Ethics

Work involving animal experimentation had been conducted according to local ethical standards. All experimental procedures were approved by a local ethical committee (permits 2023-A007 and 2023-D018).

Review #1 (Public review) https://doi.org/10.7554/eLife.87674.3.sa1
Review #2 (Public review) https://doi.org/10.7554/eLife.87674.3.sa2
Author response https://doi.org/10.7554/eLife.87674.3.sa3

## Additional files

### Supplementary files

• Supplementary file 1. Custom RPMI medium for culture and ATP analysis. Composition of custom RPMI medium for culture (upper) and ATP analysis (lower). "-" means 0 mg/L.

• Supplementary file 2. In vitro tracer analysis for 5-FU-treated HSCs. Results of tracer analysis using U-$^{13}C_6$-glucose with HSCs from mice treated with PBS or 5-FU. Each section contains raw data from the glycolytic system, TCA cycle, and $P\sim$NAS from top to bottom. Data from three individual experiments are described for each. All values represent average metabolite levels in single HSCs obtained by dividing the metabolite levels detected in HSCs (compared to internal standards) by the number of HSCs used in the analysis.

• Supplementary file 3. In vitro tracer analysis for oligomycin-treated HSCs. Results of tracer analysis using U-$^{13}C_6$-glucose with HSCs treated with DMSO (Oligomycin-) or oligomycin (Oligomycin+). Each section contains raw data from the glycolytic system, TCA cycle, and $P\sim$NAS from top to bottom. Data from four individual experiments are described for each. All values represent average metabolite levels in single HSCs, obtained by dividing the metabolite levels detected in HSCs (compared to internal standards) by the number of HSCs used in the analysis.

• Supplementary file 4. $^{13}C$ quantitative metabolic flux analysis. Metabolic flux values of each enzyme obtained from 100 trials of $^{13}C$ quantitative metabolic flux analysis for PBS-treated (left), 5-FU-treated (middle), and OXPHOS-inhibited HSCs (right).

• Supplementary file 5. In vivo tracer analysis for 5-FU treated mice. Results of tracer analysis during continuous in vivo administration of U-$^{13}C_6$-glucose to mice treated with 5-FU or PBS. A sheet is prepared for each metabolite and each contains two tables. The A.U. table (left) shows the metabolite levels detected in the four biological replicates in the 5-FU and PBS groups, obtained by dividing the metabolite levels detected in HSCs (compared to internal standards) by the number of HSCs used in the analysis. The ratio table (right) shows the calculated percentage of labeled metabolites among detected metabolites, where 12 C indicates unlabeled metabolites and 13Cn indicates n-carbon labeled metabolites by U-$^{13}C_6$-glucose.

• Supplementary file 6. Primer list for genotyping PCR.

• MDAR checklist

### Data availability

RNA sequence data were deposited in GEO (accession number GSE260765). All data generated or analyzed during this study are included in the manuscript and supporting files; source data files have been provided for all figures.

The following dataset was generated:

| Author(s) | Year | Dataset title | Dataset URL | Database and Identifier |
|---|---|---|---|---|
| Watanuki S, Kobayashi H, Sorimachi Y, Haraguchi M, Tamaki S, Murakami K, Nishiyama A, Tamura T, Takubo K | 2024 | Context-Dependent Modification of PFKFB3 in Hematopoietic Stem Cells Promotes Anaerobic Glycolysis and Ensures Stress Hematopoiesis | https://www.ncbi.nlm.nih.gov/geo/query/acc.cgi?acc=GSE260765 | NCBI Gene Expression Omnibus, GSE260765 |

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

# Appendix 1

## Appendix 1—key resources table

| Reagent type (species) or resource | Designation | Source or reference | Identifiers | Additional information |
|---|---|---|---|---|
| Strain, strain background (*Mus musculus*, male and female) | C57BL/6JJmsSlc Ly5.2⁺ | Japan SLC (Shizuoka, Japan) | N/A | |
| Strain, strain background (*Mus musculus*, male and female) | C57BL/6J-Ly5.1 | CLEA Japan (Shizuoka, Japan) | N/A | Utilized for hematopoietic cell transplantation studies to distinguish donor and recipient cells. |
| Strain, strain background (*Mus musculus*, male and female) | GO-ATeam2 mice | Generated in Yamamoto laboratory | N/A | Used for ATP analysis |
| Strain, strain background (*Mus musculus*, male and female) | Ubc-GFP mice | The Jackson Laboratory | Stock No: 007076 | |
| Strain, strain background (*Mus musculus*, male and female) | mVenus-p27K- mice | Provided by Kitamura Laboratory | N/A | Used for cell cycle analysis |
| Antibody | Anti-mouse CD4-PerCP-Cy5.5 (clone: RM4-5, rat monoclonal) | TONBO biosciences | Cat# 65–0042 U100; RRID:AB_2621876 | (0.5 µL, 1 µL, or 2 µL/mouse) |
| Antibody | Anti-mouse CD8a-PerCP-Cy5.5 (clone: 53–6.7, rat monoclonal) | TONBO biosciences | Cat# 65–0081 U100; RRID:AB_2621882 | (0.5 µL, 1 µL, or 2 µL/mouse) |
| Antibody | Anti-mouse B220-PerCP-Cy5.5 (clone: RA3-6B2, rat monoclonal) | TONBO biosciences | Cat# 65–0452 U100; RRID:AB_2621892 | (0.5 µL, 1 µL, or 2 µL/mouse) |
| Antibody | Anti-mouse B220-APC (clone: RA3-6B2, rat monoclonal) | BioLegend | Cat# 103212; RRID:AB_312997 | (0.5 µL, 1 µL, or 2 µL/mouse) |
| Antibody | Anti-mouse Ter-119-PerCP-Cy5.5 (clone: TER-119, rat monoclonal) | TONBO biosciences | Cat# 65–5921 U100 | (0.5 µL, 1 µL, or 2 µL/mouse) |
| Antibody | Anti-mouse Gr1 (Ly-6G/6 C)-PerCP-Cy5.5 (clone: RB6-8C5, rat monoclonal) | BioLegend | Cat# 108428; RRID:AB_893558 | (0.5 µL, 1 µL, or 2 µL/mouse) |
| Antibody | Anti-mouse Gr1-PE-Cy7 (clone: RB6-8C5, rat monoclonal) | TONBO biosciences | Cat# 60–5931 U100; RRID:AB_2621870 | (0.5 µL, 1 µL, or 2 µL/mouse) |
| Antibody | Anti-mouse Mac1 (CD11b)-PerCP-Cy5.5 (clone: M1/70, rat monoclonal) | TONBO biosciences | Cat# 65–0112 U100; RRID:AB_2621885 | (0.5 µL, 1 µL, or 2 µL/mouse) |
| Antibody | Anti-mouse Mac1-PE-Cy7 (clone: M1/70, rat monoclonal) | TONBO biosciences | Cat# 60–0112 U100; RRID:AB_2621836 | (0.5 µL, 1 µL, or 2 µL/mouse) |
| Antibody | Anti-mouse CD45.1-PE (clone: A20, mouse monoclonal) | BD biosciences | Cat# 553776; RRID:AB_395044 | (1 µL/mouse) |
| Antibody | Anti-mouse CD45.1-Alexa Fluor700 (clone: A20, mouse monoclonal) | BioLegend | Cat# 110724; RRID:AB_493733 | (1 µL/mouse) |
| Antibody | Anti-mouse CD45.2-FITC (clone: 104, mouse monoclonal) | BD biosciences | Cat# 553772; RRID:AB_395041 | (1 µL/mouse) |
| Antibody | Anti-mouse Sca-1 (Ly-6A/E)-PE-Cy7 (clone: E13-161.7, rat monoclonal) | BioLegend | Cat# 122514; RRID:AB_756199 | (0.5 µL, 1 µL, or 2 µL/mouse) |
| Antibody | Anti-mouse c-Kit (CD117)-APC-Cy7 (clone: 2B8, rat monoclonal) | BioLegend | Cat# 105826; RRID:AB_1626278 | (0.5 µL, 1 µL, or 2 µL/mouse) |
| Antibody | CD117 MicroBeads Mouse | Miltenyi Biotec | Cat# 130-091-224 | (1:5) |
| Antibody | Anti-mouse CD150-PE (clone: TC15-12F12.2, rat monoclonal) | BioLegend | Cat# 115904; RRID:AB_313683 | (0.5 µL, 1 µL, or 2 µL/mouse) |
| Antibody | Anti-mouse CD150-BV421 (clone: TC15-12F12.2, rat monoclonal) | BioLegend | Cat# 115926; RRID:AB_2562190 | (0.5 µL, 1 µL, or 2 µL/mouse) |
| Antibody | Anti-mouse CD150-APC (clone: TC15-12F12.2, armenian hamster monoclonal) | BioLegend | Cat# 115910; RRID:AB_493460 | (0.5 µL, 1 µL, or 2 µL/mouse) |

*Appendix 1 Continued on next page*

*Appendix 1 Continued*

| Reagent type (species) or resource | Designation | Source or reference | Identifiers | Additional information |
|---|---|---|---|---|
| Antibody | Anti-mouse CD48-FITC (clone: HM48-1, armenian hamster monoclonal) | BioLegend | Cat# 103404; RRID:AB_313019 | (0.5 µL, 1 µL, or 2 µL/mouse) |
| Antibody | Anti-mouse CD48-APC (clone: HM48-1, qrmenian hamster monoclonal) | BioLegend | Cat# 103411; RRID:AB_571996 | (0.5 µL, 1 µL, or 2 µL/mouse) |
| Antibody | Anti-mouse CD48-BV510 (clone: HM48-1, armenian hamster monoclonal) | BD biosciences | Cat# 563536 | (0.5 µL, 1 µL, or 2 µL/mouse) |
| Antibody | Anti-mouse CD48-Alexa Fluor700 (clone: HM48-1, armenian hamster monoclonal) | BioLegend | Cat# 103426; RRID:AB_10612754 | (0.5 µL, 1 µL, or 2 µL/mouse) |
| Antibody | Anti-mouse CD41-APC (clone: MWReg30, rat monoclonal) | BioLegend | Cat# 133914; RRID:AB_11125581 | (0.5 µL, 1 µL, or 2 µL/mouse) |
| Antibody | Anti-CD34-BV421 (clone: RAM34, rat monoclonal) | BD biosciences | Cat# 562608; RRID:AB_11154576 | (0.5 µL, 1 µL, or 2 µL/mouse) |
| Antibody | Anti-CD34-FITC (clone: RAM34, rat monoclonal) | Invitrogen | Cat# 11-0341-82; RRID:AB_465021 | (0.5 µL, 1 µL, or 2 µL/mouse) |
| Antibody | Anti-Flt3 (CD135)-APC (clone: A2F10, rat monoclonal) | BioLegend | Cat# 135310; RRID:AB_2107050 | (0.5 µL, 1 µL, or 2 µL/mouse) |
| Antibody | Anti-CD127 (IL-7Rα) (clone: A7R34, rat monoclonal) | BioLegend | Cat# 135023; RRID:AB_10897948 | (0.5 µL, 1 µL, or 2 µL/mouse) |
| Antibody | Anti-CD201 (EPCR)-PE (clone: RCR-16, rat monoclonal) | BioLegend | Cat# 141504; RRID:AB_10899579 | (0.5 µL, 1 µL, or 2 µL/mouse) |
| Antibody | Anti-Ki67-Alexa Fluor555 (clone: B56, mouse monoclonal) | BD biosciences | Cat# 558617 | (10 µL/sample) |
| Antibody | Anti-Ki67 Monoclonal Antibody (SolA15), eFluor 660, eBioscience (Clone: SolA15, mouse monoclonal) | Invitrogen | Cat# 50-5698-82; RRID:AB_2574235 | (10 µL/sample) |
| Antibody | Fc-block (anti-mouse CD16/32) (clone: 2.4-G2, rat monoclonal) | BD biosciences | Cat# 553142; RRID:AB_394657 | (2 µL/mouse) |
| Antibody | Anti-CD16/CD32 Monoclonal Antibody (93), Alexa Fluor 700 (clone: 93, rat monoclonal) | Invitrogen | Cat# 56-0161-82; RRID:AB_493994 | (2 µL/mouse) |
| Antibody | Phospho-Rb (Ser807/811) (D20B12) XP (rabbit monoclonal) | Cell Signaling Technology | Cat# 8516 | (1:200) |
| Antibody | Anti-human CD8-APC (clone: SK1, mouse monoclonal) | Biolegend | Cat# 344721; RRID:AB_2075390 | (1 µL/sample) |
| Antibody | Recombinant anti-PFKFB3 antibody (rabbit monoclonal) | Abcam | Cat# ab181861 | (1:100) |
| Antibody | Anti-PFK2 (Ser467) antibody (rabbit polyclonal) | Bioss | Cat# bs-3331R | (1:100) |
| Antibody | Recombinant anti-methyl-PFKFB3 antibody (rabbit polyclonal) | Obtained from Takehiro Yamamoto at Keio University DOI: 10.1038/ncomms4480 | N/A | (1:100) |
| Gene (*Mus musculus*) | *Pfkfb3* | This paper | N/A | Details are as described in Methods |
| Gene (*Mus musculus*) | *Pfkfb3CA* | This paper | N/A | Details are as described in Methods |
| Recombinant DNA reagent | pMYs-IRES-GFP | Obtained from Toshio Kitamura at IMUST | N/A | Used as backbone vector for gene overexpression |
| Recombinant DNA reagent | pMYs-IRES-human CD8 | Obtained from Go Nagamatsu at Kyushu University | N/A | Used as backbone vector for gene overexpression |

*Appendix 1 Continued on next page*

*Appendix 1 Continued*

| Reagent type (species) or resource | Designation | Source or reference | Identifiers | Additional information |
|---|---|---|---|---|
| Recombinant DNA reagent | *Pfkfb3*-knockout gRNA (s) | Custom made in lab or purchased from Synthego, Inc. | N/A | Details are as described in Methods |
| Recombinant DNA reagent | *Rosa*-knockout gRNA | Custom made in lab | N/A | Details are as described in Methods |
| Recombinant DNA reagent | *CD45*-knockout gRNA | Custom made in lab | N/A | Details are as described in Methods |
| Chemical compound, drug | IST | Thermo Fisher Scientific | Cat# 41400–045 | |
| Chemical compound, drug | Penicilin | Meiji Seika | PGLD755 | |
| Chemical compound, drug | Streptomycin sulfate | Meiji Seika | SSDN1013 | |
| Chemical compound, drug | Sodium selenite | Nacalai Tesque | Cat# 11707–04 | |
| Chemical compound, drug | Fetal bovine serum | Biowest | Cat# S1820-500 | |
| Chemical compound, drug | Fetal bovine serum | Thermo Fisher Scientific | Cat# 10270–106 | |
| Chemical compound, drug | Bovine serum albumin | Sigma Aldrich | Cat# A4503-50G/100 G | |
| Chemical compound, drug | 2-mercapto ethanol (2-ME) 1000 x | Life Technologies | Cat# 21985–023 | |
| Chemical compound, drug | Thymidine | Tokyo Chemical Industry Co., Ltd. | Cat# T0233 | |
| Chemical compound, drug | RPMI 1640 Amino Acids Solution (50×) | Sigma Aldrich | Cat# R7131 | |
| Chemical compound, drug | MEM Vitamin Solution (100×) | Sigma Aldrich | Cat# M6895 | |
| Chemical compound, drug | L-glutamine | Sigma Aldrich | Cat# G8540 | |
| Chemical compound, drug | L-alanine | Sigma Aldrich | Cat# A7469 | |
| Chemical compound, drug | L-Serine | Sigma Aldrich | Cat# S4311 | |
| Chemical compound, drug | D(+)-Glucose | Wako | Cat# 049–31165 | |
| Chemical compound, drug | $^{13}$C-glucose | Sigma Aldrich | Cat# 389374 | |
| Chemical compound, drug | 2-NBDG | Cayman Chemical | Cat# 11046 | |
| Chemical compound, drug | Cytochalasin B | Wako | Cat# 030–17551 | |
| Chemical compound, drug | Phloretin | TCI chemicals | Cat# P1966 | |
| Chemical compound, drug | 2-morpholinoethanesulfonic acid | Wako | Cat# 341–01622 | |
| Chemical compound, drug | methionine sulfone | Alfa Aesar | Cat# A17027 | |
| Chemical compound, drug | Sodium L-lactate | Sigma Aldrich | Cat# L7022 | |
| Chemical compound, drug | Cholesterol Lipid Concentrate (250 X) | Gibco | Cat# 12531018 | |
| Chemical compound, drug | 100mM-Sodium Pyruvate Solution | Nacalai tesque | Cat# 06977–34 | |
| Chemical compound, drug | Sodium Hydroxide | Wako | Cat# 194–18865 | |
| Chemical compound, drug | 5-fluorouracil | Kyowa Hakko Kirin | N/A | |
| Chemical compound, drug | 2-Deoxy-D-Glucose | Tokyo Chemical Industry Co., Ltd. | Cat# D0051 | |
| Chemical compound, drug | Oligomycin | Cell Signaling Technology | Cat# 9996 L | |
| Chemical compound, drug | FCCP | Sigma Aldrich | Cat# C2920 | |
| Chemical compound, drug | Rotenone | Sigma Aldrich | Cat# R8875 | |
| Chemical compound, drug | Etomoxir (sodium salt) | Cayman chemical | Cat# 11969 | |
| Chemical compound, drug | 6-diazo-5-oxo-L-nor-Leucine | Cayman chemical | Cat# 17580 | |
| Chemical compound, drug | Verapamil | Sigma Aldrich | Cat# V4629 | |
| Chemical compound, drug | N-acetyl-cysteine | Tokyo Chemical Industry Co., Ltd. | Cat# A0905 | |

*Appendix 1 Continued*

| Reagent type (species) or resource | Designation | Source or reference | Identifiers | Additional information |
|---|---|---|---|---|
| Chemical compound, drug | AZ PFKFB3 26 | R&D systems | Cat# 5675 | |
| Chemical compound, drug | Dorsomorphin dihydrochloride | Santa Cruz Biotechnology | Cat# sc-361173 | |
| Chemical compound, drug | LKB1/AAK1 dual inhibitor | Chem Scene | Cat# CS-0342 | |
| Chemical compound, drug | PKM2 inhibitor(compound 3 k) | Selleck | Cat# S8616 | |
| Chemical compound, drug | Recombinant Murine SCF | PeproTech | Cat# 250–03 | |
| Chemical compound, drug | Recombinant Human TPO | PeproTech | Cat# 300–18 | |
| Chemical compound, drug | α-hemolysin | Sigma Aldrich | Cat# H9395 | |
| Chemical compound, drug | Potassium Chloride | Wako | Cat# 7447-40-7 | |
| Chemical compound, drug | Sodium Chloride | Wako | Cat# 7647-14-5 | |
| Chemical compound, drug | Calcium Nitrate Tetrahydrate | Wako | Cat# 13477-34-4 | |
| Chemical compound, drug | Magnesium Sulfate (Anhydrous) | Wako | Cat# 7487-88-9 | |
| Chemical compound, drug | Sodium Hydrogen Carbonate | Wako | Cat# 144-55-8 | |
| Chemical compound, drug | Disodium Hydrogenphosphate 12-Water | Wako | Cat# 10039-32-4 | |
| Chemical compound, drug | Glutathione reduced form | Tokyo Chemical Industry Co., Ltd. | Cat# G0074 | |
| Chemical compound, drug | Ethylene Glycol Bis(β-aminoethylether)-N,N,N',N'-tetraacetic Acid | Nacalai tesque | Cat# 15214–21 | |
| Chemical compound, drug | HEPES | Wako | Cat# 7365-45-9 | |
| Chemical compound, drug | Magnesium Chloride | Wako | Cat# 7786-30-3 | |
| Chemical compound, drug | Adenosine 5'-triphosphate magnesium salt | Sigma Aldrich | Cat# A9187 | |
| Chemical compound, drug | DMSO | Sigma Aldrich | Cat# D8418 | |
| Chemical compound, drug | Ethanol | Nacalai tesque | Cat# 14712–63 | |
| Chemical compound, drug | Methanol | Nacalai tesque | Cat# 21914–03 | |
| Chemical compound, drug | Chloroform | Nacalai tesque | Cat# 08401–65 | |
| Chemical compound, drug | Hoechst 33432 | Thermo Fisher Scientific | Cat# H3570 | (10 µg/mL) |
| Chemical compound, drug | Propidium iodide | Thermo Fisher Scientific | Cat# P3566 | (1:1000) |
| Chemical compound, drug | Flow-Check Fluorspheres | Beckman Coulter | Cat# 7547053 | |
| Chemical compound, drug | TrueCut Cas9 Protein v2 | Thermo Fisher Scientific | Cat# A36498 | |
| Chemical compound, drug | ExTaq | Takara bio | Cat# RR001 | |
| Chemical compound, drug | NotI | Nippon Gene | Cat# 312–01453 | |
| Chemical compound, drug | EcoRI | Nippon Gene | Cat# 314–00112 | |
| Chemical compound, drug | RetroNectin (Recombinant Human Fibronectin Fragment) | Takara | Cat# T100A | |
| Chemical compound, drug | UltraPure DNase_RNase-Free Distilled Water | Invitrogen | Cat# 10977015 | |
| Chemical compound, drug | GSK3368715 | MedChemExpress | Cat# HY-128717A | |
| Commercial assay or kit | RNeasy Mini Kit | QIAGEN | Cat# 74104 | |
| Commercial assay or kit | SuperScript VILO | Thermo Fisher Scientific | Cat# 11754–050 | |
| Commercial assay or kit | 2-mercapto ethanol | Sigma Aldrich | Cat# M6250 | |
| Commercial assay or kit | Flow Cytometry Size Calibration Kit (nonfluorescent microspheres) | Invitrogen | Cat# F13838 | |

*Appendix 1 Continued*

| Reagent type (species) or resource | Designation | Source or reference | Identifiers | Additional information |
|---|---|---|---|---|
| Commercial assay or kit | "Cellno" ATP assay reagent Ver.2 | Toyo B-Net Corporation | CA2-50 | |
| Commercial assay or kit | Fixation and Permeabilization Solution | BD Biosciences | Cat# 554722 | |
| Commercial assay or kit | Perm/Wash Buffer | BD Biosciences | Cat# 554723 | |
| Commercial assay or kit | CellROX Deep Red Reagent | Invitrogen | Cat# C10422 | |
| Commercial assay or kit | SMART-Seq v4 Ultra Low Input RNA Kit for Sequencing | Clontech | Cat# Z4888N | |
| Commercial assay or kit | NEBNext Ultra DNA Library Prep Kit for Illumina | New England BioLabs | Cat# E7370S | |
| Commercial assay or kit | CUGA7 gRNA Synthesis Kit | Nippon Gene | Cat# 314–08691 | |
| Commercial assay or kit | Extract-N-Amp Blood PCR Kit | Merck | Cat# XNAB2-1KT | |
| Commercial assay or kit | Wizard SV Gel and PCR Clean-Up System | Promega | Cat# A9281 | |
| Commercial assay or kit | BD Pharmingen FITC BrdU Flow Kit | BD Biosciences | Cat# 559619 | |
| Commercial assay or kit | BD Pharmingen PE Annexin V Apoptosis Detection Kit I | BD Biosciences | Cat# 559763 | |
| Commercial assay or kit | FAOBlue | Funakoshi | Cat# FDV-0033 | |
| Software, algorithm | R v3.5.2 | *R Development Core Team, 2018* | | http://www.r-project.org |
| Software, algorithm | TopHat v2.0.13 | 10.1186/gb-2013-14-4-r36; *Kim et al., 2013* | | https://ccb.jhu.edu/software/tophat/index.shtml |
| Software, algorithm | Cufflinks v2.2.1 | 10.1038/nbt.1621; *Trapnell et al., 2012* | | http://cole-trapnell-lab.github.io/cufflinks/ |
| Software, algorithm | GSEA software v4.3.0 | Broad Institute; *Subramanian et al., 2005* | | https://www.gsea-msigdb.org/gsea/index.jsp |
| Software, algorithm | FlowJo version 9 | BD Biosciences | | https://www.flowjo.com/ |
| Software, algorithm | TIDE v3.3.0 | 10.1093/nar/gku93; *Brinkman et al., 2014* | | https://tide.nki.nl/ |
| Software, algorithm | OpenMebius | 10.1155/2014/627014; *Kajihata et al., 2014* | | http://www-shimizu.ist.osaka-u.ac.jp/hp/en/software/OpenMebius.html |

