## [Editor Report · eLife assessment]

This **important** study provides novel strategies to overcome certain limitations when investigating the metabolism of hematopoietic stem cells, mainly due to their low abundance. The study provides **compelling** evidence suggesting that proliferative hematopoietic stem cells mainly use glycolysis (rather than mitochondrial OXPHOS or TCA cycle) as their primary energy source during emergency hematopoiesis. The article provides direct links between metabolic features and cell proliferation and explores alternative energy sources, and is of great interest to stem cell biologists.

---

## [Referee Report · Review #1 (Public review)]

Watanuki et al used metabolomic tracing strategies of U-13C6-labeled glucose and 13C-MFA to quantitatively identify the metabolic programs of HSCs during steady-state, cell-cycling, and OXPHOS inhibition. They found that 5-FU administration in mice increased anaerobic glycolytic flux and decreased ATP concentration in HSCs, suggesting that HSC differentiation and cell cycle progression are closely related to intracellular metabolism and can be monitored by measuring ATP concentration. Using the GO-ATeam2 system to analyze ATP levels in single hematopoietic cells, they found that PFKFB3 can accelerate glycolytic ATP production during HSC cell cycling by activating the rate-limiting enzyme PFK of glycolysis. Additionally, by using Pfkfb3 knockout or overexpressing strategies and conducting experiments with cytokine stimulation or transplantation stress, they found that PFKFB3 governs cell cycle progression and promotes the production of differentiated cells from HSCs in proliferative environments by activating glycolysis. Overall, in their study, Watanuki et al combined metabolomic tracing to quantitatively identify metabolic programs of HSCs and found that PFKFB3 confers glycolytic dependence onto HSCs to help coordinate their response to stress.

---

## [Referee Report · Review #2 (Public review)]

In the manuscript Watanuki et al. define the metabolic profile of HSCs in stress/proliferative (myelosuppression with 5-FU), and mitochondrial inhibition and homeostatic conditions. Their conclusions are that during proliferation HSCs rely more on glycolysis (as other cell types) while HSCs in homeostatic conditions are mostly dependent on mitochondrial metabolism. Mitochondrial inhibition is used to demonstrate that blocking mitochondrial metabolism results in similar features of proliferative conditions.

The authors used state-of-the-art technologies that allow metabolic readout in a limited number of cells like rare HSCs. These applications could be of help in the field since one of the major issues in studying HSCs metabolism is the limited sensitivity of the "standard" assays, which make them not suitable for HSC studies.

---

## [Author Response]

The following is the authors’ response to the original reviews.

**Reviewer #1:**
Watanuki et al used metabolomic tracing strategies of U-13C6-labeled glucose and 13C-MFA to quantitatively identify the metabolic programs of HSCs during steady-state, cell-cycling, and OXPHOS inhibition. They found that 5-FU administration in mice increased anaerobic glycolytic flux and decreased ATP concentration in HSCs, suggesting that HSC differentiation and cell cycle progression are closely related to intracellular metabolism and can be monitored by measuring ATP concentration. Using the GO-ATeam2 system to analyze ATP levels in single hematopoietic cells, they found that PFKFB3 can accelerate glycolytic ATP production during HSC cell cycling by activating the rate-limiting enzyme PFK of glycolysis. Additionally, by using Pfkfb3 knockout or overexpressing strategies and conducting experiments with cytokine stimulation or transplantation stress, they found that PFKFB3 governs cell cycle progression and promotes the production of differentiated cells from HSCs in proliferative environments by activating glycolysis. Overall, in their study, Watanuki et al combined metabolomic tracing to quantitatively identify metabolic programs of HSCs and found that PFKFB3 confers glycolytic dependence onto HSCs to help coordinate their response to stress. Even so, several important questions need to be addressed as below:

We sincerely appreciate the constructive feedback from the reviewer. Additional experiments and textual improvements have been made to the manuscript based on your valuable suggestions. In particular, the major revisions are as follows: First, we investigated the extent to which other metabolites, not limited to the glycolytic system, affect metabolism in HSCs after 5-FU treatment. Second, the extent to which PFKFB3 contributes to the expansion of the HSPC pool in the bone marrow was adjusted to make the description more accurate based on the data. Finally, we overexpressed PFKFB3 in HSCs derived from GO-ATeam2 mice and confirmed that PRMT1 inhibition did not reduce the ATP concentration. We believe that the reviewer's valuable comments have further deepened our knowledge of the significance of glycolytic activation by PFKFB3 that we have demonstrated. Our response to the "Recommendations for Authors" is listed first, followed by our responses to all "Public Review" comments as follows:

**(Recommendations For The Authors):**
1. The methods used in key experiments should be described in more detail. For example, in the section on ‘Conversion of GO-ATeam2 fluorescence to ATP concentration’, the knock-in strategy for GO-ATeam2 should be described, as well as U-13C6 -glucose tracer assays.

As per your recommendation, we have described the key experimental method in more detail in the revised manuscript: the GO-ATeam2 knock-in method was reported by Yamamoto et al. 1. Briefly, they used a CAG promoter-based knock-in strategy targeting the Rosa26 locus to generate GO-ATeam2 knock-in mice. A description of the method has been added to Methods and the reference has been added to the citation.

For the U-13C6-glucose tracer analysis, the following points were added to describe the details of the analysis: First, a note was added that the number of cells used for the in vitro tracer analysis was the number of cells used for each sample. Second, we added the solution from which the cells were collected by sorting. We added that the incubation was performed under 1% O2 and 5% CO2.

1. Confusing image label of Supplemental Figure 1H should be corrected in line 253.

We have corrected the incorrect figure caption on line 217 in the revised manuscript to "Supplemental Figure 1N" as you suggested.

1. The percentage of the indicated cell population should also be shown in Figure S1B.

As you indicated, we have included the percentages for each population in Supplemental Figure 1B.

1. Please pay attention to the small size of the marks in the graph, such as in Figure S1F and so on.

As you indicated, we have corrected the very small text contained in Figure S1F. Similar corrections have been made to Figures S1B and S5A.

1. Please pay attention to the label of line in Figure S6A-D.

Thank you very much for the advice. We have added line labels to the graph in the original Figures S6A–D.

(Specific comments)1. Based on previous reports, the authors expanded the LSK gate to include as many HSCs as possible (Supplemental Figure 1B). However, while they showed the gating strategy on Day 6 after 5-FU treatment, results from other time-points should also be displayed to ensure the strict selection of time-points.

Thank you for pointing this out. First, we did not enlarge the Sca-1 gating in this study. We apologize for any confusion caused by the incomplete description. The gating of c-Kit is based on that shown by Umemoto et al (Figure EV1A) 2, who used 250 mg/kg 5-FU, so their c-Kit reduction is more pronounced than ours.

We followed this study and compared c-Kit expression in Lin-Sca-1+CD150+CD48-EPCR+ gates to BMMNCs on day 6 after 5-FU administration (150 mg/kg). The results are shown below.

**Author response image 2. sa3fig2:** 

Since the MFI of c-Kit was downregulated, we used gating that extended the c-Kit gate to lower-expression regions on day 6 after 5-FU administration (revised Figure S1C).At other time points, LSK gating was the same as in the PBS-treated group, as noted in the Methods.

1. In Figure 1, the authors examined the metabolite changes on Day 6 after 5-FU treatment. However, it is important to consider whether there are any dynamic adjustments to metabolism during the early and late stages of 5-FU treatment in HSCs compared to PBS treatment, in order to coordinate cell homeostasis despite no significant changes in cell cycle progression at other time-points.

Thank you for pointing this out. Below are the results of the GO-ATeam2 analysis during the very early phase (day 3) and late phase (day 15) after 5-FU administration (revised Figures S7A–H).

**Author response image 3. sa3fig3:** 

In the very early phase, such as day 3 after 5-FU administration, cell cycle progression had not started (Figure S1C) and was not preceded by metabolic changes. Meanwhile, in the late phase, such as day 15 after 5-FU administration, the cell cycle and metabolism returned to a steady state. In summary, the timing of the metabolic changes coincided with that of cell cycle progression. This point is essential for discussing the cell cycle-dependent metabolic system of HSCs and has been newly included in the Results (page 11, lines 321-323).

1. As is well known, ATP can be produced through various pathways, including glycolysis, the TCA cycle, the PPP, NAS, lipid metabolism, amino acid metabolism and so on. Therefore, it is important to investigate whether treatment with 5-FU or oligomycin affects these other metabolic pathways in HSCs.

As the reviewer pointed out, ATP production by systems other than the glycolytic system of HSCs is also essential. In this revised manuscript, we examined the effects of the FAO inhibitor (Etomoxir, 100 µM) and the glutaminolysis inhibitor 6-diazo-5-oxo-L-norleucine (DON, 2mM) alone or in combination on the ATP concentration of HSCs after PBS or 5-FU treatment. As shown below, there was no apparent decrease in ATP concentration (revised Figures S7J–M).

**Author response image 4. sa3fig4:** 

Fatty acid β-oxidation activity was also measured in 5-FU-treated HSCs using the fluorescent probe FAOBlue and was unchanged compared to PBS-treated HSCs (revised Figure S7N).

**Author response image 5. sa3fig5:** 

Notably, the addition of 100 µM etomoxir plus glucose and PFKFB3 inhibitors resulted in a rapid decrease in ATP concentration in HSCs (revised Figures S7O–P). This indicates that etomoxir partially mimics the effect of oligomycin, suggesting that at a steady state, OXPHOS is driven by FAO, but can be compensated by the acceleration of the glycolytic system by PFKFB3. Meanwhile, the exposure of HSCs to PFKFB3 inhibitors in addition to 2 mM DON, which is an extremely high dose considering that the Ki value of DON for glutaminase is 6 µM, did not reduce ATP (revised Figures S7O–P). This suggests that ATP production from glutaminolysis is limited in HSCs at a steady state.

**Author response image 6. sa3fig6:** 

These points suggest that OXPHOS is driven by fatty acids at a steady state, but unlike the glycolytic system, FAO is not further activated by HSCs after 5-FU treatment. The results of these analyses and related descriptions are included in the revised manuscript (page 11, lines 332-344).

1. In part 2, they showed that oligomycin treatment of HSCs exhibited activation of the glycolytic system, but what about the changes in ATP concentration under oligomycin treatment? Are other metabolic systems affected by oligomycin treatment?

Thank you for your thoughtful comments. The relevant results we have obtained so far with the GO-ATeam2 system are as follows: First, OXPHOS inhibition in the absence of glucose significantly decreases the ATP concentration of HSCs (Figure 4C). Meanwhile, OXPHOS inhibition in the presence of glucose maintains the ATP concentration of HSCs (Figure 5B). Since it is difficult to imagine a completely glucose-free environment in vivo, it is thought that ATP concentration is maintained by the acceleration of the glycolytic system even under hypoxic or other conditions that inhibit OXPHOS.

Meanwhile, glucose tracer analysis shows that OXPHOS inhibition suppresses nucleic acid synthesis (NAS) except for the activation of the glycolytic system (Figures 2C–F). This is because phosphate groups derived from ATP are transferred to nucleotide mono-/di-phosphate in NAS, but OXPHOS, the main source of ATP production, is impaired, along with the enzyme conjugated with OXPHOS in the process of NAS (dihydroorotate dehydrogenase, DHODH). We have added a new paragraph in the Discussion section (page 17, lines 511-515) to provide more insight to the reader by summarizing and discussing these points.

1. In Figure 5M, it would be helpful to include a control group that was not treated with 2-DG. Additionally, if Figure 5L is used as the control, it is unclear why the level of ATP does not show significant downregulation after 2-DG treatment. Similarly, in Figure 5O, a control group with no glucose addition should be included.

Thank you for your advice. The experiments corresponding to the control groups in Figures 5M and O were in Figures 5L and N, respectively, but we have combined them into one graph (revised Figures 5L–M). The results more clearly show that *Pfkfb3* overexpression enhances sensitivity to 2-DG, but also enhances glycolytic activation upon oligomycin administration.

**Author response image 7. sa3fig7:** 

1. In this study, their findings suggest that PFKFB3 is required for glycolysis of HSCs under stress, including transplantation. In Figure 7B, the results showed that donor-derived chimerism in PB cells decreased relative to that in the WT control group during the early phase (1 month post-transplant) but recovered thereafter. Although the transplantation cell number is equal in two groups of donor cells, it is unclear why the donor-derived cell count decreased in the 2-week post-transplantation period and recovered thereafter in the Pfkgb3 KO group. Therefore, they should provide an explanation for this. Additionally, they only detected the percentage of donor-derived cells in PB but not from BM, which makes it difficult to support the argument for Increasing the HSPC pool.

As pointed out by the reviewer, it is interesting to note that the decrease in peripheral blood chimerism in the *Pfkfb3* knockout is limited to immediately after transplantation and then catches up with the control group (Figure 7B). We attribute this to the fact that HSPC proliferation is delayed immediately after transplantation in *Pfkfb3* deficiency, but after a certain time, PB cells produced by the delayed proliferating HSPCs are supplied. In support of this, the *Pfkfb3* knockout HSPCs did not exhibit increased cell death after transplantation (Figure 7K), while a delayed cell cycle was observed (Figures 7G–J). A description of this point has been added to the Discussion (page 19, lines 573-579).

In addition, the knockout efficiency in bone marrow cells could not be verified because the number of cells required for KO efficiency analysis was not available. Therefore, we have added a statement on this point and have toned down our overall claim regarding the extent to which PFKFB3 is involved in the expansion of the HSPC pool (page 15, lines 474-476).

1. In Figure 7E, they collected the BM reconstructed with Pfkfb3- or Rosa-KO HSPCs two months after transplantation, and then tested their resistance to 5-FU. However, the short duration of the reconstruction period makes it difficult to draw conclusions about the effects on steady-state blood cell production.

We agree that we cannot conclude from this experiment alone that PFKFB3 is completely unnecessary in steady state because, as you pointed out, the observation period of the experiment in Figure 7E is not long. We have toned down the claim by stating that PFKFB3 is only less necessary in steady-state HSCs compared to proliferative HSCs (page 15, lines 460-461).

1. PFK is allosterically activated by PFKFB, and other members of the PFKFB family could also participate in the glycolytic program. Therefore, they should investigate their function in contributing to glycolytic plasticity in HSCs during proliferation. Additionally, they should also analyze the protein expression and modification levels of other members. Although PFKFB3 is the most favorable for PFK activation, the role of other members should also be explored in HSC cell cycling to provide sufficient reasoning for choosing PFKFB3.

To further justify why we chose PFKFB3 among the PFKFB family members, we reviewed our data and the publicly available Gene Expression Commons (GEXC) 3. PFKFB3 is the most highly expressed member of the PFKFB family in HSCs (revised Figure 4F), and its expression increases with proliferation (Author response image 9). In addition to this, we have also cited the literature 4 indicating that AZ PFKFB3 26 is a PFKFB3-specific inhibitor that we used in this paper, and added a note to this point (that it is specific) (page 11, lines 327-329). Through these revisions, we sought to strengthen the rationale for Pfkfb3 as the primary target of the analysis.

**Author response image 8. sa3fig8:** 

**Author response image 9. sa3fig9:** 

1. In this study, the authors identified PRMT1 as the upstream regulator of PFKFB3 that is involved in the glycolysis activation of HSCs. However, PRMT1 is also known to participate in various transcriptional activations. Thus, it is important to determine whether PRMT1 affects glycolysis through transcriptional regulation or through its direct regulation of PFKFB3? Additionally, the authors should investigate whether PRMT1i inhibits ATP production in normal HSCs. Moreover, could we combine Figure 6I and 6J for analysis. Finally, the authors could conduct additional rescue experiments to demonstrate that the effect of PRMT1 inhibitors on ATP production can be rescued by overexpression of PFKFB3.

Although PRMT1 inhibition reduced m-PFKFB3 levels in HSCs, 5-FU treatment also reduced or did not alter *Pfkfb3* transcript levels (Figures 6B, G) and the expression of genes such as Hoxa7/9/10, Itga2b, and Nqo1, which are representative transcriptional targets of PRMT1, in proliferating HSCs after 5-FU treatment (revised Figure S9).

**Author response image 10. sa3fig10:** 

These results suggest that PRMT1 promotes PFKFB3 methylation, which increases independently of transcription in HSCs after 5-FU treatment.

A summary analysis of the original Figures 6I and 6J is shown below (revised Figure 6I).

**Author response image 11. sa3fig11:** 

Finally, we tested whether the inhibition of the glycolytic system and the decrease in ATP concentration due to PRMT1 inhibition could be rescued by the retroviral overexpression of PFKFB3. We found that PFKFB3 overexpression did not decrease the ATP concentration in HSCs due to PRMT1 inhibition (revised Figure 6J). Therefore, PFKFB3 overexpression mitigated the decrease in ATP concentration caused by PRMT1 inhibition. These data and related statements have been added to the revised manuscript (page 14, lines 427-428).

**Author response image 12. sa3fig12:** 

**Reviewer #2:**
In the manuscript Watanuki et al. want to define the metabolic profile of HSCs in stress/proliferative (myelosuppression with 5-FU), and mitochondrial inhibition and homeostatic conditions. Their conclusions are that during proliferation HSCs rely more on glycolysis (as other cell types) while HSCs in homeostatic conditions are mostly dependent on mitochondrial metabolism. Mitochondrial inhibition is used to demonstrate that blocking mitochondrial metabolism results in similar features of proliferative conditions.

The authors used state-of-the-art technologies that allow metabolic readout in a limited number of cells like rare HSCs. These applications could be of help in the field since one of the major issues in studying HSCs metabolism is the limited sensitivity of the“"standard”" assays, which make them not suitable for HSC studies.

However, the observations do not fully support the claims. There are no direct evidence/experiments tackling cell cycle state and metabolism in HSCs. Often the observations for their claims are indirect, while key points on cell cycle state-metabolism, OCR analysis should be addressed directly.

We sincerely appreciate the reviewer's constructive comments. Thank you for highlighting the importance of the highly sensitive metabolic assay developed in this study and the findings based on it. Meanwhile, the reviewer's comments have made us aware of areas where we can further improve this manuscript. In particular, in the revised manuscript, we have performed further studies to demonstrate the link between the cell cycle and metabolic state. Specifically, we further subdivided HSCs by the uptake of in vivo-administered 2-NBDG and performed cell cycle analysis. Next, HSCs after PBS or 5-FU treatment were analyzed by a Mito Stress test using the Seahorse flux analyzer, including ECAR and OCR, and a more direct relationship between the cell cycle state and the metabolic system was found. We believe that the reviewer's valuable suggestions have helped us clarify more directly the importance of the metabolic state of HSCs in response to cell cycle and stress that we wanted to show and emphasize the usefulness of the GO-ATeam2 system. Our response to "Recommendations For The Authors" is listed first, followed by our responses to all comments in "Public Review" as follows:

**(Recommendations For The Authors):**
In general, I believe it would be important:1. to directly associate cell cycle state with metabolic state. For example, by sorting HSC (+/- 5FU) based on their cell cycle state (exploiting the mouse model presented in the manuscript or by defining G0/G1/G2-S-M via Pyronin/Hoechst staining which allow to sort live cells) and follow the fate of radiolabeled glucose.

Thank you for raising these crucial points. Unfortunately, it was difficult to perform the glucose tracer analysis by preparing HSCs with different cell cycle states as you suggested due to the amount of work involved. In particular, in the 5-FU group, more than 60 mice per group were originally required for an experiment, and further cell cycle-based purification would require many times that number of mice, which we felt was unrealistic under current technical standards. As an alternative, we administered 2-NBDG to mice and fractionated HSCs at the 2-NBDG fluorescence level for cell cycle analysis. The results are shown below (revised Figure S1M). Notably, even in the PBS-treated group, HSCs with high 2-NBDG uptake were more proliferative than those with low 2-NBDG uptake and are comparable to HSCs after 5-FU treatment, although the overall population of HSCs exiting the G0 phase and entering the G1 phase increased after 5-FU treatment. In both PBS/5-FU-treated groups, these large differences in cell cycle glucose utilization suggest a direct link between HSC proliferation and glycolysis activation. If a more sensitive type of glucose tracer analysis becomes available in the future, it may be possible to directly address the reviewer's comments. We see this as a topic for the future. The descriptions of the above findings and perspectives have been added to the Results and Discussion section (page 7, lines 208-214, page 20, lines 607-610).

**Author response image 13. sa3fig13:** 

1. Use other radio labeled substrates (fatty acid, glutamate)

Thank you very much for your suggestion. While this is an essential point for future studies, we believe it is not the primary focus of the paper. We are planning another research project on tracer analysis using labeled fatty acids and glutamates, which we will report on in the near future. We have clearly stated in the Abstract and Introduction of the revised manuscript, that the focus of this study is on changes in glucose metabolism when HSCs are stressed (page 3, line 75 and 87, page 5, lines 135).

Instead, we added the following analyses of metabolic changes in fatty acids and glutamate using the GO-ATeam2 system. HSCs derived from GO-ATeam2 mice treated with PBS or 5-FU were used to measure changes in ATP concentrations after exposure to the fatty acid beta-oxidation (FAO) inhibitor etomoxir and the glutaminolysis inhibitor 6-diazo-5-oxo-L-norleucine (DON). Etomoxir was used at 100 µM, a concentration that inhibits FAO without inhibiting mitochondrial electron transfer complex I, as previously reported 5. DON was used at 2 mM, a concentration that sufficiently inhibits the enzyme as the Ki for glutaminase is 6 µM. In this experiment, etomoxir alone, DON alone, or etomoxir and DON in combination did not decrease the ATP concentration of HSCs in the PBS and 5-FU groups (revised Figures S7J–M), suggesting that FAO and glutaminolysis were not essential for ATP production in HSCs in the short term. Thus, according to the analysis using the GO-Ateam2 system, HSCs exposed to acute stresses change the efficiency of glucose utilization (accelerated glycolytic ATP production) rather than other energy sources. Since there are reports that FAO and glutaminolysis are required for HSC maintenance in the long term 5,6, compensatory pathways may be able to maintain ATP levels in the short term. A description of these points has been added to the Discussion (page 11, lines 332-344).

**Author response image 14. sa3fig14:** 

1. Include OCR analyses.

In addition to the ECAR data of the Mito Stress test (original Figures 2G–H), OCR data were added to the revised manuscript (revised Figures 2H, S3D). Compared to c-Kit+ myeloid progenitors (LKS- cells), HSC showed a similar increase in ECAR, while the decrease in OCR was relatively limited. A possible explanation for this is that glycolytic and mitochondrial metabolism are coupled in c-Kit+ myeloid progenitors, whereas they are decoupled in HSCs. This is also suggested by the glucose plus oligomycin experiment in Figures 5B, C, and S6A–D (orange lines). In summary, in HSCs, glycolytic and mitochondrial ATP production are decoupled and can maintain ATP levels by glycolytic ATP production alone, whereas in progenitors including GMPs, the two ATP production systems are constantly coupled, and glycolysis alone cannot maintain ATP concentration. We have added descriptions of these points in the Results and Discussion section (page 8, lines 240-243, page 18, lines 558-561).

**Author response image 15. sa3fig15:** 

Next, a Mito Stress test was performed using HSCs derived from PBS- or 5-FU-treated mice in the presence or absence of oligomycin (revised Figures 1G–H, S3A–B). Without oligomycin treatment, ECAR in 5-FU-treated HSCs was higher than in PBS-treated HSCs, and OCR was unchanged. Oligomycin treatment increased ECAR in both PBS- and 5-FU-treated HSCs, whereas OCR was unchanged in PBS-treated HSCs, but significantly decreased in 5-FU-treated HSCs. Changes in ECAR in response to oligomycin differed between HSC proliferation or differentiation: ECAR increased in 5-FU-treated HSCs but not in LKS- progenitors (original Figures 2G–H). This suggests a metabolic feature of HSCs in which the coupling of OXPHOS with glycolysis seen in LKS- cells is not essential in HSCs even after cell cycle entry. The results and discussion of this experiment have been added to page 7, lines 194-201 and page 18, lines 558-561.

**Author response image 16. sa3fig16:** 

1. Correlate proliferation-mitochondrial inhibition-metabolic state

We agree that it is important to clarify this point. First, OXPHOS inhibition and proliferation similarly accelerate glycolytic ATP production with PFKFB3 (Figures 4G, I, and 5F–I). Meanwhile, oligomycin treatment rapidly decreases ATP in HSCs with or without 5-FU administration (Figure 4C). These results suggest that OXPHOS is a major source of ATP production both at a steady state and during proliferation, even though the analysis medium is pre-saturated with hypoxia similar to that in vivo. This has been added to the Discussion section (page 17, lines 520-523).

1. Tune down the claim on HSCs in homeostatic conditions since from the data it seems that HSCs rely more on anaerobic glycolysis.

Thanks for the advice. The original Figures S2C, D, F, and G show that HSC is dependent on the anaerobic glycolytic system even at a steady state, so we have toned down our claims (page 7, lines 192-194).

1. For proliferative HSCs mitochondrial are key. When you block mitochondria with oligomycin there's the biggest drop in ATP.

In the revised manuscript, we have tried to highlight the key findings that you have pointed out. First, we mentioned in the Discussion (page 17, lines 523-525) that previous studies suggested the importance of mitochondria in proliferating HSCs. Meanwhile, the GO-ATeam2 and glucose tracer analyses in this study newly revealed that the glycolytic system activated by PFKFB3 is activated during the proliferative phase, as shown in Figure 4C. We also confirmed that mitochondrial ATP production is vital in proliferating HSCs, and we hope to clarify the balance between ATP-producing pathways and nutrient sources in future studies.

1. To better clarify this point authors, authors should do experiments in hypoxic conditions and compare it to oligomycin treatment and showing that mito-inhibition acts differently on HSCs (considering that all these drugs are toxic for mitochondria and induce rapidly stress responses ex: mitophagy).

We apologize for any confusion caused by not clearly describing the experimental conditions. As pointed out by the reviewer, we also recognize the importance of experiments in a hypoxic environment. All GO-ATeam2 analyses were performed in a medium saturated sufficiently under hypoxic conditions and analyzed within minutes, so we believe that the medium did not become oxygenated (page S5-S6, lines 160-163 in the Methods). Despite being conducted under such hypoxic conditions, the substantial decrease in ATP after oligomycin treatment is intriguing (original Figures 4C, 5B, 5C). The p50 value of mitochondria (the partial pressure of oxygen at which respiration is half maximal) is 0.1 kPa, which is less than 0.1% of the oxygen concentration at atmospheric pressure 7. Thus, biochemically, it is consistent that OXPHOS can maintain sufficient activity even in a hypoxic environment like the bone marrow. We are currently embarking on a study to determine ATP concentration in physiological hypoxic conditions using in vivo imaging within the bone marrow, which we hope to report in a separate project. We have discussed these points, technical limitations, and perspectives in the Discussion section (page 20, lines 610-612).

• In Figure 1 C, D, E and F, the comparison should be done as unpaired t test and the control group should not be 1 as the cells comes from different individuals.

Thank you very much for pointing this out. We have reanalyzed and revised the figures (revised Figures 1C–F)

**Author response image 17. sa3fig17:** 

• In Figure S2A, the post-sorting bar of 6PG, R5P and S7P are missing.

Metabolites below the detection threshold (post-sorting samples of 6PG, R5P, and S7P) are now indicated as N.D. (not detected) (revised Figure S2A).

**Author response image 18. sa3fig18:** 

• In the 2NBDG experiments, authors should add the appropriate controls, since it has been shown that 2NBDG cellular uptake do not correctly reflect glucose uptake (Sinclair LV, Immunometabolism 2020) (a cell type dependent variations) thus inhibitors of glucose transporters should be added as controls (cytochalasin B; 4,6-O-ethylidene-a-D-glucose) it would be quite challenging to test it in vivo but it would be sufficient to show that in vitro in the different HSPCs analyzed.

We appreciate the essential technical point raised by the reviewer. In the revised manuscript, we performed a 2-NBDG assay with cytochalasin B and phloretin as negative controls. After PBS treatment, 2-NBDG uptake was higher in 5-FU-treated HSCs compared to untreated HSCs. This increase was inhibited by both cytochalasin B and phloretin. In PBS-treated HSCs, cytochalasin B did not downregulate 2-NBDG uptake, whereas phloretin did. Although cytochalasin B inhibits glucose transporters (GLUTs), it is also an inhibitor of actin polymerization. Therefore, its inhibitory effect on GLUTs may be weaker than that of phloretin. We have revised the figure (revised Figure S1L) and added the corresponding description (page 7, lines 207-208).

**Author response image 19. sa3fig19:** 

• S5C: authors should show the cell number for each population. If there's a decreased in % in Lin- that will be reflected in all HSPCs. Comparing the proportion of the cells doesn't show the real impact on HSPCs.

Thank you for your insightful point. In the revision, we compared the numbers, not percentages, of HSPCs and found no difference in the number of cells in the major HSPC fractions in Lin-. The figure has been revised (revised Figure S6C) and the corresponding description has been added (page 10, lines 296-299).

**Author response image 20. sa3fig20:** 

Minor:1. In S1 F-G is not indicated in which day post 5FU injection is done the analysis. I assume on day 6 but it should be indicated in the figure legend and/or text.

Thank you for pointing this out. As you assumed, the analysis was performed on day 6. The description has been added to the legend of the revised Figure S1G.

1. S1K is not described in the text. What are proliferative and quiescence-maintaining conditions? The analyses are done by flow using LKS SLAM markers after culture? How long was the culture?

Thank you for your comments. First, the figure citation on line 250 was incorrect and has been corrected to Figure S1N. Regarding the proliferative and quiescence-maintaining conditions, we have previously reported on these 8. In brief, these are culture conditions that maintain HSC activity at a high level while allowing for the proliferation or maintenance of HSCs in quiescence, achieved by culturing under fatty acid-rich, hypoxic conditions with either high or low cytokine concentrations. Analysis was performed after one week of culture, with the HSC number determined by flow cytometry based on the LSK-SLAM marker. While these are mentioned in the Methods section, we have added a description in the main text to highlight these points for the reader (page 7, lines 214-217).

1. In Figure 5G, why does the blue line (PFKFB3 inhibitor) go up in the end of the real-time monitoring? Does it mean that other compensatory pathway is turned on?

As you have pointed out, we cannot rule out the possibility that other unknown compensatory ATP production pathways were activated. We have added a note in the Discussion section to address this (page 18, lines 555-556).

1. In Figure S6H&J, the reduction is marginal. Does it mean that PKM2 is not important for ATP production in HSCs?

The activity of the inhibitor is essential in the GO-ATeam2 analysis. The commercially available PKM2 inhibitors have a higher IC50 value (IC50 = 2.95 μM in this case). Nevertheless, the effect of reducing the ATP concentration was observed in progenitor cells, but not in HSCs. The report by Wang et al. 9 on the analysis using a PKM2-deficient model suggests a stronger effect on progenitor cells than on HSCs. Our results are similar to those of the previous report.

(Specific comments)Specifically, there are several major points that rise concerns about the claims:1. The gating strategy to select HSCs with enlarged Sca1 gating is not convincing. I understand the rationale to have a sufficient number of cells to analyze, however this gating strategy should be applied also in the control group. From the FACS plot seems that there are more HSCs upon 5FU treatment (Figure S1b). How that is possible? Is it because of the 20% more of cycling cells at day 6? To prove that this gating strategy still represents a pure HSC population, authors should compare the blood reconstitution capability of this population with a "standard" gated population. If the starting population is highly heterogeneous then the metabolic readout could simply reflect cell heterogeneity.

Thank you for pointing this out. First, we did not enlarge the Sca-1 gating in this study. We apologize for any confusion caused by the incomplete description. The gating of c-Kit is based on that shown by Umemoto et al (Figure EV1A) 2, who used 250 mg/kg 5-FU, so their c-Kit reduction is more pronounced than ours.

We followed this study and compared c-Kit expression in the Lin-Sca-1+CD150+CD48-EPCR+ gates to BMMNCs on day 6 after 5-FU administration (150 mg/kg). The results are shown below.

**Author response image 21. sa3fig21:** 

Since the MFI of c-Kit was downregulated, we used gating that extended the c-Kit gate to lower expression regions on day 6 after 5-FU administration (revised Figure S1C).

At other time points, LSK gating was the same as in the PBS-treated group, as noted in the Methods.

The reason why the number of HSCs appears to be higher in the 5-FU group is because most of the differentiated blood cells were lost due to 5-FU administration and the same number of cells as in the PBS group were analyzed by FACS, resulting in a relatively higher number of HSCs. The legend of Figure S1 shows that the number of HSCs in both the PBS and 5-FU groups appeared to increase because the same number of BMMNCs was obtained at the time of analysis (page S22, lines 596-598).

Regarding cellular heterogeneity, from a metabolic point of view, the heterogeneity in HSCs is rather reduced by 5-FU administration. As shown in Figure S3A–C, this is simulated under stress conditions, such as after 5-FU administration or during OXPHOS inhibition, where the flux variability in each enzymatic reaction is significantly reduced. GO-ATeam2 analysis after 5-FU treatment showed no increase in cell population variability. After 2-DG treatment, ATP concentrations in HSCs were widely distributed from 0 mM to 0.8 mM in the PBS group, while more than 80% of those in the 5-FU group were less than 0.4 mM (Figures 4B, D). HSCs may have a certain metabolic diversity at a steady state, but under stress conditions, they may switch to a more specialized metabolism with less cellular heterogeneity in order to adapt.

1. S2 does not show major differences before and after sorting. However, a key metabolite like Lactate is decreased, which is also one of the most present. Wouldn't that mean that HSCs once they move out from the hypoxic niche, they decrease lactate production? Do they decrease anaerobic glycolysis? How can quiescent HSC mostly rely on OXPHOS being located in hypoxic niche?1. Since HSCs in the niche are located in hypoxic regions of the bone marrow, would that not mimic OxPhos inhibition (oligomycin)? Would that not mean that HSCs in the niche are more glycolytic (anaerobic glycolysis)?1. In Figure 5B, the orange line (Glucose+OXPHOS inhibition) remains stable, which means HSCs prefer to use glycolysis when OXPHOS is inhibited. Which metabolic pathway would HSCs use under hypoxic conditions? As HSCs resides in hypoxic niche, does it mean that these steady-state HSCs prefer to use glycolysis for ATP production? As mentioned before, mitochondrial inhibition can be comparable at the in vivo condition of the niche, where low pO2 will "inhibit" mitochondria metabolism.

Thank you for the first half of comment 2 on the technical features of our approach. First, as you have pointed out, there is minimal variation and stable detection of many metabolites before and after sorting (Figure S2A), suggesting that isolation from the hypoxic niche and sorting stress do not significantly alter metabolite detection performance. This is consistent with a previous report by Jun et al. 10. Meanwhile, lactate levels decreased by sorting. Therefore, if the activity of anaerobic glycolysis was suppressed in stressed HSCs, it may be difficult to detect these metabolic changes with our tracer analysis. However, in this study, several glycolytic metabolites, including an increase in lactate, were detected in HSCs from 5-FU-treated mice compared with HSCs from PBS-treated mice that were similarly sorted and prepared, suggesting an increase in glycolytic activity. In other words, we may have been fortunate to detect the stress-induced activation of the glycolytic system beyond the characteristic of our analysis system that lactate levels tend to appear lower than they are. Given that damage to the bone marrow hematopoiesis tends to alleviate the low-oxygen status of the niche 11, we postulate that this upregulated aerobic glycolysis arises intrinsically in HSCs rather than from external conditions.

The second half of comment 2, and comments 7 and 10, are essential and overlapping comments and will be answered together. Although genetic analyses have shown that HSCs produce ATP by anaerobic glycolysis in low-oxygen environments 9,12, our GO-ATeam2 analysis in this study confirmed that HSCs also generate ATP via mitochondria. This is also supported by Ansó's prior findings where the knockout of the Rieske iron–sulfur protein (RISP), a constituent of the mitochondrial electron transport chain, impairs adult HSC quiescence and bone marrow repopulation 13. Bone marrow is a physiologically hypoxic environment (9.9–32.0 mmHg 11). However, the p50 value of mitochondria (the partial pressure of oxygen at which respiration is half maximal) is below 0.1% oxygen concentration at atmospheric pressure (less than 1 mmHg) 7. This suggests that OXPHOS can retain sufficient activity even under physiologically hypoxic conditions. We are currently initiating efforts to discern ATP concentrations in vivo within the bone marrow under physiological hypoxia. This will be reported in a separate project in the future. Admittedly, when we began this research, we did not anticipate the significant mitochondrial reliance of HSCs. As we previously reported, the metabolic uncoupling of glycolysis and mitochondria 12 may enable HSCs to activate only glycolysis, and not mitochondria, under stress conditions such as post-5-FU administration, suggesting a unique metabolic trait of HSCs. We have included these technical limitations and perspectives in the Discussion section (page 17, lines 520-523).

1. The authors performed challenging experiments to track radiolabeled glucose, which are quite remarkable. However, the data do not fully support the conclusions. Mitochondrial metabolism in HSCs can be supported by fatty acid and glutamate, thus authors should track the fate of other energy sources to fully discriminate the glycolysis vs mito-metabolism dependency. From the data on S2 and Fig1 1C-F, the authors can conclude that upon 5FU treatment HSCs increase glycolytic rate.1. FIG.2B-C: Increase of Glycolysis upon oligomycin treatment is common in many different cell types. As explained before, other radiolabeled substrates should be used to understand the real effect on mitochondria metabolism.

Thank you for your suggestion. While this is essential for future studies, we believe it is not the primary focus of the paper. We are planning another research project on tracer analysis using labeled fatty acids and glutamates, which we will report on in the near future. We have clearly stated in the Abstract and Introduction of the revised manuscript that the focus of this study is on changes in glucose metabolism when HSCs are stressed (page 3, line 75 and 87, page 5, lines 135).

Instead, we have added the following analyses of metabolic changes in fatty acids and glutamate using the GO-ATeam2 system: HSCs derived from GO-ATeam2 mice treated with PBS or 5-FU were used to measure changes in ATP concentrations after exposure to the fatty acid beta-oxidation (FAO) inhibitor etomoxir and the glutaminolysis inhibitor 6-diazo-5-oxo-L-norleucine (DON). Etomoxir was used at 100 µM, a concentration that inhibits FAO without inhibiting mitochondrial electron transfer complex I, as previously reported 5. DON was used at 2 mM, a concentration that sufficiently inhibits the enzyme as the Ki for glutaminase is 6 µM. In this experiment, etomoxir alone, DON alone, or etomoxir and DON in combination did not decrease the ATP concentration of HSCs in the PBS and 5-FU groups (revised Figures S7J–M), suggesting that FAO and glutaminolysis were not essential for ATP production in HSCs in the short term. Thus, according to the analysis using the GO-Ateam2 system, HSCs exposed to acute stresses change the efficiency of glucose utilization (accelerated glycolytic ATP production) rather than other energy sources. Since there are reports that FAO and glutaminolysis are required for HSC maintenance in the long term 5,6, compensatory pathways may be able to maintain ATP levels in the short term. A description of these points has been added to the Discussion (page 17, lines 525-527).

**Author response image 22. sa3fig22:** 

Fatty acid β-oxidation activity was also measured in 5-FU-treated HSCs using the fluorescent probe FAOBlue and was unchanged compared to PBS-treated HSCs (revised Figure S7N).

**Author response image 23. sa3fig23:** 

Notably, the addition of 100 µM etomoxir plus glucose and PFKFB3 inhibitors resulted in a rapid decrease in ATP concentration in HSCs (revised Figures S7O–P). This indicates that etomoxir partially mimics the effect of oligomycin, suggesting that at a steady state, OXPHOS is driven by FAO, but can be compensated by the acceleration of the glycolytic system by PFKFB3. Meanwhile, the exposure of HSCs to PFKFB3 inhibitors in addition to 2 mM DON did not reduce ATP (revised Figures S7O–P). This suggests that ATP production from glutaminolysis is limited in HSCs at a steady state.

**Author response image 24. sa3fig24:** 

These points suggest that OXPHOS is driven by fatty acids at a steady state, but unlike the glycolytic system, FAO is not further activated by HSCs after 5-FU treatment. The results of these analyses and related descriptions are included in the revised manuscript (page 11, lines 332-344).

1. In Figure S1, 5-FU leads to the induction of cycling HSCs and in figure 1, 5-FU results in higher activation of glycolysis. Would it be possible to correlate these two phenotypes together? For example, by sorting NBDG+ cells and checking the cell cycle status of these cells?

We appreciate the reviewer’s insightful comments. We administered 2-NBDG to mice and fractionated HSCs at the 2-NBDG fluorescence level for cell cycle analysis. The results are shown below (revised Figure S1M). Notably, even in the PBS-treated group, HSCs with high 2-NBDG uptake were more proliferative than HSCs with low 2-NBDG uptake and were comparable to HSCs after 5-FU treatment, although the overall population of HSCs that exited the G0 phase and entered the G1 phase increased after 5-FU treatment. In both PBS/5-FU-treated groups, these profound differences in cell cycle glucose utilization suggest a direct link between HSC proliferation and glycolysis activation. Descriptions of the above findings and perspectives have been added to the Results and Discussion section (page 7, lines 208-214, page 20, lines 607-610).

**Author response image 25. sa3fig25:** 

1. Why are only ECAR measurements (and not OCR measurements) shown? In Fig.2G, why are HSCs compared with cKit+ myeloid progenitors, and not with MPP1? The ECAR increased observed in HSC upon oligomycin treatment is shared with many other types of cells. However, cKit+ cells have a weird behavior. Upon oligo treatment cKit+ cells decrease ECAR, which is quite unusual. The data of both HSCs and cKit+ cells could be clarified by adding OCR curves. Moreover, it is recommended to run glycolysis stress test profile to assess the dependency to glycolysis (Glucose, Oligomycin, 2DG).

In addition to the ECAR data of the Mito Stress test (original Figures 2G–H), OCR data were added in the revised manuscript (revised Figures 2H, S3D). Compared to c-Kit+ myeloid progenitors (LKS- cells), HSC exhibited a similar increase in ECAR, while the decrease in OCR was relatively limited. This may be because glycolytic and mitochondrial metabolism are coupled in c-Kit+ myeloid progenitors, whereas they are decoupled in HSCs. This is also suggested by the glucose plus oligomycin experiment in Figures 5B, C, and S6A–D (orange lines). In summary, in HSCs, glycolytic and mitochondrial ATP production are decoupled and can maintain ATP levels by glycolytic ATP production alone, whereas in progenitors including GMPs, the two ATP production systems are constantly coupled, and glycolysis alone cannot maintain the ATP concentration. While we could not conduct a glycolysis stress test, we believe that PFKFB3-dependent glycolytic activation, which is evident in the oligomycin+glucose+PFKFB3i experiment, is only apparent in HSCs when subjected to glucose+oligomycin treatment (original Figures 5F–I). We have added descriptions of these points in the Results and Discussion section (page 8, lines 240-243, page 18, lines 558-561).

**Author response image 26. sa3fig26:** 

FIG.3 A-C. As mentioned previously, the flux analyses should be integrated with data using other energy sources. If cycling HSCs are less dependent to OXPHOS, what happen if you inhibit OXHPHOS in 5-FU condition? Since the authors are linking OXPHOS inhibition and upregulation of Glycolysis to increase proliferation, do HSCs proliferate more when treated with oligomycin?

First, please see our response to comments 3 and 5 regarding the first part of this comment about the flux analysis of other energy sources. According to the analysis using the GO-Ateam2 system, stressed HSCs change the efficiency of glucose utilization (accelerated glycolytic ATP production) rather than other energy sources. The change in ATP concentration after OXPHOS inhibition for 5-FU-treated HSCs is shown in Figures 4C and E, suggesting that the activity of OXPHOS itself does not increase. HSCs after oligomycin treatment and HSCs after 5-FU treatment are similar in that they activate glycolytic ATP production. However, inhibition of OXPHOS did not induce the proliferation of HSCs (original Figure S1K). This suggests that proliferation activates glycolysis and not that activation of the glycolytic system induces proliferation. This similarity and dissimilarity of glycolytic activation upon proliferation and OXPHOS inhibition is discussed in the Discussion section (page 16-17, lines 505-515).

1. FIG.4 shows that in vivo administration of radiolabeled glucose especially marks metabolites of TCA cycle and Glycolysis. The authors interpret enhanced anaerobic glycolysis, but I am not sure this is correct; if more glycolysis products go in the TCA cycle, it might mean that HSC start engaging mitochondrial metabolism. What do the authors think about that?

Thank you for pointing this out. We believe that the data are due to two differences in the experimental features between in vivo (Figure S5) and in vitro (Figures 1 and S2) tracer analysis. The first difference is that in in vivo tracer analysis, unlike in vitro, all cells can metabolize U-13C6-glucose. Another difference is that after glucose labeling in vivo, it takes approximately 120–180 minutes to purify HSCs to extract metabolites, and processing on ice may result in a gradual progression of metabolic reactions within HSCs. As a result, in vivo tracer analysis may detect an increased influx of labeled carbon derived from U-13C6-glucose into the TCA cycle over an extended period. However, it is difficult to interpret whether this influx of labeled carbon is derived from the direct influx of glycolysis or the re-uptake by HSCs of metabolites that have been metabolized to other metabolites in other cells. Meanwhile, as shown in Figure 4C using the GO-ATeam2 system, ATP production from mitochondria is not upregulated by 5-FU treatment. This suggests that even if the direct influx from glycolysis into the TCA cycle is increased, the rate of ATP production does not exceed that of glycolysis. Despite these technical caveats in interpretation, the results of in vivo and in vitro tracer analyses are considered essential. In particular, we consider the increased labeling of metabolites involved in glycolysis and nucleotide synthesis to be crucial. We have added a discussion of these points, including experimental limitations (page 17-18, lines 530-545).

1. FIG.4: the experimental design is not clear. Are BMNNCs stained and then put in culture? Is it 6-day culture or BMNNCs are purified at day 6 post 5FU? FIG-4B-C The difference between PBS vs 5FU conditions are the most significant; however, the effect of oligomycin in both conditions is the most dramatic one. From this readout, it seems that HSCs are more dependent on mitochondria for energy production both upon 5FU treatment and in PBS conditions.

We apologize for the incomplete description of the experimental details. The experiment involved dispensing freshly stained BMMNC with surface antigens into the medium and immediately subjecting them to flow cytometry analysis. For post-5-FU treatment HSCs, mice were administered with 5-FU (day 1), and freshly obtained BMMNCs were analyzed on day 6. The analysis of HSCs and progenitors was performed by gating each fraction within the BMMNC (original Figure S5A). We have added these details to ensure that readers can grasp these aspects more clearly (page S5, lines 155-158).

As pointed out by the reviewer, we understand that HSCs produce more ATP through OXPHOS. However, ATP production by glycolysis, although limited, is observed under steady-state conditions (post-PBS treatment HSC), and its reliance increases during the proliferation phase (post-5-FU treatment HSC) (original Figures 4B, D). Until now, discussions on energy production in HSCs have focused on either glycolysis or mitochondrial functions. However, with the GO-ATeam2 system, it has become possible for the first time to compare their contributions to ATP production and evaluate compensatory pathways. As a result, it became evident that while OXPHOS is the main source of ATP production, the reliance on glycolysis plastically increases in response to stress. This has led to a better understanding of HSC metabolism. These points are included in the Discussion as well (page 16, lines 479-488).

1. FIG.6H should be extended with cell cycle analyses. There are no differences between 5FU and ctrl groups. If 5FU induces HSCs cycling and increases glycolysis I would expect higher 2-NBDG uptake in the 5FU group. How do the authors explain this?

Thank you for your comments. In the original Figure 6H, we found that 2-NBDG uptake correlated with m-PFKFB3 levels in both the 5-FU and PBS groups. m-PFKFB3 levels remained low in the few HSCs with low 2-NBDG uptake in the 5-FU group.

In the revised manuscript, to directly relate glucose utilization to the cell cycle, we administered 2-NBDG to mice and fractionated HSCs at the 2-NBDG fluorescence level for cell cycle analysis. The results are shown below (revised Figure S1M). Notably, even in the PBS-treated group, HSCs with high 2-NBDG uptake were more proliferative than those with low 2-NBDG uptake and are comparable to HSCs after 5-FU treatment, although the overall population of HSCs that exited the G0 phase and entered the G1 phase increased after 5-FU treatment. The large differences in glucose utilization per cell cycle observed in both PBS/5-FU-treated groups suggest a direct link between HSC proliferation and glycolysis activation. Descriptions of the above findings have been added to the Results and Discussion (page 7, lines 208-214, page 20, lines 607-610).

**Author response image 27. sa3fig27:** 

1. In S7 the experimental design is not clear. What are quiescent vs proliferative conditions? What does it mean "cell number of HSC-derived colony"? Is it a CFU assay? Then you should show colony numbers. When HSCs proliferate, they need more energy thus inhibition of metabolism will impact proliferation. What happens if you inhibit mitochondrial metabolism with oligomycin?

Regarding the proliferative and quiescence-maintaining conditions, we have previously reported on these 8. In brief, these are culture conditions that maintain HSC activity at a high level while allowing for the proliferation or maintenance of HSCs in quiescence, achieved by culturing under fatty acid-rich, hypoxic conditions with either high or low cytokine concentrations. Analysis was performed after one week of culture, with the HSC number determined by flow cytometry based on the LSK-SLAM marker. While these are mentioned in the Methods section, we have added a description in the main text to highlight these points for the reader (page 7, lines 214-217).

In vitro experiments with the oligomycin treatment of HSCs showed that OXPHOS inhibition activates the glycolytic system, but does not induce HSC proliferation (original Figure S1K). This suggests that proliferation activates glycolysis and not that activation of the glycolytic system induces proliferation. This similarity and dissimilarity of glycolytic activation upon proliferation and OXPHOS inhibition is discussed in the Discussion (page 16-17, lines 505-515).

1. In FIG 7 since homing of HSCs is influenced by the cell cycle state, should be important to show if in the genetic model for PFKFB3 in HSCs there's a difference in homing efficiency.

In response to the reviewer's comments, we knocked out *Pfkfb3* in HSPCs derived from Ubc-GFP mice, transplanted 200,000 HSPCs into recipients (C57BL/6 mice) post-8.5Gy irradiation, and harvested the bone marrow of recipients after 16 h to compare homing efficiency (revised Figure S10H). Even with the knockout of *Pfkfb3*, no significant difference in homing efficiency was detected compared to the control group (Rosa knockout group). These results suggest that the short-term reduction in chimerism due to *Pfkfb3* knockout is not due to decreased homing efficiency or cell death by apoptosis (Figure 7K) but a transient delay in cell cycle progression. We have added descriptions regarding these findings in the Results and Discussion sections (page 15, lines 470-471, page 19, lines 576-578).

**Author response image 28. sa3fig28:** 

1. Yamamoto M, Kim M, Imai H, Itakura Y, Ohtsuki G. Microglia-Triggered Plasticity of Intrinsic Excitability Modulates Psychomotor Behaviors in Acute Cerebellar Inflammation. Cell Rep. 2019;28(11):2923-2938 e2928.

2. Umemoto T, Johansson A, Ahmad SAI, et al. ATP citrate lyase controls hematopoietic stem cell fate and supports bone marrow regeneration. EMBO J. 2022:e109463.

3. Seita J, Sahoo D, Rossi DJ, et al. Gene Expression Commons: an open platform for absolute gene expression profiling. PLoS One. 2012;7(7):e40321.

4. Boyd S, Brookfield JL, Critchlow SE, et al. Structure-Based Design of Potent and Selective Inhibitors of the Metabolic Kinase PFKFB3. J Med Chem. 2015;58(8):3611-3625.

5. Ito K, Carracedo A, Weiss D, et al. A PML–PPAR-δ pathway for fatty acid oxidation regulates hematopoietic stem cell maintenance. Nat Med. 2012;18(9):1350-1358.

6. Oburoglu L, Tardito S, Fritz V, et al. Glucose and glutamine metabolism regulate human hematopoietic stem cell lineage specification. Cell Stem Cell. 2014;15(2):169-184.

7. Gnaiger E, Mendez G, Hand SC. High phosphorylation efficiency and depression of uncoupled respiration in mitochondria under hypoxia. Proc Natl Acad Sci U S A. 2000;97(20):11080-11085.

8. Kobayashi H, Morikawa T, Okinaga A, et al. Environmental Optimization Enables Maintenance of Quiescent Hematopoietic Stem Cells Ex Vivo. Cell Rep. 2019;28(1):145-158 e149.

9. Wang YH, Israelsen WJ, Lee D, et al. Cell-state-specific metabolic dependency in hematopoiesis and leukemogenesis. Cell. 2014;158(6):1309-1323.

10. Jun S, Mahesula S, Mathews TP, et al. The requirement for pyruvate dehydrogenase in leukemogenesis depends on cell lineage. Cell Metab. 2021;33(9):1777-1792 e1778.

11. Spencer JA, Ferraro F, Roussakis E, et al. Direct measurement of local oxygen concentration in the bone marrow of live animals. Nature. 2014;508(7495):269-273.

12. Takubo K, Nagamatsu G, Kobayashi CI, et al. Regulation of glycolysis by Pdk functions as a metabolic checkpoint for cell cycle quiescence in hematopoietic stem cells. Cell Stem Cell. 2013;12(1):49-61.

13. Anso E, Weinberg SE, Diebold LP, et al. The mitochondrial respiratory chain is essential for haematopoietic stem cell function. Nat Cell Biol. 2017;19(6):614-625.